# Covariance-Aware Private Mean Estimation Without Private Covariance Estimation

**Gavin Brown**
Department of Computer Science
Boston University
grbrown@bu.edu

**Marco Gaboardi**
Department of Computer Science
Boston University
gaboardi@bu.edu

**Adam Smith**
Department of Computer Science
Boston University
ads22@bu.edu

**Jonathan Ullman**
Khoury College of Computer Sciences
Northeastern University
jullman@ccs.neu.edu

**Lydia Zakynthinou**
Khoury College of Computer Sciences
Northeastern University
zakynthinou.l@northeastern.edu

## Abstract

We present two sample-efficient differentially private mean estimators for $d$-dimensional (sub)Gaussian distributions with unknown covariance. Informally, given $n \gtrsim d/\alpha^2$ samples from such a distribution with mean $\mu$ and covariance $\Sigma$, our estimators output $\tilde{\mu}$ such that $\|\tilde{\mu} - \mu\|_\Sigma \leq \alpha$, where $\| \cdot \|_\Sigma$ is the *Mahalanobis distance*. All previous estimators with the same guarantee either require strong a priori bounds on the covariance matrix or require $\Omega(d^{3/2})$ samples.

Each of our estimators is based on a simple, general approach to designing differentially private mechanisms, but with novel technical steps to make the estimator private and sample-efficient. Our first estimator samples a point with approximately maximum Tukey depth using the exponential mechanism, but restricted to the set of points of large Tukey depth. Proving that this mechanism is private requires a novel analysis. Our second estimator perturbs the empirical mean of the data set with noise calibrated to the empirical covariance, without releasing the covariance itself. Its sample complexity guarantees hold more generally for subgaussian distributions, albeit with a slightly worse dependence on the privacy parameter. For both estimators, careful preprocessing of the data is required to satisfy differential privacy.

## 1 Introduction

Although the goal of statistics and machine learning is to infer properties of a population, there is a growing awareness that many statistical estimators and trained models reveal a concerning amount of information about their data set, which leads to significant concerns about the *privacy* of the individuals who have contributed sensitive information to that data set. These privacy violations have been demonstrated repeatedly via *reconstruction attacks* [21, 30, 27, 46], *membership-inference attacks* [37, 52, 10, 32, 53, 59], and instances of unwanted *memorization of training data* [13, 14, 34, 7]. In order to realize the benefits of analyzing sensitive data sets, it is crucial to develop statistical

35th Conference on Neural Information Processing Systems (NeurIPS 2021).

estimators and machine learning algorithms that make accurate inferences about the population but also protect the privacy of the individuals who contribute data.

In this work we study statistical estimators that satisfy a condition called *differential privacy* [29], which has become the standard criterion for individual privacy in statistics and machine learning. Informally, a differentially private algorithm guarantees that no attacker, regardless of their background knowledge or resources, can infer much more about any individual than they could have learned had that individual never contributed to the data set [45]. A long body of work shows that differential privacy is compatible with a wide range of tasks in statistics and machine learning, and it is now seeing deployment at companies like Google [33, 6, 57], Apple [3], Facebook [55] and LinkedIn [51], as well as statistical agencies like the U.S. Census Bureau [1, 36].

**Background: Differentially Private Mean Estimation.** We revisit differentially private estimators for one of the most fundamental tasks in all of statistics and machine learning—given $x_1, \ldots, x_n \in \mathbb{R}^d$ sampled i.i.d. from a distribution with mean $\mu \in \mathbb{R}^d$ and covariance $\Sigma \in \mathbb{R}^{d \times d}$, estimate the mean $\mu$. Mean estimation is not only an essential summary statistic in its own right, but also a building block for more sophisticated tasks like regression and stochastic optimization.

Without privacy constraints, the natural solution is to output the empirical mean $\mu_x = \frac{1}{n} \sum_i x_i$. The natural way to state the sample-complexity guarantee of the empirical mean is

$$n \gtrsim \frac{d}{\alpha^2} \implies \|\mu_x - \mu\|_\Sigma \le \alpha \,,$$

where $\gtrsim$ hides a universal multiplicative constant, and the accuracy guarantee holds with large constant probability (say, 0.99). Importantly, $\|\mu_x - \mu\|_\Sigma = \|\Sigma^{-1/2}(\mu_x - \mu)\|_2$ is the error in *Mahalanobis distance* scaled to the covariance $\Sigma$. Bounding the error in Mahalanobis distance implies that in every direction $v$, the squared error is proportional to the variance $v^T \Sigma v$ in that direction. The Mahalanobis distance is the natural way to measure the error for mean estimation, since it tightly captures the uncertainty about the true mean and is preserved under affine transformations.

Unfortunately, in high dimensions, releasing the empirical mean leads to concrete privacy breaches [32, 41]. A natural question is thus: can we design a differentially private mean estimator that performs nearly as well as the empirical mean?

Without making additional assumptions, the answer turns out to be no—every differentially private estimator incurs a large overhead in sample complexity compared to the empirical mean [42]. However, it is known that if we further assume that the distribution satisfies some additional concentration properties, we can do much better [54, 44, 41, 42]. In this work, we focus on one class of well-concentrated distributions, those that are *Gaussian* (or, in some of our results, *subgaussian*). Although assuming Gaussian data is restrictive, it is a natural starting point for understanding the complexity of estimation when the distribution is not pathological. Moreover, even in the Gaussian case, one cannot obtain error comparable to that of the empirical mean unless $n \gtrsim d$ [10, 32, 41], so we will also focus on the case where the sample size is at least as large as the dimension.

For Gaussian data, if the analyst has prior information about the covariance matrix in the form of a matrix $A$ and bound $\kappa \ge 1$ such that $A \preceq \Sigma \preceq \kappa A$,[1] then there is a folklore private estimator $\mathcal{A}(x)$, based on a line of work initiated by Karwa and Vadhan [44, 41, 5, 2], that finds an approximate range for the data, truncates the points to within that range, and runs the Gaussian mechanism on the resulting empirical mean. This estimator achieves

$$n \gtrsim \frac{d}{\alpha^2} + \frac{d\sqrt{\kappa}}{\alpha\varepsilon} \implies \|\mathcal{A}(x) - \mu\|_\Sigma \le \alpha, \tag{1}$$

where $\varepsilon$ is the *privacy parameter* controlling the level of privacy (see Definition 2.2), with stronger privacy as $\varepsilon \to 0$.[2] Here $\gtrsim$ hides a universal multiplicative constant and polylogarithmic factors of $d, \frac{1}{\delta}, \frac{1}{\varepsilon}$ and $\frac{1}{\alpha}$; the accuracy guarantee holds with large constant probability. We can interpret this result as showing that the additional cost of privacy is small provided that the user has a strong a priori

---

[1]Given two covariance matrices $A, B \in \mathbb{R}^{d \times d}$, the notation $A \preceq B$ indicates that in every direction $v \in \mathbb{R}^d$ the variance $v^T B v$ is at least as large as $v^T A v$. More generally, $A \preceq B$ iff $B - A$ is positive semidefinite.

[2]To simplify the discussion, we focus only on the $\varepsilon$ parameter, although our results and many of those we discuss require relaxations of differential privacy such as approximate [28] or concentrated [26, 8] differential privacy, which have different parameterizations.

bound on the covariance so that $\kappa$ is small (e.g. $\kappa$ is a constant), and also that the privacy guarantee is not too strong (e.g. $\varepsilon \geq \alpha$). In particular, setting $\kappa = 1$ corresponds to the *known-covariance* setting, where the guarantee in (1) is known to be minimax optimal up to polylogarithmic factors [32, 41] among all differentially private estimators.

However, the sample complexity in (1) grows asymptotically with $\sqrt{\kappa}$, a large price to pay for the user's uncertainty. Intuitively, this degradation arises because the algorithm perturbs the empirical mean $\mu_x$ with noise from a *spherical* Gaussian distribution, whose magnitude must be proportional to the largest variance in any direction, so the noise is unnecessarily large in the directions with small variance. In contrast, when the user is very uncertain about the covariance, there are estimators with a weaker dependence on $\kappa$ but a *superlinear* dependence on the dimension. In particular, there is an estimator [41, Theorem 4.3] with an error guarantee of the form

$$ n \gtrsim \frac{d}{\alpha^2} + \frac{d}{\alpha\varepsilon} + \frac{d^{3/2}\log^{1/2}\kappa}{\varepsilon} \implies \|\mathcal{A}(x) - \mu\|_\Sigma \leq \alpha. \tag{2} $$

Here $\gtrsim$ hides a universal multiplicative constant and logarithmic factors of $d, \frac{1}{\varepsilon}, \frac{1}{\alpha}, \log\kappa$, and $\rho$, where $\|\mu\|_2 \leq \rho$; the accuracy guarantee holds with large constant probability. Without any prior information about the covariance, the best known approach is to estimate the mean by learning the entire distribution—both mean and covariance—which is the more difficult task considered in [2, Theorem 4.6]. Doing so incurs an even worse dependence on the dimension:

$$ n \gtrsim \frac{d^2}{\alpha^2} + \frac{d^2}{\alpha\varepsilon} \implies \|\mathcal{A}(x) - \mu\|_\Sigma \leq \alpha. \tag{3} $$

Here $\gtrsim$ hides a universal multiplicative constant and logarithmic factors of $\frac{1}{\delta}$ and $\frac{1}{\alpha}$; the accuracy guarantee holds with large constant probability.

**The Covariance-Estimation Bottleneck.** The bottleneck in the algorithms above is privately obtaining a good spectral approximation to the covariance, i.e. a matrix $A$ such that $A \preceq \Sigma \preceq 2A$. With such an estimate, one can apply the known-covariance approach in (1). Without privacy constraints, the empirical covariance will have this spectral-approximation property when the sample size is $n \gtrsim d$. However, all known private covariance estimators require $n = \Omega(d^{3/2})$ samples, and there is evidence that this is an inherent limitation, as $\Omega(d^{3/2})$ samples are necessary for solving this task for a worst-case data distribution [31].

**Our Work: Sample-Efficient Private Mean Estimation.** We circumvent this apparent difficulty of covariance estimation by designing an algorithm that adapts the noise it adds to the distribution's covariance without actually providing an explicit covariance estimate, nearly matching the optimal sample complexity (1) for the known-covariance setting.

**Theorem 1.1** (Informal). *For $\alpha \leq 1$, there is an $(\varepsilon, \delta)$-differentially private estimator $\mathcal{A}(\cdot)$ such that if $x = (x_1, \ldots, x_n)$ are sampled from $\mathcal{N}(\mu, \Sigma)$ for unknown $\mu$ and $\Sigma$ of full rank,*

$$ n \gtrsim \frac{d}{\alpha^2} + \frac{d}{\alpha\varepsilon} + \frac{\log(1/\delta)}{\varepsilon} \implies \|\mathcal{A}(x) - \mu\|_\Sigma \leq \alpha. $$

*The above guarantee holds with high probability over the sample $x$ and the randomness of $\mathcal{A}$. Here $\gtrsim$ hides a universal multiplicative constant and a logarithmic factor of $\frac{1}{\alpha}$.*

For the formal statement, see Theorem 3.2. Our estimator is based on privately sampling a point of large *Tukey depth*. Tukey depth generalizes the notion of quantiles to multiple dimensions; it is known to be a good robust estimator of the Gaussian mean. The natural way to sample such a point privately is to use the *exponential mechanism* (as in the concurrent work of [47]), but sampling from a distribution over the entire domain $\mathbb{R}^d$ will not have finite sample complexity. Our innovation is to sample from a *data-dependent* domain consisting only of points of large Tukey depth, which necessitates careful preprocessing and privacy analysis.

We emphasize that the sample complexity of this estimator is optimal up to polylogarithmic factors. However, the estimator is not computationally efficient. An interesting open problem is to design an estimator matching the guarantee of Theorem 1.1 with running time polynomial in the dimension.

**Beyond Gaussian Distributions.** A natural question is how much the assumption of Gaussian data can be relaxed without blowing up the sample complexity. Our second result is an alternative

estimator, based on a completely different technique, that will give a similar guarantee for any distribution with *subgaussian tails*. For our purposes, we say that $P$ with mean $\mu$ and covariance $\Sigma$ is subgaussian if, for every direction $u \in \mathbb{R}^d$, the tails of the distribution decay as fast as a univariate normal distribution with mean $u^T\mu$ and variance $Cu^T\Sigma u$ for some constant $C$. That is, for every $\lambda$, $\mathbb{E}[e^{\lambda u^T(P-\mu)}] \le e^{C\lambda^2(u^T\Sigma u)/2}$. More generally, our estimator works for any distribution such that the empirical covariance matrix converges rapidly to the population covariance matrix and typical samples are close to the mean in Mahalanobis distance.

**Theorem 1.2** (Informal). *For $\alpha \le 1$, there is an $(\varepsilon, \delta)$-differentially private estimator $\mathcal{A}(\cdot)$ such that if $x = (x_1, \ldots, x_n)$ are sampled from any subgaussian distribution with unknown mean $\mu$ and unknown covariance $\Sigma$ of full rank,*

$$n \gtrsim \frac{d}{\alpha^2} + \frac{d\log(1/\delta)}{\alpha\varepsilon^2} \implies \|\mathcal{A}(x) - \mu\|_\Sigma \le \alpha.$$

*The above guarantee holds with high probability over the sample $x$ and the randomness of $\mathcal{A}$. Here $\gtrsim$ hides a universal multiplicative constant and polylogarithmic factors of $d, \frac{1}{\delta}, \frac{1}{\varepsilon}$, and $\frac{1}{\alpha}$.*

For the formal statement, see Theorem 4.4. This estimator is based on another simple approach—we perturb the empirical mean $\mu_x$ with noise scaled to $\Sigma_x$, where $\Sigma_x$ is the exact (not private) empirical covariance. We show that this approach satisfies differential privacy if the data set satisfies certain concentration properties, which we enforce using a careful preprocessing step.

Both of our estimators generalize beyond Gaussian distributions in different directions, not fully captured by our theorems. Although the Tukey depth estimator will only return an approximation to the mean when the distribution is symmetric and will not generalize to arbitrary subgaussian distributions, it returns an approximate *median* for distributions satisfying some natural regularity conditions. In contrast, the empirically rescaled estimator generalizes to distributions that are well-concentrated, in the sense that typical samples from the distribution are close to the mean in Mahalanobis distance with respect to the empirical covariance, which captures much more than just subgaussian distributions. Exploring the extent to which each estimator can be generalized is an interesting direction for future work.

**Limitations.** Our estimators are not computationally efficient. We provide finite implementations which construct and search over a sufficiently fine grid, resulting in running time exponential in the dimension $d$ and sample size $n$. We also stress that the error guarantees of our estimators only hold for families of distributions which satisfy certain regularity conditions, as described above. We leave these two interesting directions of improvement to future work.

**Societal Impact.** Since our algorithms are inefficient, they will not see practical use without further research. More broadly, our work serves the goal of increasing the array of differentially private algorithms, and thereby enables safer access to privacy-sensitive data. Although such tools might support corporations and governments in harmful data-collection efforts, we believe that the existence of these tools is beneficial on balance.

## 1.1 Techniques

**Tukey-depth Mechanism.** Our first algorithm adapts a well-known approach to estimating the location of a distribution with differential privacy. Briefly, we sample from the distribution defined by the exponential mechanism [50] based on the Tukey depth, but restricted to a data-dependent set of possible outputs—those points with Tukey depth at least $\frac{1}{4}$. To ensure differential privacy, we add a private check that the data set is "safe," which we perform before running the main mechanism.

In more detail, our starting point is the *exponential mechanism* [50]. In this context, the exponential mechanism samples a point $y \in \mathbb{R}^d$ from the distribution with probability density roughly proportional to $\exp(-\varepsilon q(x; y))$, where $q(x; y)$ is a score function that indicates how good a match $y$ is for the data set at hand. To instantiate the mechanism, one must choose (i) a score function that rewards values $y$ that are close to mean $\mu$ in the unknown Mahalanobis metric, and (ii) a set of candidate values $y$ from which to sample. For (i), we choose $q(x; y) = nT_x(y)$ where $T_x$ is the *Tukey depth* of a point, defined as

$$T_x(y) = \frac{1}{n} \cdot \min_{v \in \mathbb{R}^d} \left| \left\{ x_i \in x : \langle x_i, v \rangle \ge \langle y, v \rangle \right\} \right|. \tag{4}$$

For normally distributed data, the Tukey depth ranges from 0 (outside the convex hull of the data points) to about $1/2$ (near the mean $\mu$). The point of maximal Tukey depth, called the *Tukey median*, is well-known as a robust estimator of the mean of a Gaussian distribution. In general, the expectation of Tukey depth over the draw of the data can be cleanly described in terms of the Gaussian cumulative distribution function. See the supplementary material for further technical details.

Using the exponential mechanism with Tukey depth as the score function is a well-established idea. In one dimension, it is now the standard algorithm for approximating the median (e.g. [54]), and its high-dimensional variant was studied in previous [43] and concurrent [47] work.[3] However, on its own, it is not sufficient for our needs. The challenge is in specifying the set of potential outputs $y$ from which we sample (step (ii) above). In order to reliably output a value $y$ such that $\|y - \mu\|_\Sigma$ is small, we must sample from a set of outputs with $\Sigma$-norm that is not too large. For that, however, it would seem that one needs a rough approximation to $\Sigma$, which is exactly what we want to avoid.

We circumvent the barrier by sampling from a data-defined set *without releasing a description of that set*. Specifically, consider the algorithm which samples from the exponential mechanism restricted to points with Tukey depth at least $1/4$. A standard concentration argument shows that this set is roughly the ellipsoid $\{y : \|y - \mu\|_\Sigma \leq c\}$ for a modest constant $c$. Running the exponential mechanism on this set returns a good approximation to the mean (with $\Sigma$-norm $o(1)$) when $n = \omega(d/\varepsilon)$.

This gives us an accurate algorithm, dubbed $\mathcal{M}$. The remaining challenge is that $\mathcal{M}$ is not, on its own, differentially private. Specifically, there are data sets $x$ for which the volume of the set of Tukey-depth-$1/4$ points changes drastically when a small number of records in $x$ are changed. To address this, we identify a set of *safe* data sets $x$—these are data sets such that $\mathcal{M}$ behaves similarly on all data sets $x'$ that are neighbors of $x$. We show that normally distributed data sets are typically safe and, furthermore, require many insertions or deletions of records to be made unsafe. This allows us to apply the *propose-test-release* (PTR) framework of [25] to obtain an algorithm that is accurate for nicely distributed data and differentially private in the worst case.

Our modification of the exponential mechanism is quite general. It is similar in flavor to the GAP-MAX variant [11, 58, 15–17] as well as the top-$k$-of-$k'$ approach of [24]. However, we do not know how to obtain our results using those variants since they are specific to the discrete setting and appear to require knowledge of the volume of the level sets of the score function. Such knowledge is not obviously available in our setting.

**Empirically Rescaled Gaussian Mechanism.** The well-known *Gaussian mechanism* perturbs the empirical mean $\mu_x$ with noise drawn from $\mathcal{N}(0, \sigma^2\mathbb{I})$, for a scale parameter $\sigma$ that is chosen based on a priori information about the data. In particular, $\sigma^2$ must scale linearly with $\|\Sigma\|_2^2$, the maximum variance in any direction. Since the noise is spherical, the error will be too large in directions with small variance, and so this mechanism cannot in general achieve a good estimate in Mahalanobis distance.

Our approach relies on the following simple idea: If the data set $x$ is drawn i.i.d. from $\mathcal{N}(\mu, \Sigma)$, and the number of samples is a bit larger than the dimension $d$, then the empirical covariance $\Sigma_x$ is a good approximation to the true covariance in spectral norm. When this holds, perturbing $\mu_x$ with noise drawn from $\mathcal{N}(0, C^2\Sigma_x)$ for $C \ll \frac{1}{\sqrt{d}}$ will be a good estimate of the mean in Mahalanobis distance. Thus, we want to understand when perturbing $\mu_x$ in this way can be made differentially private.

Adding noise from $\mathcal{N}(0, C^2\Sigma_x)$ will not be private for worst-case data sets. To see this, consider a pair of adjacent data sets, one of which lies in a proper subspace of dimension $d - 1$ and the other of which has full rank. For one of these data sets, our mean estimate will always lie in the proper subspace, while for the other it will lie outside of this subspace with probability 1, making the two cases easy to distinguish.

---

[3]We became aware of Liu et al.'s work [47] while we were working on this project. Our empirically rescaled Gaussian algorithm is entirely independent of their work, but the presentation and parts of the analysis of our Tukey-based algorithm were influenced by their approach.

Liu et al. consider, among other algorithms, a version of the Tukey depth algorithm where one samples from a fixed box whose dimensions are determined by a priori bounds on the covariance matrix. We analyze a more complex procedure, where the set from which one samples is data-dependent. Liu et al. aim to solve a different problem from the one we address here, but the two analyses overlap (notably in volume computations and concentration arguments that relate empirical Tukey depth to the underlying distribution).

Our main observation is that such pathological examples should not arise when the data sets are sampled from a distribution, such as a Gaussian, that satisfies strong *concentration properties*. For example, if $x$ and $x'$ are adjacent data sets of i.i.d. samples from the same Gaussian, then $\mu_x$ and $\mu_{x'}$ will be similar, as will $\Sigma_x$ and $\Sigma_{x'}$. To take advantage of these nice distributions, we define a family of "good data sets" that captures certain properties of typical samples from a Gaussian. Roughly, a data set $x$ is *good* if $\Sigma_x$ is invertible and, for every $x_i$, $\|x_i - \mu_x\|_{\Sigma_x} \lesssim \sqrt{d \log n}$. Our main technical contribution is to show that if $x$ and $x'$ differ on a small number of samples, and both data sets are good, then their empirical means and empirical covariances are close. Thus, $\mu_x + \mathcal{N}(0, C^2\Sigma_x)$ and $\mu_{x'} + \mathcal{N}(0, C^2\Sigma_{x'})$ will be indistinguishable in the sense required for $(\varepsilon, \delta)$-differential privacy.

However, we need to define our estimator on data sets that are *not* good in such a way that the estimator will be differentially private in the worst case. To do so, we privately test whether the input data set lies close to the good set and, if needed, we project the data into the family of good data sets. This preprocessing step will have no effect when the data is Gaussian, and any pair (Gaussian or not) of adjacent data sets $x$ and $x'$ will be mapped to a pair of good data sets $\tilde{x}$ and $\tilde{x}'$ that differ on a small number of examples.

This projection step is stated abstractly in Algorithm 2 as finding the minimizer over an infinite family of data sets. In the supplementary material, we present a concrete, exponential-time algorithm that searches over a discrete grid of candidate datasets. We leave the task of identifying more efficient algorithms as an interesting problem for future work.

## 1.2 Additional Related Work

**Differentially Private Mean and Covariance Estimation.** The line of work most relevant to ours was initiated by Karwa and Vadhan [44], who established optimal private mean and variance estimators of univariate Gaussians with sample complexity $\tilde{O}(1/\alpha^2 + 1/\alpha\varepsilon)$, without requiring a priori bounds on the parameters. Previously, Smith [54] gave estimators for asymptotically normal statistics (which include the mean of a Gaussian) with optimal convergence rates for a certain range of privacy parameters. In the multivariate setting, a series of works [41, 11, 2] gives algorithms for Gaussian mean estimation with known covariance that have a near-optimal sample complexity of $\tilde{O}(d/\alpha^2 + d/\alpha\varepsilon)$. We note that [2] obtain the best bound among these works, but the guarantees of the estimator from [41] extend naturally to subgaussian distributions as well. [12] also studied mean and covariance estimation of subgaussian distributions, but their setting requires strong a priori bounds on the parameters. In concurrent work, [39] give a differentially private estimator for our unknown parameter setting which, for the same sample complexity as ours, has an error guarantee of $\|\hat{\mu} - \mu\|_2 \le \alpha\|\Sigma^{1/2}\|_2$. This result is strictly weaker than ours. However, their estimator, in contrast to ours, has the pleasant property of being computationally efficient.

Beyond (sub)Gaussian distributions, [4, 9, 42] study differentially private mean estimation under weaker moment assumptions.

In addition to the work we discuss in the introduction [41, 2], recent work focuses on practical private mean and covariance estimation in univariate [23] and multivariate [5] settings, although these approaches still require explicit private covariance estimation. Dwork et al. [31] gave a lower bound of $\Omega(d^{3/2})$ for privately estimating the empirical covariance, albeit via a non-Gaussian distribution. Together, these results serve as evidence that $\Omega(d^{3/2})$ samples are necessary for private covariance estimation for Gaussian data as well.

**Robust Statistical Estimation.** Robust statistical estimation [40, 49], which dates to at least 1960 [56] and remains an active area of research [18–20, 22, 38], studies the problem of estimating distribution parameters when an $\alpha$-fraction of the data may be adversarially corrupted. As noted by Dwork and Lei [25], robust statistics and differential privacy have similar goals, and private estimators are often inspired by robust estimators, but the models are formally incomparable.

More recent work aims to give algorithms which satisfy both constraints simultaneously [47, 35]. Specifically, independently from our work, Liu et al. [47] propose a simple mechanism for Gaussian mean estimation with known covariance, given $\alpha$-corrupted data sets and an a priori bound on the range of the mean $\|\mu\|_\infty \le \rho$. This estimator has sample complexity $\tilde{O}(d/\alpha^2 + d\log\rho/\alpha\varepsilon)$. The algorithm runs the exponential mechanism [50] in the given range, using the Tukey depth of a point as its score. We observe that the same mechanism gives a solution for the problem we study—mean

estimation in Mahalanobis distance with unknown covariance and no corruptions—but this solution requires a priori bounds on the mean and covariance, which are not required by our algorithms.

# 2 Preliminaries

We write $x \sim P$ to denote that $x$ is drawn from distribution $P$ and $x \sim P^{\otimes n}$ if $x$ consists of $n$ i.i.d. draws from $P$. We write $[n] = \{1, \ldots, n\}$. We define the Hamming distance between two data sets $x, y$ of size $n$ by $D_H(x, y) = |\{i \in [n] : x_i \neq y_i\}|$. For data set $x$ and set $S \subseteq \mathbb{R}^{n \times d}$, we write $D_H(x, S) = \min_{z \in S} D_H(x, z)$.

Let $x, x' \in \mathcal{X}^n$ be two data sets of size $n$. We say that $x, x'$ are *neighboring data sets* if $D_H(x, x') \leq 1$, and denote this by $x \sim x'$. Differentially private algorithms have *indistinguishable* output distributions on neighboring data sets.

**Definition 2.1** (($\varepsilon, \delta$)-indistinguishability)**.** *Two distributions $P, Q$ over domain $\mathcal{W}$ are $(\varepsilon, \delta)$-indistinguishable, denoted by $P \approx_{\varepsilon, \delta} Q$, if for any measurable subset $W \subseteq \mathcal{W}$,*

$$\Pr_{w \sim P}[w \in W] \leq e^\varepsilon \Pr_{w \sim Q}[w \in W] + \delta \quad and \quad \Pr_{w \sim Q}[w \in W] \leq e^\varepsilon \Pr_{w \sim P}[w \in W] + \delta.$$

**Definition 2.2** (Differential Privacy [29])**.** *A randomized algorithm $\mathcal{A} \colon \mathcal{X}^n \to \mathcal{W}$ is $(\varepsilon, \delta)$-differentially private if for all neighboring datasets $x, x'$ we have $\mathcal{A}(x) \approx_{\varepsilon, \delta} \mathcal{A}(x')$.*

We next describe well-known mechanisms which serve as building blocks for our algorithms.

**Lemma 2.3** (Laplace Mechanism [29])**.** *Let $f : \mathcal{X}^n \to \mathbb{R}$, data set $x \in \mathcal{X}^n$, and privacy parameter $\varepsilon$. The* Laplace Mechanism *returns $\tilde{f}(x) = f(x) + \mathrm{Lap}(\Delta_f/\varepsilon)$, where $\Delta_f = \max_{x \sim x'} |f(x) - f(x')|$ is the* global sensitivity *of $f$. The Laplace Mechanism is $(\varepsilon, 0)$-differentially private.*

**Lemma 2.4** (Gaussian Mechanism [29])**.** *Let $f : \mathcal{X}^n \to \mathbb{R}^d$, data set $x \in \mathcal{X}^n$, and privacy parameters $\varepsilon, \delta$. The* Gaussian Mechanism *returns $\tilde{f}(x) = f(x) + \mathcal{N}(0, \sigma^2 \mathbb{I})$, where $\sigma = \Delta_f \sqrt{2 \log(5/4\delta)}/\varepsilon$ and $\Delta_f = \max_{x \sim x'} \|f(x) - f(x')\|_2$ is the* global $\ell_2$-sensitivity *of $f$. The Gaussian Mechanism is $(\varepsilon, \delta)$-differentially private.*

# 3 Tukey Depth Mechanism

We start with a score function $q(x; y)$ and wish to sample from the exponential mechanism, proportional to $\exp(\varepsilon \cdot q(x; y)/2)$, but restricting the sampling to the set of points with score at least $t$. Denote this set $\mathcal{Y}_{t,x} = \{y \in \mathcal{Y} : q(x; y) \geq t\}$ and call the resulting distribution $\mathcal{M}_{\varepsilon, t}(x)$. Unfortunately, sampling directly from $\mathcal{M}_{\varepsilon, t}(x)$ may not be private. To address this, we try to sample only from data sets that are "safe" with respect to privacy, i.e., have distributions that are indistinguishable from those of their neighbors.

**Definition 3.1** (Safety)**.** *Data set $x$ is $(\varepsilon, \delta, t)$-safe if, for all $x' \sim x$, we have $\mathcal{M}_{\varepsilon, t}(x) \approx_{\varepsilon, \delta} \mathcal{M}_{\varepsilon, t}(x')$. Let $\mathtt{SAFE}_{(\varepsilon, \delta, t)} \subseteq \mathcal{X}^n$ be the set of safe data sets, and let $\mathtt{UNSAFE}_{(\varepsilon, \delta, t)} = \mathcal{X}^n \setminus \mathtt{SAFE}_{(\varepsilon, \delta, t)}$ be its complement.*

Following the propose-test-release framework of [25], we check if the input is *far* from $\mathtt{UNSAFE}_{(\varepsilon, \delta, t)}$. Since the distance check itself is private and indistinguishability *is* the definition of safety, the proof of privacy becomes straightforward. Such an abstract definition, however, does not yield much insight into what safe data sets look like. Below, we show that one can establish safety via a simple condition on the volumes of sets of the form $\mathcal{Y}_{t \pm \eta, x}$ (for certain values of $\eta$), which allows us to show that Gaussian data are far from $\mathtt{UNSAFE}_{(\varepsilon, \delta, t)}$ with high probability.

---

**Algorithm 1** Restricted Exponential Mechanism $\mathcal{A}^E_{\varepsilon, \delta, t}(x)$

---

**Require:** Data space $\mathcal{X}$. Output space $\mathcal{Y} \cup \{\mathtt{FAIL}\}$. Data set $x \in \mathcal{X}^n$. Score function $q : \mathcal{X}^n \times \mathcal{Y} \to \mathbb{R}$, with global sensitivity 1 in the first argument. Privacy parameters $\varepsilon, \delta > 0$. Minimum score $t$.
1: $h \leftarrow D_H(x, \mathtt{UNSAFE}_{(\varepsilon, \delta, t)})$
2: **if** $h + z < \frac{\log(1/2\delta)}{\varepsilon}$ for $z \sim \mathrm{Lap}(1/\varepsilon)$ **then return** FAIL.
3: **return** $\hat{y} \sim \mathcal{M}_{\varepsilon, t}(x)$, where $\mathcal{M}_{\varepsilon, t}(x) \propto \begin{cases} \exp\left\{\frac{\varepsilon q(x; y)}{2}\right\} & \text{if } y \in \mathcal{Y}_{t,x} \\ 0 & \text{otherwise} \end{cases}$

---

## 3.1 Main Algorithm

We estimate the mean by instantiating Algorithm 1 with $t = n/4$ and score function $q(x; y) = nT_x(y)$, where $T_x(y)$ is the (empirical) Tukey depth, defined in Equation (4). Observe that $nT_x(y)$ has sensitivity 1, since for any halfspace the fraction of points it contains can change by at most $\frac{1}{n}$ when we change one data point.

**Theorem 3.2** (Privacy and Accuracy of the Tukey-Depth Mechanism). *For any $\varepsilon, \delta > 0$, Algorithm 1 is $(2\varepsilon, e^\varepsilon \delta)$-differentially private. There exists an absolute constant $C$ such that, for any $0 < \alpha, \beta, \varepsilon < 1$, $0 < \delta \leq \frac{1}{2}$, mean $\mu$, and positive definite $\Sigma$, if $x \sim \mathcal{N}(\mu, \Sigma)^{\otimes n}$ and*

$$ n \geq C \left( \frac{d + \log(1/\beta)}{\alpha^2} + \frac{d \log(1/\alpha) + \log(1/\beta)}{\alpha\varepsilon} + \frac{\log(1/\delta)}{\varepsilon} \right), \tag{5} $$

*then with probability at least $1 - 3\beta$, Algorithm 1 with $q(x; y) = nT_x(y)$ returns $\mathcal{A}^E_{\varepsilon, \delta, n/4}(x) = \hat{\mu}$ such that $\|\hat{\mu} - \mu\|_\Sigma \leq \alpha$.*

We focus here on the accuracy analysis. Privacy follows from standard propose-test-release calculations [25], which we include in the supplementary material.

## 3.2 Accuracy Analysis

The proof of accuracy proceeds in four stages. Using standard analysis, we first relate the empirical Tukey depth of a point $y$ to its Mahalanobis distance $\|y - \mu\|_\Sigma$ via the *expectation* of Tukey depth under $\mathcal{N}(\mu, \Sigma)$. Since Tukey depth is defined as a minimum over halfspaces, which have Vapnik-Chervonekis dimension $d + 1$, one can show via uniform convergence that the empirical measure concentrates around its expectation. This portion ends with a standard lemma relating the sets $\mathcal{Y}_{np,x}$, where $p \in (0, 1/2)$, to ellipsoids defined by Mahalanobis distance.

The remaining three steps, which are new to this work, begin with a characterization of the set SAFE defined above, which provides conditions under which a data set is far from UNSAFE. We discuss this in more detail below. The third stage uses that characterization and the tools we developed to show that Gaussian data is typically far from UNSAFE, establishing that Algorithm 1 has a small probability of returning FAIL. Finally, conditioned on Algorithm 1 not returning FAIL, a similar analysis shows that with high probability the restricted sampler $\mathcal{M}_{\varepsilon,t}(x)$ returns a point with high empirical Tukey depth. Together, this yields a bound on the Mahalanobis distance to the true mean.

**A Volume Condition for Safety** We consider sets of the form $\mathcal{Y}_{t+\eta,x}$ for moderate positive and negative values of $\eta$. Recall that $\mathcal{Y}_{t+\eta,x}$ is the set of all points $y$ with score $q(x; y) = nT_x(y) \geq t + \eta$, i.e., having empirical Tukey depth with respect to $x$ at least $(t + \eta)/n$. Therefore, as $\eta$ becomes smaller, the set $\mathcal{Y}_{t+\eta,x}$ grows. We show that, if the volume of $\mathcal{Y}_{t+\eta,x}$ does not increase too quickly as $\eta$ decreases, then $x$ is far from every data set in UNSAFE$_{(\varepsilon,\delta,t)}$. In particular, this implies that $x$ itself is in SAFE$_{(\varepsilon,\delta,t)}$. These lemmas do not rely on specific features of Gaussian data or Tukey depth, which enter in only in the last two stages as described above, when we argue about typical data sets.

Before arguing about volumes directly, we sketch the proof of a lemma about the weight assigned to sets by the exponential mechanism. For any set $S \subseteq \mathcal{Y}$, denote its weight by $w_x(S) = \int_S \exp\left\{ \frac{\varepsilon q(x;y)}{2} \right\} dy$.

**Lemma 3.3.** *Assume $\delta < \frac{1}{2}$. If $w_x(\mathcal{Y}_{t+1,x}) \geq (1 - \delta) w_x(\mathcal{Y}_{t-1,x})$, then $x \in$ SAFE$_{(\varepsilon,\delta',t)}$ for $\delta' = 4e^\varepsilon \delta$.*

*Proof Sketch.* For an arbitrary adjacent $x'$, we must establish that the distributions $\mathcal{M}_{\varepsilon,t}(x)$ and $\mathcal{M}_{\varepsilon,t}(x')$ are $(\varepsilon, \delta')$-indistinguishable. The proof is elementary but non-trivial: our hypothesis only concerns $x$, not its neighbors. To illustrate, we show that $\Pr[\mathcal{M}_{\varepsilon,t}(x) \notin \text{supp}(\mathcal{M}_{\varepsilon,t}(x'))] \leq \delta'$. By definition, $\text{supp}(\mathcal{M}_{\varepsilon,t}(x')) = \mathcal{Y}_{t,x'}$. Now we move from $x'$ to $x$: the score function has sensitivity 1, so every point with $q(x; y) \geq t + 1$ satisfies $q(x'; y) \geq t$. Thus $\mathcal{Y}_{t+1,x} \subseteq \mathcal{Y}_{t,x'}$, and we have

$$ \Pr[\mathcal{M}_{\varepsilon,t}(x) \notin \text{supp}(\mathcal{M}_{\varepsilon,t}(x'))] \leq \Pr[\mathcal{M}_{\varepsilon,t}(x) \notin \mathcal{Y}_{t+1,x}] = 1 - \frac{w_x(\mathcal{Y}_{t+1,x})}{w_x(\mathcal{Y}_{t,x})}. $$

Since $\mathcal{Y}_{t,x} \subseteq \mathcal{Y}_{t-1,x}$, $w_x(\mathcal{Y}_{t,x}) \leq w_x(\mathcal{Y}_{t-1,x})$, and our hypothesis implies $\Pr[\mathcal{M}_{\varepsilon,t}(x) \notin \text{supp}(\mathcal{M}_{\varepsilon,t}(x'))] \leq \delta$, which is less than $\delta' = 4e^\varepsilon \delta$.

The rest of the proof requires similar (but slightly longer) calculations. □

We now use this lemma to establish when a data set is far from the set of unsafe data sets. Note that setting $k = 0$ below implies for all $z \in \text{UNSAFE}$ that we have $D_H(x, z) > 0$, i.e., $x \in \text{SAFE}$.

**Lemma 3.4.** *For any $k \geq 0$, if there exists a $g > 0$ such that $\frac{\text{Vol}(\mathcal{Y}_{t-k-1,x})}{\text{Vol}(\mathcal{Y}_{t+k+g+1,x})} \cdot e^{-\varepsilon g/2} \leq \delta$, then for all $z \in \text{UNSAFE}_{(\varepsilon,\delta',t)}$, with $\delta' = 4e^{\varepsilon}\delta$, we have $D_H(x, z) > k$*

*Proof.* Take some $z$ at distance at most $k$ from $x$ (if $k = 0$, set $z \leftarrow x$). We show $z \in \text{SAFE}_{(\varepsilon,\delta',t)}$. We have, from Lemma 3.3, that if $\frac{w_z(\mathcal{Y}_{t+1,z})}{w_z(\mathcal{Y}_{t-1,z})} \geq 1 - \delta$, then $z$ is safe. This assumption is equivalent to $\frac{w_z(\mathcal{Y}_{t-1,z} \setminus \mathcal{Y}_{t+1,z})}{w_z(\mathcal{Y}_{t-1,z})} \leq \delta$, which is the form we use.

First we lower bound the denominator:

$$w_z(\mathcal{Y}_{t-1,z}) \geq w_z(\mathcal{Y}_{t+g+1,z}) \geq \text{Vol}(\mathcal{Y}_{t+g+1,z})e^{\varepsilon(t+g+1)/2} \geq \text{Vol}(\mathcal{Y}_{t+g+k+1,x})e^{\varepsilon(t+g+1)/2},$$

where (crucially) the last inequality switches to the volume under $x$, and we have used the sensitivity of $q$. We use the same idea on the numerator, switching to a volume under $x$ in the first inequality:

$$w_z(\mathcal{Y}_{t-1,z} \setminus \mathcal{Y}_{t+1,z}) \leq w_z(\mathcal{Y}_{t-k-1,x} \setminus \mathcal{Y}_{t+1,z}) \leq \text{Vol}(\mathcal{Y}_{t-k-1,x})e^{\varepsilon(t+1)/2}.$$

With an upper bound on the numerator, a lower bound on the denominator, and the fact that $\frac{e^{\varepsilon(t+1)/2}}{e^{\varepsilon(t+g+1)/2}} = e^{-\varepsilon g/2}$, we have

$$\frac{w_z(\mathcal{Y}_{t-1,z} \setminus \mathcal{Y}_{t+1,z})}{w_z(\mathcal{Y}_{t-1,z})} \leq \frac{\text{Vol}(\mathcal{Y}_{t-k-1,x})}{\text{Vol}(\mathcal{Y}_{t+k+g+1,x})} \cdot e^{-\varepsilon g/2} \leq \delta,$$

so $z \in \text{SAFE}_{(\varepsilon,\delta',t)}$. □

With these tools in hand, in the supplementary material we show that, if $n = \omega(d/\varepsilon)$, typical Gaussian data satisfy the hypothesis of Lemma 3.4 for $g = n/8$ and thus are far from unsafe (implying that we are unlikely to return FAIL) and that with high probability the restricted sampler $\mathcal{M}_{\varepsilon,t}(x)$ returns a point with high Tukey depth.

## 4 Empirically Rescaled Gaussian Mechanism

In this section, we describe our second estimator. At a high level, the estimator first privately checks whether the data set $x$ is $\frac{1}{\varepsilon}$-close in Hamming distance to a good set of "roughly Gaussian" data sets $\mathcal{G}(\lambda)$. If so, it finds the closest data set to $x$ that belongs in $\mathcal{G}(\lambda)$, which we call $\tilde{x}$. Then it returns a sample $\hat{\mu}$ drawn from $\mathcal{N}(\mu_{\tilde{x}}, C^2\Sigma_{\tilde{x}})$, where $\mu_{\tilde{x}}$ and $\Sigma_{\tilde{x}}$ are the empirical mean and covariance of $\tilde{x}$ and $C$ is a scale parameter appropriately set to ensure privacy. Specifically, we use the definitions:

**Definition 4.1** (Empirical Mean and Covariance). *For data set $x \in \mathbb{R}^{3n \times d}$, the empirical mean and covariance of $x$ are respectively defined by $\mu_x = \frac{1}{n}\sum_{i=1}^{n} x_{i+2n}$ and $\Sigma_x = \frac{1}{2n}\sum_{i=1}^{n}(x_i - x_{i+n})(x_i - x_{i+n})^T$.*

In comparison with the standard empirical estimators, ours enable a simpler privacy analysis, since replacing one datapoint in $x$ only affects one term in one of the sums. For convenience, we choose the number of samples to be $3n$ so that we can pair the first two thirds to construct $\Sigma_x$ and use the last third for $\mu_x$. With these definitions at hand, the good set $\mathcal{G}(\lambda)$ is defined as follows:

**Definition 4.2** ($\lambda$-goodness). *For any $\lambda > 0$, define $\mathcal{G}(\lambda) \subseteq \mathbb{R}^{3n \times d}$ as*

$$\mathcal{G}(\lambda) \stackrel{def}{=} \left\{ x \in \mathbb{R}^{3n \times d} : \Sigma_x \text{ is invertible and } \forall i \in [3n] \ \|x_i - \mu_x\|_{\Sigma_x}^2 \leq \lambda \right\}.$$

We set $\lambda \approx d \log n$, since for this value (sub)Gaussian data will belong in $\mathcal{G}(\lambda)$ with high probability.

Finally, we note that the algorithm immediately aborts if the number of samples is less than $\frac{k\lambda \log(1/\delta)}{\varepsilon} \approx \frac{d \log(1/\delta)}{\varepsilon^2}$, a condition necessary to ensure privacy.

---
**Algorithm 2** Empirically Rescaled Gaussian Mechanism $\mathcal{A}^G_{\varepsilon,\delta,\beta}(x)$

---
**Require:** Data set $x = (x_1, \ldots, x_{3n})^T \in \mathbb{R}^{3n \times d}$. Privacy parameters $\varepsilon, \delta > 0$. Failure probability $\beta > 0$.

1: Initialize: $\lambda \leftarrow O\left(d \log \frac{n}{\beta}\right)$, $t \leftarrow \frac{1}{\varepsilon} \log \frac{1}{\beta}$, $k \leftarrow \frac{2}{\varepsilon} \log \frac{1}{\delta\beta} + 1$, $C^2 \leftarrow \frac{32k^2}{\varepsilon^2 n^2} \cdot \frac{\lambda}{1-2k\lambda/n} \cdot \log \frac{1.25}{\delta}$.

2: **if** $n = o\left(\frac{k\lambda}{\varepsilon} \log \frac{1}{\delta}\right)$ **then return** FAIL.

3: $\bar{x} \leftarrow \sigma(x)$                                      $\triangleright$ random permutation $\sigma : (\mathbb{R}^d)^{3n} \to (\mathbb{R}^d)^{3n}$

4: $h \leftarrow D_H(\bar{x}, \mathcal{G}(\lambda))$                                 $\triangleright$ distance between $\bar{x}$ and $\lambda$-goodness

5: **if** $h + r > t$ for $r \sim \mathrm{Lap}(1/\varepsilon)$ **then return** FAIL.

6: $\tilde{x} \leftarrow \arg\min_{z \in \mathcal{G}(\lambda)} D_H(\bar{x}, z)$                          $\triangleright$ projection to $\lambda$-goodness

7: **return** $\hat{\mu} \sim \mathcal{N}(\mu_{\tilde{x}}, C^2 \Sigma_{\tilde{x}})$

---

We remark that the sample size check in line 2 and the setting of $\lambda$ are not well-defined, as they are stated with asymptotic notation. Although it is possible to compute the constants for these steps, we exclude them in favor of a cleaner analysis.

We prove accuracy for the following family of distributions:

**Definition 4.3** (Subgaussian distribution). *Let $P_{\mu,\Sigma}$ be a distribution over $\mathbb{R}^d$ with mean $\mu$ and covariance $\Sigma \succ 0$. For a constant $c > 0$, we say that $P_{\mu,\Sigma}$ is subgaussian with parameter $c\Sigma$, if for all $u \in \mathbb{R}^d$ such that $\|u\|_2 = 1$, $\mathbb{E}_{v \sim P_{\mu,\Sigma}}[e^{\lambda u^T(v-\mu)}] \leq e^{c\lambda^2(u^T \Sigma u)/2}$ for all $\lambda \in \mathbb{R}$.*

We write $P_{\mu,\Sigma} \in \mathrm{subG}(c\Sigma)$. The definition appears in relevant literature [48, 22] and intuitively asks that the distribution concentrates "at least as well" as a Gaussian along every univariate projection.

We now state the guarantees of Algorithm 2. We defer their proof to the supplementary material.

**Theorem 4.4** (Privacy and Accuracy of the Empirically Rescaled Gaussian Mechanism). *For any $\varepsilon > 0$, $0 < \delta < 1$, Algorithm 2 is $(3\varepsilon, e^\varepsilon(1 + e^\varepsilon)\delta)$-differentially private. There exists an absolute constant $C$ such that, for any $0 < \alpha, \beta, \varepsilon, \delta < 1$, mean $\mu$, and positive definite $\Sigma$, if $x \sim P_{\mu,\Sigma}^{\otimes n}$, where $P_{\mu,\Sigma} \in \mathrm{subG}(c\Sigma)$ for some constant $c > 0$, and*

$$n \geq C\left(\frac{d}{\alpha^2} \log \frac{1}{\beta} + \frac{d}{\alpha\varepsilon^2} \log^3 \frac{1}{\delta\beta} \cdot \log \frac{d \log(1/\delta\beta)}{\alpha\varepsilon}\right), \tag{6}$$

*then with probability at least $1 - 3\beta$, Algorithm 2 returns $\mathcal{A}^G_{\varepsilon,\delta,\beta}(x) = \hat{\mu}$ such that $\|\hat{\mu} - \mu\|_\Sigma \leq \alpha$.*

## Acknowledgments and Disclosure of Funding

We thank the anonymous NeurIPS reviewers for useful suggestions on the presentation of this manuscript. Gavin Brown and Adam Smith were supported in part by NSF award CCF-1763786 as well as a Sloan Foundation research award. Marco Gaboardi was supported in part by NSF award CCF-2040222, CCF-1718220, CNS-1565365, and CNS-2040215. Jonathan Ullman and Lydia Zakynthinou were supported by NSF grants CCF-1750640, CNS-1816028, and CNS-1916020. Lydia Zakynthinou was also supported by a Facebook Fellowship.

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
