the univariate and multivariate setting. [41, 49] study private mean estimation in the Gaussian case under the more strict constraint of local differential privacy.

When the covariance is unknown, a natural approach would be to estimate it and use one of the mean estimators above. In addition to the work we discuss in the introduction [50, 2], recent work focuses on practical private mean and covariance estimation in univariate [28] and multivariate [5] settings, although these approaches still require explicit private covariance estimation.

**Robust Statistical Estimation.** Robust statistical estimation [48, 60], which dates to at least 1960 [71] and remains an active area of research [22, 23, 24, 26, 45], studies the problem of estimating distribution parameters

when an $\alpha$-fraction of the data may be adversarially corrupted. As noted by Dwork and Lei [30], robust statistics and differential privacy have similar goals, and private estimators are often inspired by robust estimators, but the models are formally incomparable.

More recent work aims to give algorithms which satisfy both constraints simultaneously [58, 42]. Specifically, independently from our work, Liu et al. [58] propose a simple mechanism for Gaussian mean estimation with known covariance, given $\alpha$-corrupted data sets and an a priori bound on the range of the mean $\|\mu\|_\infty \le \rho$. This estimator has sample complexity $\tilde{O}(d/\alpha^2 + d \log \rho / \alpha\varepsilon)$. The algorithm runs the exponential mechanism [61] in the given range, using the Tukey depth of a point as its score. We observe that the same mechanism gives a solution for the problem we study—mean estimation in Mahalanobis distance with unknown covariance and no corruptions—but this solution requires a priori bounds on the mean and covariance, which are not required by our algorithms.

**Lower Bounds.** Starting from the univariate case, [53] prove that $\Omega(\log(1/\delta)/\varepsilon\alpha)$ samples are necessary for Gaussian estimation even when the variance is known. For multivariate mean estimation, the investigation of sample complexity lower bounds has been driven by *membership-inference attacks* (sometimes called *tracing* or *fingerprinting*) [10, 68, 12, 69, 38]. Using this technique, Kamath et al. [50] show a lower bound of $\Omega(d/\alpha\varepsilon)$ for Gaussian estimation with identity covariance, which clearly extends to our problem as well. The sample complexity of our estimators matches this lower bound up to logarithmic factors and has no dependence on a priori bounds on the parameters.

Moreover, we conjecture that any optimal differentially private Gaussian mean estimator with unknown covariance would have to go beyond current techniques, which compute a private estimate of the covariance matrix as an intermediate step. Recall that the best known sample complexity bound for privately computing a matrix $A$ such that $\mathbb{I} \preceq A\Sigma A \preceq 2\mathbb{I}$ for (sub)Gaussian distributions is $\tilde{O}(d^{3/2} \log^{1/2} \kappa/\varepsilon)$ [50]. In addition, Dwork et al. [37] gave a lower bound of $\Omega(d^{3/2})$ for estimating the empirical covariance matrix, but only when the data is sampled from a worst-case distribution. Together, these results serve as evidence that $\Omega(d^{3/2})$ samples are necessary for private covariance estimation even for the case of Gaussian data.

## 1.3 Organization

We first provide background on differential privacy (Section 2) and refer the reader to Appendix A for a review of the relevant linear algebra. In Section 3, we present our Tukey-Depth Mechanism and prove its privacy and sample complexity guarantees (c.f. Theorem 1.1). In Section 4, we present our Empirically Rescaled Gaussian Mechanism and prove its guarantees for Gaussian distributions; the extension to subgaussian data (c.f. Theorem 1.2) is in Appendix B. We present finite (yet computationally inefficient) implementations of both algorithms in Appendix C. Appendix D contains additional proofs which did not appear in the body of the paper.

## 2 Preliminaries

We write $x \sim P$ to denote that $x$ is drawn from distribution $P$ and $x \sim P^{\otimes n}$ if $x$ consists of $n$ i.i.d. draws from $P$. In particular, we consider data sets $x = (x_1, \ldots, x_n) \in \mathbb{R}^{n \times d}$ which consist of $n$ i.i.d. samples, each drawn from a $d$-dimensional distribution $P$ with mean $\mu \in \mathbb{R}^d$ and covariance $\Sigma \in \mathbb{R}^{d \times d}$. We write $[n] = \{1, \ldots, n\}$. We define the Hamming distance between two data sets $x, y$ of size $n$ by $D_H(x, y) = |\{i \in [n] : x_i \ne y_i\}|$. For data set $x$ and set $S \subseteq \mathbb{R}^{n \times d}$, we write $D_H(x, S) = \min_{z \in S} D_H(x, z)$. We denote the natural logarithm by $\log$.

Let $x, x' \in \mathcal{X}^n$ be two data sets of size $n$. We say that $x, x'$ are *neighboring data sets* if $D_H(x, x') \le 1$, and denote this by $x \sim x'$. Differentially private algorithms have *indistinguishable* output distributions on neighboring data sets.

**Definition 2.1** (($\varepsilon, \delta$)-indistinguishability). Two distributions $P, Q$ over domain $\mathcal{W}$ are ($\varepsilon, \delta$)-indistinguishable, denoted by $P \approx_{\varepsilon,\delta} Q$, if for any measurable subset $W \subseteq \mathcal{W}$,

$$\Pr_{w \sim P}[w \in W] \le e^\varepsilon \Pr_{w \sim Q}[w \in W] + \delta \quad \text{and} \quad \Pr_{w \sim Q}[w \in W] \le e^\varepsilon \Pr_{w \sim P}[w \in W] + \delta.$$

**Definition 2.2** (Differential Privacy [34]). *A randomized algorithm $\mathcal{A}\colon \mathcal{X}^n \to \mathcal{W}$ is $(\varepsilon, \delta)$-differentially private if for all neighboring datasets $x, x'$ we have $\mathcal{A}(x) \approx_{\varepsilon,\delta} \mathcal{A}(x')$.*

A crucial property of differential privacy is that it composes adaptively. We say that $M$ is an *adaptive composition* of $M_1, \ldots, M_T$ if it consists of a sequence of mechanisms $M_1(x), M_2(x), \ldots, M_T(x)$, executed on data set $x$, where each mechanism $M_t(x)$ depends on the outputs of $M_1(x), \ldots, M_{t-1}(x)$.

**Lemma 2.3** (Composition [34]). *If $M_1, \ldots, M_T$ are $(\varepsilon_1, \delta_1), \ldots, (\varepsilon_T, \delta_T)$-differentially private respectively and $M$ is their adaptive composition, then $M$ is $(\varepsilon, \delta)$-differentially private for $\varepsilon = \sum_{t=1}^{T} \varepsilon_t$ and $\delta = \sum_{t=1}^{T} \delta_t$.*

We next describe well-known mechanisms which serve as building blocks for our algorithms.

**Definition 2.4** (Laplace Mechanism [34]). *Let $f : \mathcal{X}^n \to \mathbb{R}$, data set $x \in \mathcal{X}^n$, and privacy parameter $\varepsilon$. The Laplace Mechanism returns*

$$\tilde{f}(x) = f(x) + \mathrm{Lap}\left(\frac{\Delta_f}{\varepsilon}\right),$$

where $\Delta_f = \max_{x \sim x'} |f(x) - f(x')|$ is the *global sensitivity* of $f$.

**Lemma 2.5** ([34]). *The Laplace Mechanism is $(\varepsilon, 0)$-differentially private.*

**Definition 2.6** (Gaussian Mechanism, [34]). *Let $f : \mathcal{X}^n \to \mathbb{R}^d$, data set $x \in \mathcal{X}^n$, and privacy parameters $\varepsilon, \delta$. The Gaussian Mechanism returns*

$$\tilde{f}(x) = f(x) + \mathcal{N}(0, \sigma^2 \mathbb{I}), \text{ where } \sigma = \Delta_f \sqrt{2 \log(1.25/\delta)}/\varepsilon$$

and $\Delta_f = \max_{x \sim x'} \|f(x) - f(x')\|_2$ is the *global $\ell_2$-sensitivity* of $f$.

**Lemma 2.7** ([34]). *The Gaussian Mechanism is $(\varepsilon, \delta)$-differentially private.*

# 3 Tukey Depth Mechanism

We start with a score function $q(x; y)$ and wish to sample from the exponential mechanism, proportional to $\exp(\varepsilon \cdot q(x; y)/2)$, but restricting the sampling to the set of points with score at least $t$. Denote this set $\mathcal{Y}_{t,x} = \{y \in \mathcal{Y} : q(x; y) \geq t\}$ and call the resulting distribution $\mathcal{M}_{\varepsilon,t}(x)$. Unfortunately, sampling directly from $\mathcal{M}_{\varepsilon,t}(x)$ may not be private. To address this, we try to sample only from data sets that are "safe" with respect to privacy, i.e., have distributions that are indistinguishable from those of their neighbors.

**Definition 3.1** (Safety). *Data set $x$ is $(\varepsilon, \delta, t)$-safe if, for all $x' \sim x$, we have $\mathcal{M}_{\varepsilon,t}(x) \approx_{\varepsilon,\delta} \mathcal{M}_{\varepsilon,t}(x')$. Let $\mathrm{SAFE}_{(\varepsilon,\delta,t)} \subseteq \mathcal{X}^n$ be the set of safe data sets, and let $\mathrm{UNSAFE}_{(\varepsilon,\delta,t)} = \mathcal{X}^n \setminus \mathrm{SAFE}_{(\varepsilon,\delta,t)}$ be its complement.*

Following the propose-test-release framework of [30], we check if the input is *far* from $\mathrm{UNSAFE}_{(\varepsilon,\delta,t)}$. Since the distance check itself is private and indistinguishability *is* the definition of safety, the proof of privacy becomes straightforward. Such an abstract definition, however, does not yield much insight into what safe data sets look like. Below, we show that one can establish safety via a simple condition on the volumes of sets of the form $\mathcal{Y}_{t\pm\eta,x}$ (for certain values of $\eta$), which allows us to show that Gaussian data are far from $\mathrm{UNSAFE}_{(\varepsilon,\delta,t)}$ with high probability.

---

**Algorithm 1** Restricted Exponential Mechanism $\mathcal{A}^E_{\varepsilon,\delta,t}(x)$

---

**Require:** Data space $\mathcal{X}$. Output space $\mathcal{Y} \cup \{\mathrm{FAIL}\}$. Data set $x \in \mathcal{X}^n$. Score function $q : \mathcal{X}^n \times \mathcal{Y} \to \mathbb{R}$, with global sensitivity 1 in the first argument. Privacy parameters $\varepsilon, \delta > 0$. Minimum score $t$.

1: $h \leftarrow D_H(x, \mathrm{UNSAFE}_{(\varepsilon,\delta,t)})$

2: **if** $h + z < \frac{\log(1/2\delta)}{\varepsilon}$ for $z \sim \mathrm{Lap}(1/\varepsilon)$ **then return** FAIL.

3: **return** $\hat{y} \sim \mathcal{M}_{\varepsilon,t}(x)$, where $\mathcal{M}_{\varepsilon,t}(x) \propto \begin{cases} \exp\left\{\frac{\varepsilon q(x;y)}{2}\right\} & \text{if } y \in \mathcal{Y}_{t,x} \\ 0 & \text{otherwise} \end{cases}$

---

## 3.1 Main Algorithm

We estimate the mean by instantiating Algorithm 1 with $t = n/4$ and score function $q(x; y) = nT_x(y)$, where $T_x(y)$ is the (empirical) Tukey depth, defined as

$$T_x(y) \overset{\text{def}}{=} \min_{v \in \mathbb{R}^d} \frac{1}{n} \left| \left\{ x_i \in x : \langle x_i, v \rangle \geq \langle y, v \rangle \right\} \right|. \tag{6}$$

Observe that $nT_x(y)$ has sensitivity 1, since for any halfspace the fraction of points it contains can change by at most $\frac{1}{n}$ when we change one data point.

**Theorem 3.2** (Privacy and Accuracy of the Tukey-Depth Mechanism). *For any $\varepsilon, \delta > 0$, Algorithm 1 is $(2\varepsilon, e^\varepsilon \delta)$-differentially private. There exists an absolute constant $C$ such that, for any $0 < \alpha, \beta, \varepsilon < 1$, $0 < \delta \leq \frac{1}{2}$, mean $\mu$, and positive definite $\Sigma$, if $x \sim \mathcal{N}(\mu, \Sigma)^{\otimes n}$ and*

$$n \geq C \left( \frac{d + \log(1/\beta)}{\alpha^2} + \frac{d \log(1/\alpha) + \log(1/\beta)}{\alpha \varepsilon} + \frac{\log(1/\delta)}{\varepsilon} \right), \tag{7}$$

*then with probability at least $1 - 3\beta$, Algorithm 1 with $q(x; y) = nT_x(y)$ returns $\mathcal{A}^E_{\varepsilon, \delta, n/4}(x) = \hat{\mu}$ such that $\|\hat{\mu} - \mu\|_\Sigma \leq \alpha$.*

In particular, setting $\delta = \frac{1}{n^2}$ and $\beta = \frac{1}{n}$, it suffices to take a sample of size

$$n = \tilde{O} \left( \frac{d}{\alpha^2} + \frac{d}{\alpha \varepsilon} \right).$$

## 3.2 Accuracy Analysis

The proof of accuracy proceeds in four stages. Using standard analysis, we first relate the empirical Tukey depth of a point $y$ to its Mahalanobis distance $\|y - \mu\|_\Sigma$ via the *expectation* of Tukey depth under $\mathcal{N}(\mu, \Sigma)$. Since Tukey depth is defined as a minimum over halfspaces, which have Vapnik-Chervonekis dimension $d + 1$, one can show via uniform convergence that the empirical measure concentrates around its expectation, with some error that we denote $\alpha_1$. This portion ends with a standard lemma relating the sets $\mathcal{Y}_{np,x}$, where $p \in (0, 1/2)$, to ellipsoids defined by Mahalanobis distance.

The remaining three steps, which are new to this work, begin with a characterization of the set SAFE defined above, which provides conditions under which a data set is far from UNSAFE. The third stage uses that characterization and the tools we developed to show that Gaussian data is typically far from UNSAFE, establishing that Algorithm 1 has a small probability of returning FAIL. Finally, conditioned on Algorithm 1 not returning FAIL, a similar analysis shows that with high probability the restricted sampler $\mathcal{M}_{\varepsilon, t}(x)$ returns a point with empirical Tukey depth at most $\alpha_2$ far from optimal. Combined with the error $\alpha_1$ from above, this yields a bound on the Mahalanobis distance to the true mean.

### 3.2.1 Relating Tukey Depth to Mahalanobis Distance

The first steps in our analysis imply that, privacy considerations aside, points with high Tukey depth are good estimators for the Gaussian mean. These arguments are standard; for a recent application to differentially private estimation see the concurrent work of Liu et al. [58].

The *expected Tukey depth*, $T_{\mathcal{N}(\mu, \Sigma)}$, is a population version of the empirical fraction defined above. For brevity, we define $P = \mathcal{N}(\mu, \Sigma)$ and write

$$T_{\mathcal{N}(\mu, \Sigma)}(y) = T_P(y) \overset{\text{def}}{=} \min_v \Pr_{X \sim P}[\langle X, v \rangle \geq \langle y, v \rangle]. \tag{8}$$

The expected Tukey depth is cleanly characterized in terms of Mahalanobis distance and $\Phi$, the CDF of the standard univariate Gaussian.[4] We restate and prove the following standard claim as Proposition D.2 in Appendix D.

---

[4] We also use $\Phi^{-1}$, the *quantile function*. Both $\Phi$ and $\Phi^{-1}$ are continuous and strictly increasing, and $\Phi^{-1}$ satisfies $-\Phi^{-1}(x) = \Phi^{-1}(1-x)$.

**Proposition 3.3.** *For any $\mu, y \in \mathbb{R}^d$ and positive definite $\Sigma$, $T_{\mathcal{N}(\mu,\Sigma)}(y) = T_P(y) = \Phi(-\|y - \mu\|_\Sigma)$.*

To move between Mahalanobis distance and empirical Tukey depth, we require that the latter is close to its population analog. We call data sets where this holds "typical."

**Definition 3.4** (Typicality). Data set $x$ is $\alpha_1$-*typical* for $\alpha_1 > 0$ if, for all $y \in \mathbb{R}^d$, $|T_x(y) - T_P(y)| \leq \alpha_1$.

We now point out that the typical data set is, in fact, $\alpha_1$-typical. We use the fact that, since the set of halfspaces has VC dimension $d + 1$, we have uniform convergence between the empirical and expected fractions of data points in the halfspace [73]. Then (as discussed in [27]) one need only observe that this result carries over to Tukey depth, since it is defined in terms of halfspaces. See [14] for discussion of these and other convergence results. Our exact statment comes from the recent [58], which analyzes the exponential mechanism on Tukey depth for the purpose of robust and private mean estimation.

**Lemma 3.5** (Convergence of Tukey Depth, [73, 27, 58]). *There exists a constant $c$ such that for any $\alpha_1, \beta > 0$ if $n \geq c\left(\frac{d + \log(1/\beta)}{\alpha_1^2}\right)$, then $x \sim \mathcal{N}(\mu, \Sigma)^{\otimes n}$ is $\alpha_1$-typical with probability at least $1 - \beta$.*

We will often manipulate subsets of points that have scores above a certain value, so let

$$\mathcal{Y}_{t,x} \stackrel{\text{def}}{=} \{y \in \mathcal{Y} : q(x; y) \geq t\}.$$

Note that, by construction, $\mathcal{Y}_{t,x} = \text{supp}(\mathcal{M}_{\varepsilon,t}(x))$. We will need to control the ratio of volumes of these spaces, and have the following useful lemma for $\alpha_1$-typical data sets.

**Lemma 3.6** (Volume Ratio). *Let $p, q \in (0, 1/2)$. If $x$ is $\alpha_1$-typical, then*

$$\frac{\text{Vol}(\mathcal{Y}_{np,x})}{\text{Vol}(\mathcal{Y}_{nq,x})} \leq \left(\frac{\Phi^{-1}(1 - p + \alpha_1)}{\Phi^{-1}(1 - q - \alpha_1)}\right)^d. \tag{9}$$

*Proof.* Let $\mathcal{B}_r$ denote the set of points $y$ such that $\|y - \mu\|_\Sigma \leq r$. By definition, $y \in \mathcal{Y}_{np,x} \Rightarrow T_x(y) \geq p$. Thus, by typicality and Proposition 3.3, $\Phi(-\|y - \mu\|_\Sigma) = T_P(y) \geq p - \alpha_1$. Taking inverses, we have

$$\|y - \mu\|_\Sigma \leq -\Phi^{-1}(p - \alpha_1) = \Phi^{-1}(1 - p + \alpha_1). \tag{10}$$

So $\mathcal{Y}_{np,x} \subseteq \mathcal{B}_{\Phi^{-1}(1-p+\alpha_1)}$. Similarly, since $\|y - \mu\|_\Sigma \leq \Phi^{-1}(1 - q - \alpha_1)$ implies $T_P(y) \geq q + \alpha_1$, we have that $\mathcal{B}_{\Phi^{-1}(1-q-\alpha_1)} \subseteq \mathcal{Y}_{nq,x}$. Using the fact that $\text{Vol}(\mathcal{B}_r) = c_d|\Sigma|^{1/2}r^d$ (where $c_d$ depends only on $d$), we arrive at the claimed upper bound. $\qed$

### 3.2.2 A Volume Condition for Safety

We consider sets of the form $\mathcal{Y}_{t+\eta,x}$ for moderate positive and negative values of $\eta$. Recall that $\mathcal{Y}_{t+\eta,x}$ is the set of all points $y$ with score $q(x; y) = nT_x(y) \geq t + \eta$, i.e., having empirical Tukey depth with respect to $x$ at least $(t + \eta)/n$. Therefore, as $\eta$ becomes smaller, the set $\mathcal{Y}_{t+\eta,x}$ grows. We show that, if the volume of $\mathcal{Y}_{t+\eta,x}$ does not increase too quickly as $\eta$ decreases, then $x$ is far from every data set in $\text{UNSAFE}_{(\varepsilon,\delta,t)}$. In particular, this implies that $x$ itself is in $\text{SAFE}_{(\varepsilon,\delta,t)}$. These lemmas do not rely on specific features of Gaussian data or Tukey depth, which enter in only in the last two stages as described above, when we argue about typical data sets. This analysis, along with the remaining accuracy analysis of Algorithm 1, is new to this work.

Before arguing about volumes directly, we prove a lemma about the weight assigned to sets by the exponential mechanism. For any set $S \subseteq \mathcal{Y}$, denote its weight by $w_x(S) = \int_S \exp\left\{\frac{\varepsilon q(x;y)}{2}\right\} dy$.

**Lemma 3.7.** *Assume $\delta < \frac{1}{2}$. If $w_x(\mathcal{Y}_{t+1,x}) \geq (1 - \delta)w_x(\mathcal{Y}_{t-1,x})$, then $x \in \text{SAFE}_{(\varepsilon,\delta',t)}$ for $\delta' = 4e^\varepsilon \delta$.*

*Proof.* First, observe that the hypothesis implies $\frac{w_x(\mathcal{Y}_{t-1,x})}{w_x(\mathcal{Y}_{t+1,x})} \leq \frac{1}{1-\delta}$. Since $\frac{1}{1-\delta} = 1 + \delta + \delta^2 + \cdots = 1 + \delta\left(\frac{1}{1-\delta}\right)$ and $\delta \leq \frac{1}{2}$, we have $\frac{w_x(\mathcal{Y}_{t-1,x})}{w_x(\mathcal{Y}_{t+1,x})} \leq 1 + 2\delta$.

Fix an event $E \subseteq \mathcal{Y}$ and a data set $x'$ adjacent to $x$. We show $\Pr[\mathcal{M}_{\varepsilon,t}(x) \in E] \le e^\varepsilon \Pr[\mathcal{M}_{\varepsilon,t}(x') \in E] + \delta'$ and $\Pr[\mathcal{M}_{\varepsilon,t}(x') \in E] \le e^\varepsilon \Pr[\mathcal{M}_{\varepsilon,t}(x) \in E] + \delta'$, which, since $x'$ is an arbitrary neighbor, establishes that $x$ is safe. The work in the proof is to use our hypothesis about $x$ to imply statements about $x'$, for which we have no explicit assumptions other than adjacency to $x$.

Let $S = \mathcal{Y}_{t,x} \cap \mathcal{Y}_{t,x'}$ be the intersection of the supports of $\mathcal{M}_{\varepsilon,t}(x)$ and $\mathcal{M}_{\varepsilon,t}(x')$. We have

$$
\begin{aligned}
\Pr[\mathcal{M}_{\varepsilon,t}(x) \in E] &= \Pr[\mathcal{M}_{\varepsilon,t}(x) \in E \cap S] + \Pr[\mathcal{M}_{\varepsilon,t}(x) \in E \setminus S] \\
&= \frac{\Pr[\mathcal{M}_{\varepsilon,t}(x) \in E \cap S]}{\Pr[\mathcal{M}_{\varepsilon,t}(x') \in E \cap S]} \Pr[\mathcal{M}_{\varepsilon,t}(x') \in E \cap S] + \Pr[\mathcal{M}_{\varepsilon,t}(x) \in E \setminus S] \\
&\le \frac{\Pr[\mathcal{M}_{\varepsilon,t}(x) \in E \cap S]}{\Pr[\mathcal{M}_{\varepsilon,t}(x') \in E \cap S]} \Pr[\mathcal{M}_{\varepsilon,t}(x') \in E] + \Pr[\mathcal{M}_{\varepsilon,t}(x) \notin \mathcal{Y}_{t,x'}].
\end{aligned}
\tag{11}
$$

We upper bound the ratio by upper bounding it for any point $y \in S$. The normalizing constants for $\mathcal{M}_{\varepsilon,t}(x)$ and $\mathcal{M}_{\varepsilon,t}(x')$ may differ, and the score functions at $y$ can differ by at most 1, so we have $\frac{\Pr[\mathcal{M}_{\varepsilon,t}(x)=y]}{\Pr[\mathcal{M}_{\varepsilon,t}(x')=y]} \le e^{\varepsilon/2} \cdot \frac{w_{x'}(\mathcal{Y}_{t,x'})}{w_x(\mathcal{Y}_{t,x})}$. Using our assumption on the volumes, we can upper bound the ratio of normalizing constants as well. The first step to do so is straightforward: for any set $A$, $w_{x'}(A) \le e^{\varepsilon/2} w_x(A)$. The second inequality, however, is subtle and uses the sensitivity of $q(\cdot;\cdot)$ in a different way: any point with score $q(x';y) \ge t$ has score $q(x;y) \ge t-1$. Thus we have $\mathcal{Y}_{t,x'} \subseteq \mathcal{Y}_{t-1,x}$ and can write

$$
\begin{aligned}
e^{\varepsilon/2} \cdot \frac{w_{x'}(\mathcal{Y}_{t,x'})}{w_x(\mathcal{Y}_{t,x})} &\le e^\varepsilon \cdot \frac{w_x(\mathcal{Y}_{t,x'})}{w_x(\mathcal{Y}_{t,x})} \\
&\le e^\varepsilon \cdot \frac{w_x(\mathcal{Y}_{t-1,x})}{w_x(\mathcal{Y}_{t+1,x})} \le e^\varepsilon(1+2\delta).
\end{aligned}
$$

Similarly, we have $\mathcal{Y}_{t+1,x} \subseteq \mathcal{Y}_{t,x'}$. This allows us to apply our hypothesis a second time.

$$
\Pr[\mathcal{M}_{\varepsilon,t}(x) \notin \mathcal{Y}_{t,x'}] \le \Pr[\mathcal{M}_{\varepsilon,t}(x) \notin \mathcal{Y}_{t+1,x}] = 1 - \frac{w_x(\mathcal{Y}_{t+1,x})}{w_x(\mathcal{Y}_{t,x})} \le 1 - \frac{w_x(\mathcal{Y}_{t+1,x})}{w_x(\mathcal{Y}_{t-1,x})} \le \delta.
$$

Thus, continuing from Equation (11), we have

$$
\begin{aligned}
\Pr[\mathcal{M}_{\varepsilon,t}(x) \in E] &\le e^\varepsilon(1+2\delta)\Pr[\mathcal{M}_{\varepsilon,t}(x') \in E] + \delta \\
&\le e^\varepsilon \Pr[\mathcal{M}_{\varepsilon,t}(x') \in E] + (1+2e^\varepsilon)\delta.
\end{aligned}
$$

We now upper bound for $\Pr[\mathcal{M}_{\varepsilon,t}(x') \in E]$ in a similar manner. First,

$$
\begin{aligned}
\Pr[\mathcal{M}_{\varepsilon,t}(x') \notin \mathcal{Y}_{t,x}] &\le \Pr[\mathcal{M}_{\varepsilon,t}(x') \in \mathcal{Y}_{t-1,x} \setminus \mathcal{Y}_{t,x}] = \frac{w_{x'}(\mathcal{Y}_{t-1,x} \setminus \mathcal{Y}_{t,x})}{w_{x'}(\mathcal{Y}_{t,x})} \\
&\le e^\varepsilon \frac{w_x(\mathcal{Y}_{t-1,x} \setminus \mathcal{Y}_{t,x})}{w_x(\mathcal{Y}_{t,x})} \\
&= e^\varepsilon \frac{w_x(\mathcal{Y}_{t-1,x}) - w_x(\mathcal{Y}_{t,x})}{w_x(\mathcal{Y}_{t,x})},
\end{aligned}
\tag{12}
$$

where the first inequality holds since $\operatorname{supp}(\mathcal{M}_{\varepsilon,t}(x')) \setminus \mathcal{Y}_{t,x} = \{y \in \mathcal{Y} : q(x;y) = t-1 \text{ and } q(x';y) = t\} \subseteq \{y \in \mathcal{Y} : q(x;y) = t-1\} = \mathcal{Y}_{t-1,x} \setminus \mathcal{Y}_{t,x}$. Since $\frac{w_x(\mathcal{Y}_{t-1,x})}{w_x(\mathcal{Y}_{t,x})} \le \frac{w_x(\mathcal{Y}_{t-1,x})}{w_x(\mathcal{Y}_{t+1,x})} \le 1+2\delta$, (12) is at most $2e^\varepsilon\delta$. For the ratio, we have

$$
\frac{\Pr[\mathcal{M}_{\varepsilon,t}(x')=y]}{\Pr[\mathcal{M}_{\varepsilon,t}(x)=y]} \le e^{\varepsilon/2} \frac{w_x(\mathcal{Y}_{t,x})}{w_{x'}(\mathcal{Y}_{t,x'})} \le e^\varepsilon \frac{w_x(\mathcal{Y}_{t,x})}{w_x(\mathcal{Y}_{t,x'})} \le e^\varepsilon \frac{w_x(\mathcal{Y}_{t-1,x})}{w_x(\mathcal{Y}_{t+1,x})} \le e^\varepsilon(1+2\delta).
$$

Thus $\Pr[\mathcal{M}_{\varepsilon,t}(x') \in E] \le e^\varepsilon \Pr[\mathcal{M}_{\varepsilon,t}(x) \in E] + 4e^\varepsilon\delta$. $\qquad\square$

We now use this lemma to establish when a data set is far from the set of unsafe data sets. Note that setting $k = 0$ below implies for all $z \in \mathtt{UNSAFE}$ that we have $D_H(x, z) > 0$, i.e., $x \in \mathtt{SAFE}$.

**Lemma 3.8.** *For any $k \geq 0$, if there exists a $g > 0$ such that $\frac{\mathrm{Vol}(\mathcal{Y}_{t-k-1,x})}{\mathrm{Vol}(\mathcal{Y}_{t+k+g+1,x})} \cdot e^{-\varepsilon g/2} \leq \delta$, then for all $z \in \mathtt{UNSAFE}_{(\varepsilon, \delta', t)}$, with $\delta' = 4e^\varepsilon \delta$, we have $D_H(x, z) > k$*

*Proof.* Take some $z$ at distance at most $k$ from $x$ (if $k = 0$, set $z \leftarrow x$). We show $z \in \mathtt{SAFE}_{(\varepsilon, \delta', t)}$. We have, from Lemma 3.7, that if $\frac{w_z(\mathcal{Y}_{t+1,z})}{w_z(\mathcal{Y}_{t-1,z})} \geq 1 - \delta$, then $z$ is safe. This assumption is equivalent to $\frac{w_z(\mathcal{Y}_{t-1,z} \setminus \mathcal{Y}_{t+1,z})}{w_z(\mathcal{Y}_{t-1,z})} \leq \delta$, which is the form we use.

First we lower bound the denominator:

$$w_z(\mathcal{Y}_{t-1,z}) \geq w_z(\mathcal{Y}_{t+g+1,z}) \geq \mathrm{Vol}(\mathcal{Y}_{t+g+1,z}) e^{\varepsilon(t+g+1)/2} \geq \mathrm{Vol}(\mathcal{Y}_{t+g+k+1,x}) e^{\varepsilon(t+g+1)/2},$$

where (crucially) the last inequality switches to the volume under $x$, and we have used the sensitivity of $q$. We use the same idea on the numerator, switching to a volume under $x$ in the first inequality:

$$w_z(\mathcal{Y}_{t-1,z} \setminus \mathcal{Y}_{t+1,z}) \leq w_z(\mathcal{Y}_{t-k-1,x} \setminus \mathcal{Y}_{t+1,z}) \leq \mathrm{Vol}(\mathcal{Y}_{t-k-1,x}) e^{\varepsilon(t+1)/2}.$$

With an upper bound on the numerator, a lower bound on the denominator, and the fact that $\frac{e^{\varepsilon(t+1)/2}}{e^{\varepsilon(t+g+1)/2}} = e^{-\varepsilon g/2}$, we have

$$\frac{w_z(\mathcal{Y}_{t-1,z} \setminus \mathcal{Y}_{t+1,z})}{w_z(\mathcal{Y}_{t-1,z})} \leq \frac{\mathrm{Vol}(\mathcal{Y}_{t-k-1,x})}{\mathrm{Vol}(\mathcal{Y}_{t+k+g+1,x})} \cdot e^{-\varepsilon g/2} \leq \delta,$$

so $z \in \mathtt{SAFE}_{(\varepsilon, \delta', t)}$. $\qquad\square$

### 3.2.3 Typical Gaussian Data Are Far from Unsafe

With Lemma 3.8, we can show that $\alpha_1$-typical data sets are far from $\mathtt{UNSAFE}$. We ask for an additional $\frac{\log(1/\beta)}{\varepsilon}$ distance beyond the threshold to ensure that we pass the distance test with high probability.

**Lemma 3.9** (Typically Far from $\mathtt{UNSAFE}$). *Assume that $x$ is $\alpha_1$-typical for $\alpha_1 \leq \frac{1}{10}$. There exists a constant $c$ such that, for any $\beta, \delta, \varepsilon > 0$ with $\varepsilon \leq 1$ and $\delta \leq \frac{1}{2}$, if $n \geq c\left(\frac{d + \log(1/\beta\delta)}{\varepsilon}\right)$ then $x$ is $\frac{\log(1/2\beta\delta)}{\varepsilon}$-far from $\mathtt{UNSAFE}_{(\varepsilon, \delta, n/4)}$.*

*Proof.* We use Lemma 3.8, which asks for a $g > 0$ such that $\frac{\mathrm{Vol}(\mathcal{Y}_{t-k-1,x})}{\mathrm{Vol}(\mathcal{Y}_{t+k+g+1,x})} \cdot e^{-\varepsilon g/2} \leq \frac{\delta}{4e^\varepsilon}$ to imply that $x$ is $k$-far from $\mathtt{UNSAFE}_{(\varepsilon, \delta, t)}$. We take $g = \frac{n}{8}$, so $t - k - 1 = n\left(\frac{1}{4} - \frac{k+1}{n}\right)$ and $t + g + k + 1 = n\left(\frac{3}{8} + \frac{k+1}{n}\right)$. We apply Lemma 3.6 to bound the ratio of volumes:

$$\frac{\mathrm{Vol}(\mathcal{Y}_{t-k-1,x})}{\mathrm{Vol}(\mathcal{Y}_{t+k+g+1,x})} \leq \left(\frac{\Phi^{-1}\left(\frac{3}{4} + \frac{k+1}{n} + \alpha_1\right)}{\Phi^{-1}\left(\frac{5}{8} - \frac{k+1}{n} - \alpha_1\right)}\right)^d. \tag{13}$$

We want both arguments to the quantile functions to be bounded away from $1/2$ and $1$, for which it suffices to use our assumption of $\alpha_1 \leq \frac{1}{10}$ and ask that $\frac{k+1}{n} < \frac{1}{100}$. This means that we must have $n \gtrsim (1/\varepsilon) \log(1/\beta\delta)$.

With both quantiles equal to constants, there is a constant $c'$ such that

$$\frac{\mathrm{Vol}(\mathcal{Y}_{t-k-1,x})}{\mathrm{Vol}(\mathcal{Y}_{t+k+g+1,x})} \cdot e^{-\varepsilon g/2} \leq e^{c'd - n\varepsilon/16}, \tag{14}$$

so we require $n \geq c\left(\frac{d + \log(1/\delta)}{\varepsilon}\right)$ for some constant $x$ to make (14) at most $\frac{\delta}{4e^\varepsilon}$, noting that $e^\varepsilon \leq e$. $\qquad\square$

### 3.2.4 Restricted Exponential Mechanism is Accurate

For our final lemma in the accuracy analysis, we show that the restricted sampler $\mathcal{M}_{\varepsilon,t}(x)$, when run on $\alpha_1$-typical data sets, with high probability returns a point with high empirical Tukey depth.

**Lemma 3.10** (Accuracy of $\mathcal{M}_{\varepsilon,t}(x)$). *Assume that $x$ is $\alpha_1$-typical for $\alpha_1 < \frac{1}{10}$. For any $\beta > 0$ and $\alpha_2 \geq 2\alpha_1$, we have, for some constant $c$,*

$$\Pr_{y \sim \mathcal{M}_{n/4}(x)}\left[T_x(y) < \frac{1}{2} - \alpha_2\right] \leq \left(\frac{c}{\alpha_2 - 2\alpha_1}\right)^d e^{-\alpha_2 n\varepsilon/4}. \tag{15}$$

*Proof.* Let BAD be the set of points with empirical Tukey depth below $\frac{1}{2} - \alpha_2$, and GOOD those with score above $\frac{1}{2} - \frac{\alpha_2}{2}$. Let $y \sim \mathcal{M}_{n/4}(x)$.

$$\Pr[y \in \text{BAD}] \leq \frac{\Pr[y \in \text{BAD}]}{\Pr[y \in \text{GOOD}]} \leq \frac{\text{Vol}(\text{BAD})\exp\left\{\frac{n\varepsilon}{2}\left(\frac{1}{2} - \alpha_2\right)\right\}}{\text{Vol}(\text{GOOD})\exp\left\{\frac{n\varepsilon}{2}\left(\frac{1}{2} - \frac{\alpha_2}{2}\right)\right\}} \leq \frac{\text{Vol}(\mathcal{Y}_{n/4,x})}{\text{Vol}(\mathcal{Y}_{n(\frac{1}{2} - \frac{\alpha_2}{2}),x})} \cdot e^{-\alpha_2 n\varepsilon/4}. \tag{16}$$

By Lemma 3.6, the ratio of volumes is at most $\left(\frac{\Phi^{-1}(3/4 + \alpha_1)}{\Phi^{-1}(\frac{1}{2} + \frac{\alpha_2}{2} - \alpha_1)}\right)^d$. Since $\alpha_1 \leq \frac{1}{10}$, $\Phi^{-1}(3/4 + \alpha_1)$ is at most a constant.

As $\alpha_2 - 2\alpha_1$ tends to 0, the demoninator approaches 0 as well. To finish the proof, then, we show that, for any $z > 0$, $\Phi^{-1}\left(\frac{1}{2} + z\right) \geq \sqrt{2\pi}z$ or, equivalently, $\frac{1}{2} + z \geq \Phi(\sqrt{2\pi}z)$. Since $e^{-x^2/2} \leq 1$, we have

$$\Phi(\sqrt{2\pi}z) \leq \frac{1}{2} + \frac{1}{\sqrt{2\pi}}\int_0^{\sqrt{2\pi}z} 1\,dx = \frac{1}{2} + z. \tag{17}$$

$\square$

We are now ready to prove the main theorem.

*Proof of Theorem 3.2.* Set $\alpha_1 = c_0\alpha$ for a constant $c_0$ to be determined later, and set $\alpha_2 = 3\alpha_1$. By Lemma 3.5, with probability at least $1 - \beta$, $x$ is $\alpha_1$-typical. If $x$ is $\alpha_1$-typical, by Lemma 3.9 it is at least $\frac{\log(1/2\delta\beta)}{\varepsilon}$-far from $\text{UNSAFE}_{(\varepsilon,\delta,t)}$. This implies that Algorithm 1 returns FAIL with probability at most $2\beta$: by the CDF of the Laplace distribution,

$$\Pr[\text{FAIL}] \leq \Pr[x \text{ not } \alpha_1\text{-typical}] + \Pr\left[\frac{\log(1/2\delta\beta)}{\varepsilon} + Z \leq \frac{\log(1/2\delta)}{\varepsilon}\right] \tag{18}$$

$$\leq \beta + \Pr\left[Z \leq -\frac{\log(1/\beta)}{\varepsilon}\right] = \beta + \frac{\beta}{2}. \tag{19}$$

If $x$ is $\alpha_1$-typical and we don't return FAIL, we instead return a sample from $\mathcal{M}_{n/4}(x)$. Lemma 3.10 tells us that, for $\alpha_1$-typical $x$,

$$\Pr_{y \sim \mathcal{M}_{n/4}(x)}\left[T_x(y) < \frac{1}{2} - \alpha_2\right] \leq \left(\frac{c}{\alpha_2 - 2\alpha_1}\right)^d e^{-\alpha_2 n\varepsilon/4} \leq e^{d\log(c/\alpha_1) - \alpha_2 n\varepsilon/4}, \tag{20}$$

using $\alpha_2 = 3\alpha_1$. Since $n$ is sufficiently large, this is at most $\beta$.

So with probability at least $1 - 3\beta$, we have $T_x(y) \geq \frac{1}{2} - \alpha_2$. Since $x$ is $\alpha_1$-typical, we have

$$T_P(y) \geq \frac{1}{2} - \alpha_1 - \alpha_2 = \frac{1}{2} - 4\alpha_1. \tag{21}$$

Recall $\Phi(-\|y - \mu\|_\Sigma) = T_P(y)$. By definition, $\Phi(-z) = \frac{1}{2} - \frac{1}{2}\text{Erf}\left(\frac{z}{\sqrt{2}}\right)$. It is easy to see that $\text{Erf}(x) \geq \text{Erf}(1) \cdot x \geq 0.84x$ for $x \in [0, 1]$ (see e.g. [16, Lemma 3.2]). It follows that

$$\Phi(-z) \leq \frac{1}{2} - \frac{0.84z}{2\sqrt{2}}.$$

Combining the above inequalities, we have that $\|y - \mu\|_\Sigma \leq \frac{8\sqrt{2}}{0.84}\alpha_1 \leq 14\alpha_1$. Setting $c_0 = 1/14$ makes this term at most $\alpha$.

Privacy follows from a standard calculation, provided in Appendix D as Proposition D.1. □

# 4 Empirically Rescaled Gaussian Mechanism

In this section, we describe our second estimator. At a high level, the estimator first privately checks whether the data set $x$ is $\frac{1}{\varepsilon}$-close in Hamming distance to a good set of "roughly Gaussian" data sets $\mathcal{G}(\lambda)$. If so, it finds the closest data set to $x$ that belongs in $\mathcal{G}(\lambda)$, which we call $\tilde{x}$. Then it returns a sample $\hat{\mu}$ drawn from $\mathcal{N}(\mu_{\tilde{x}}, C^2\Sigma_{\tilde{x}})$, where $\mu_{\tilde{x}}$ and $\Sigma_{\tilde{x}}$ are the empirical mean and covariance of $\tilde{x}$ and $C$ is a scale parameter appropriately set to ensure privacy.

Specifically, we define the empirical mean and covariance in the following (slightly non-standard) way:

**Definition 4.1** (Empirical Mean and Covariance). For data set $x \in \mathbb{R}^{3n \times d}$, the empirical mean and covariance of $x$ are respectively defined by

$$\mu_x = \frac{1}{n}\sum_{i=1}^n x_{i+2n} \quad \text{and} \quad \Sigma_x = \frac{1}{2n}\sum_{i=1}^n (x_i - x_{i+n})(x_i - x_{i+n})^T.$$

In comparison with the standard empirical estimators, ours enable a simpler privacy analysis, since replacing one datapoint in $x$ only affects one term in one of the sums. For convenience, we choose the number of samples to be $3n$ so that we can pair the first two thirds to construct $\Sigma_x$ and use the last third for $\mu_x$. Note that before accessing the data set, the algorithm randomly permutes all data points (line 3) – a technicality which pertains to the fact that $\Sigma_x$ is order-dependent. With these definitions at hand, the good set $\mathcal{G}(\lambda)$ is defined as follows:

**Definition 4.2** ($\lambda$-goodness). For any $\lambda > 0$, define $\mathcal{G}(\lambda) \subseteq \mathbb{R}^{3n \times d}$ as

$$\mathcal{G}(\lambda) \stackrel{\text{def}}{=} \left\{ x \in \mathbb{R}^{3n \times d} : \Sigma_x \text{ is invertible and } \forall i \in [3n] \ \|x_i - \mu_x\|_{\Sigma_x}^2 \leq \lambda \right\}.$$

We set $\lambda \approx d \log n$, since for this value (sub)Gaussian data will belong in $\mathcal{G}(\lambda)$ with high probability.

Finally, we note that the algorithm immediately aborts if the number of samples is less than $\frac{k\lambda \log(1/\delta)}{\varepsilon} \approx \frac{d \log(1/\delta)}{\varepsilon^2}$, a condition necessary to ensure privacy. The parameter $k \approx \frac{\log(1/\delta)}{\varepsilon}$ is an upper bound on the Hamming distance between the projections $\tilde{x}, \tilde{y}$ of any two neighboring data sets $x, y$ that pass the check in line 5, and, along with $\lambda$, plays an important role in the privacy analysis.

---

**Algorithm 2** Empirically Rescaled Gaussian Mechanism $\mathcal{A}_{\varepsilon,\delta,\beta}^G(x)$

---

**Require:** Data set $x = (x_1, \ldots, x_{3n})^T \in \mathbb{R}^{3n \times d}$. Privacy parameters $\varepsilon, \delta > 0$. Failure probability $\beta > 0$.

1: Initialize: $\lambda \leftarrow O\left(d \log \frac{n}{\beta}\right)$, $t \leftarrow \frac{1}{\varepsilon}\log\frac{1}{\beta}$, $k \leftarrow \frac{2}{\varepsilon}\log\frac{1}{\delta\beta} + 1$, $C^2 \leftarrow \frac{32k^2}{\varepsilon^2 n^2} \cdot \frac{\lambda}{1-2k\lambda/n} \cdot \log\frac{1.25}{\delta}$.

2: **if** $n = o\left(\frac{k\lambda}{\varepsilon}\log\frac{1}{\delta}\right)$ **then return** FAIL.

3: $\bar{x} \leftarrow \sigma(x)$                                   ▷ random permutation $\sigma : (\mathbb{R}^d)^{3n} \to (\mathbb{R}^d)^{3n}$

4: $h \leftarrow D_H(\bar{x}, \mathcal{G}(\lambda))$                          ▷ distance between $\bar{x}$ and $\lambda$-goodness

5: **if** $h + r > t$ for $r \sim \text{Lap}(1/\varepsilon)$ **then return** FAIL.

6: $\tilde{x} \leftarrow \arg\min_{z \in \mathcal{G}(\lambda)} D_H(\bar{x}, z)$                     ▷ projection to $\lambda$-goodness

7: **return** $\hat{\mu} \sim \mathcal{N}(\mu_{\tilde{x}}, C^2\Sigma_{\tilde{x}})$

---

We remark that the sample size check in line 2 and the setting of $\lambda$ are not well-defined, as they are stated with asymptotic notation. Although it is possible to compute the constants for these steps, we exclude them in favor of a cleaner analysis.

For simplicity, in this section we focus on Gaussian data. For the more general discussion for subgaussian data sets, see Appendix B.

**Theorem 4.3** (Privacy and Accuracy of the Empirically Rescaled Gaussian Mechanism). *For any $\varepsilon > 0$, $0 < \delta < 1$, Algorithm 2 is $(3\varepsilon, e^{\varepsilon}(1 + e^{\varepsilon})\delta)$-differentially private. There exists an absolute constant $C$ such that, for any $0 < \alpha, \beta, \varepsilon, \delta < 1$, mean $\mu$, and positive definite $\Sigma$, if $x \sim \mathcal{N}(\mu, \Sigma)^{\otimes n}$ and*

$$n \geq C\left(\frac{d}{\alpha^2}\log\frac{1}{\beta} + \frac{d}{\alpha\varepsilon^2}\log^3\frac{1}{\delta\beta} \cdot \log\frac{d\log(1/\delta\beta)}{\alpha\varepsilon}\right), \tag{22}$$

*then with probability at least $1 - 3\beta$, Algorithm 2 returns $\mathcal{A}^G_{\varepsilon,\delta,\beta}(x) = \hat{\mu}$ such that $\|\hat{\mu} - \mu\|_{\Sigma} \leq \alpha$.*

The proof of Theorem 4.3 follows by a combination of the accuracy and privacy guarantees of the algorithm, stated in Theorem 4.9 and Corollary 4.14 which we prove in the next two sections.

## 4.1 Accuracy Analysis

The crux of the proof of the sample complexity guarantee (Theorem 4.9) is the following. Suppose $n$ is large enough so that the algorithm does not fail in line 2.

- If $x \sim \mathcal{N}(\mu, \Sigma)^{\otimes 3n}$ and the number of samples is $n = O(d + \log(1/\beta))$, then with probability $1 - \beta$ over the draw of $x$, the data set $x$ is in the good set $\mathcal{G}(\lambda)$ for $\lambda = \tilde{O}(d)$ (Lemma 4.7). In particular, this holds for the permuted data set $\bar{x}$ as this is also drawn from $\mathcal{N}(\mu, \Sigma)^{\otimes 3n}$.

- This implies that with high probability over the randomness of the algorithm, $\bar{x}$ passes the Hamming distance check in line 5 and the projection of line 6 leaves it intact so that $\tilde{x} = \bar{x}$.

- It then suffices to prove that, with high probability, the returned estimator $\hat{\mu} \sim \mathcal{N}(\mu_{\bar{x}}, C^2\Sigma_{\bar{x}})$ is a good approximation of the true mean $\mu$ measured by the Mahalanobis distance with respect to the true $\Sigma$, that is, $\|\hat{\mu} - \mu\|_{\Sigma} = \tilde{O}(\sqrt{d/n} + d/\varepsilon^2 n)$ (Lemma 4.8).

For a short review of basic linear algebra facts, see Appendix A. We start by presenting a few known facts we will use in this subsection. First, we prove the following proposition, which states that if two matrices $\Sigma_1, \Sigma_2$ are good spectral approximations of one another, then the Mahalanobis distance of any vector with respect to $\Sigma_1$ is close to the one with respect to $\Sigma_2$ and vice versa.

**Proposition 4.4.** *For positive definite matrices $\Sigma_1, \Sigma_2$, if there exists a constant $\gamma \in (0, 1)$ such that*

$$(1 - \gamma)\Sigma_1 \preceq \Sigma_2 \preceq (1 + \gamma)\Sigma_1,$$

*then for any vector $v$ we have*

$$\frac{1}{\sqrt{1 + \gamma}}\|v\|_{\Sigma_1} \leq \|v\|_{\Sigma_2} \leq \frac{1}{\sqrt{1 - \gamma}}\|v\|_{\Sigma_1}.$$

*Proof.* We upper bound $\|v\|_{\Sigma_2}$; the lower bound is analogous. Since the matrices are invertible, we have $\Sigma_1^{-1} \succeq (1 - \gamma)\Sigma_2^{-1}$, i.e. $\Sigma_1^{-1} - (1 - \gamma)\Sigma_2^{-1}$ is psd. So we have

$$\begin{aligned}
(1 - \gamma)\|v\|_{\Sigma_2}^2 &= (1 - \gamma)\|v\|_{\Sigma_2}^2 - \|v\|_{\Sigma_1}^2 + \|v\|_{\Sigma_1}^2 \\
&= v^T\left((1 - \gamma)\Sigma_2^{-1} - \Sigma_1^{-1}\right)v + \|v\|_{\Sigma_1}^2 \\
&= \|v\|_{\Sigma_1}^2 - v^T\left(\Sigma_1^{-1} - (1 - \gamma)\Sigma_2^{-1}\right)v \\
&\leq \|v\|_{\Sigma_1}^2,
\end{aligned}$$

applying the fact that $v^T A v \geq 0$ for psd matrices. $\qquad\square$

We will also make use of the following standard concentration inequalities for Gaussian random variables. For a reference, see [22]. The formulation used here is from [50, Fact 3.4].

**Lemma 4.5.** *Let $u_i \sim \mathcal{N}(0, \mathbb{I})$ be i.i.d. samples for $i \in [n]$. Define the estimator $\hat{\Sigma} = \frac{1}{n} \sum_{i=1}^n u_i u_i^T$. For every $\beta > 0$, with probability at least $1 - \beta$ the following conditions both hold:*

$$\left(1 - O\left(\sqrt{\frac{d + \log(1/\beta)}{n}}\right)\right) \cdot \mathbb{I} \preceq \hat{\Sigma} \preceq \left(1 + O\left(\sqrt{\frac{d + \log(1/\beta)}{n}}\right)\right) \cdot \mathbb{I} \tag{23}$$

$$\forall i \in [n] \quad \|u_i\|_2^2 \leq O(d \log(n/\beta)) \tag{24}$$

The following generalization to non-spherical Gaussians is a straighforward implication.

**Lemma 4.6.** *Suppose $u_1, \ldots, u_n$ satisfy inequalities (23) and (24). Let $\Sigma > 0$ and let $\lambda_1$ be the largest eigenvalue of $\Sigma$. Let $z_i = \Sigma^{1/2} u_i$ for all $i \in [n]$ and define $\hat{\Sigma}_z = \frac{1}{n} \sum_{i=1}^n z_i z_i^T$. Then the following conditions both hold:*

$$\left(1 - O\left(\sqrt{\frac{d + \log(1/\beta)}{n}}\right)\right) \cdot \Sigma \preceq \hat{\Sigma}_z \preceq \left(1 + O\left(\sqrt{\frac{d + \log(1/\beta)}{n}}\right)\right) \cdot \Sigma$$

$$\forall i \in [n] \quad \|z_i\|_2^2 \leq O(\lambda_1 d \log(n/\beta))$$

We now begin the accuracy analysis.

In the next Lemma 4.7 we prove that if $n \gtrsim d$ then Gaussian data sets fall into the good set $\mathcal{G}(\lambda)$ with high probability. Intuitively, this holds since Gaussian data are already likely to satisfy the condition $\|x_i - \mu\|_\Sigma \leq \lambda$, which by design is the same as the condition on the good set, except that the true parameters are replaced by the empirical ones. The assumption $n \gtrsim d$ ensures that the empirical and true parameters are close.

**Lemma 4.7.** *Let $x \sim \mathcal{N}(\mu, \Sigma)^{\otimes 3n}$ and $n = \Omega(d + \log(1/\beta))$. There exists a $\lambda = O(d \log(n/\beta))$ such that, with probability at least $1 - \beta$ we have $x \in \mathcal{G}(\lambda)$.*

*Proof.* Since Mahalanobis distance is invariant to both changes in mean and full-rank transformations, it suffices to prove this claim for $x \sim \mathcal{N}(0, \mathbb{I})^{\otimes 3n}$.

Taking $n = \Omega(d + \log(1/\beta))$, we have that there exists a constant $\gamma \in (0, 1)$ so that $O\left(\sqrt{\frac{d + \log(1/\beta)}{n}}\right)$ is less than $\gamma$. By Lemma 4.5 with probability $1 - \beta$,

$$(1 - \gamma) \cdot \mathbb{I} \preceq \Sigma_x \preceq (1 + \gamma) \cdot \mathbb{I} \tag{25}$$

and

$$\forall i \in [3n] \quad \|x_i - \mu\|_2 \leq O(\sqrt{d \log(n/\beta)}). \tag{26}$$

These equations and Proposition 4.4 imply that $\Sigma_x$ is invertible and that for all $i$, $\|x_i - \mu\|_{\Sigma_x} = O(\sqrt{d \log(n/\beta)})$. Furthermore, Equation (26) implies $\|\mu_x - \mu\|_{\Sigma_x} \leq \frac{1}{n} \sum_{i=1}^n \|x_{i+2n} - \mu\|_{\Sigma_x} = O(\sqrt{d \log(n/\beta)})$ by the triangle inequality. We finish the proof by applying the triangle inequality one more time: $\|x_i - \mu_x\|_{\Sigma_x} \leq \|x_i - \mu\|_{\Sigma_x} + \|\mu - \mu_x\|_{\Sigma_x} = O(\sqrt{d \log(n/\beta)})$. $\square$

The next lemma bounds the error $\|\hat{\mu} - \mu\|_\Sigma$ for $\hat{\mu} \sim \mathcal{N}(\mu_x, C^2 \Sigma_x)$, where $x \sim \mathcal{N}(\mu, \Sigma)^{\otimes 3n}$. It follows directly from Gaussian concentration. The condition on the number of samples serves two purposes. First, $n = \Omega(d + \log(1/\beta))$ is required so that the empirical covariance $\Sigma_x$ is a good spectral approximation of the true covariance $\Sigma$, as before. Second, $n = \Omega(k\lambda)$ is required so that the parameter $C^2$ is well-defined. Recalling the setting of parameters $k = O(\log(1/\delta\beta)/\varepsilon)$ and $\lambda = O(d \log(n/\beta))$, both these conditions are satisfied as long as $n = \tilde{O}(d/\varepsilon)$.

**Lemma 4.8.** *Suppose $x \sim \mathcal{N}(\mu, \Sigma)^{\otimes 3n}$ and $n = \Omega(\max\{(d + \log(1/\beta)), k\lambda\})$, where parameters $k, \lambda$ are set as in Algorithm 2. Then with probability at least $1 - \beta$, for $\hat{\mu} \sim \mathcal{N}(\mu_x, C^2 \Sigma_x)$,*

$$\|\hat{\mu} - \mu\|_\Sigma = O\left(\sqrt{\frac{d}{n} \cdot \log \frac{1}{\beta}} + \frac{d}{\varepsilon^2 n} \log^2 \frac{1}{\delta\beta} \cdot \sqrt{\log \frac{n}{\beta}}\right).$$

*Proof.* By triangle inequality, we have that

$$\|\hat{\mu} - \mu\|_\Sigma \le \|\mu - \mu_x\|_\Sigma + \|\mu_x - \hat{\mu}\|_\Sigma.$$

We focus on the first term $\|\mu - \mu_x\|_\Sigma = \|\frac{1}{n}\sum_{i=1}^n \Sigma^{-1/2}(x_{i+2n} - \mu)\|_2 = \|\frac{1}{n}\sum_{j=1}^n u_i\|_2$, where $u_i \sim \mathcal{N}(0, \mathbb{I})$ for all $i \in [n]$. Since $\frac{1}{n}\sum_{j=1}^n u_i \sim \mathcal{N}(0, \frac{1}{n}\mathbb{I})$, we can write $\|\frac{1}{n}\sum_{j=1}^n u_i\|_2 = \frac{1}{\sqrt{n}}\|u'\|_2$ for $u' \sim \mathcal{N}(0, \mathbb{I})$. By Lemma 4.5, we have that $\|u'\|_2^2 = O(d\log(1/\beta))$ with probability at least $1 - \beta/2$. So with probability at least $1 - \beta/2$, it holds that

$$\|\mu - \mu_x\|_\Sigma = O\left(\sqrt{\frac{d}{n} \cdot \log\frac{1}{\beta}}\right). \tag{27}$$

We now give an upper bound for the second term. Notice that if we let $z_i = (x_i - x_{i+n})/\sqrt{2}$, then for all $i \in [n]$ $z_i$ are i.i.d. samples from $\mathcal{N}(0, \Sigma)$ and $\Sigma_x = \frac{1}{n}\sum_{i=1}^n z_i z_i^T$. Taking $n = \Omega(d + \log(1/\beta))$, so that $O\left(\sqrt{\frac{d+\log(1/\beta)}{n}}\right)$ is a sufficiently small constant $\gamma$, by Lemma 4.6 with probability $1 - \beta/4$ we have $(1 - \gamma) \cdot \Sigma \preceq \Sigma_x \preceq (1 + \gamma) \cdot \Sigma$.

It follows that, by Proposition 4.4, $\|\hat{\mu} - \mu_x\|_\Sigma = O(\|\hat{\mu} - \mu_x\|_{\Sigma_x})$. So it suffices to bound $\|\hat{\mu} - \mu_x\|_{\Sigma_x} = \|C^{-1}\Sigma_x^{-1/2}(\hat{\mu} - \mu_x)\|_2 \cdot C$.

Since $\hat{\mu} \sim \mathcal{N}(\mu_x, C^2\Sigma_x)$, equivalently, we have that $u = C^{-1}\Sigma_x^{-1/2}(\hat{\mu} - \mu_x) \sim \mathcal{N}(0, \mathbb{I})$. By Lemma 4.5, we have that with probability at least $1 - \beta/4$, $\|u\|_2^2 = O(d\log(1/\beta))$. Therefore, by union bound, with probability at least $1 - \beta/2$,

$$\|\hat{\mu} - \mu_x\|_\Sigma = O\left(C\sqrt{d\log\frac{1}{\beta}}\right). \tag{28}$$

Combining Equation (27) and (28), by union bound, with probability at least $1 - \beta$, it holds that

$$
\begin{aligned}
\|\hat{\mu} - \mu\|_\Sigma &= O\left(\sqrt{\frac{d}{n} \cdot \log\frac{1}{\beta}} + C\sqrt{d\log\frac{1}{\beta}}\right) \\
&= O\left(\sqrt{\frac{d}{n} \cdot \log\frac{1}{\beta}} + \frac{k}{\varepsilon n}\sqrt{\frac{\lambda}{1 - 2k\lambda/n}\log\frac{1.25}{\delta}}\sqrt{d\log\frac{1}{\beta}}\right) && \text{(substituting } C) \\
&= O\left(\sqrt{\frac{d}{n} \cdot \log\frac{1}{\beta}} + \frac{k}{\varepsilon n}\sqrt{d\lambda\log\frac{1}{\delta} \cdot \log\frac{1}{\beta}}\right) && \text{(since } n = \Omega(k\lambda)) \\
&= O\left(\sqrt{\frac{d}{n} \cdot \log\frac{1}{\beta}} + \frac{kd}{\varepsilon n}\sqrt{\log\frac{1}{\delta} \cdot \log\frac{1}{\beta} \cdot \log\frac{n}{\beta}}\right) && \text{(substituting } \lambda) \\
&= O\left(\sqrt{\frac{d}{n} \cdot \log\frac{1}{\beta}} + \frac{d}{\varepsilon^2 n}\left(\log\frac{1}{\delta} + \log\frac{1}{\beta}\right)\sqrt{\log\frac{1}{\delta} \cdot \log\frac{1}{\beta} \cdot \log\frac{n}{\beta}}\right) && \text{(substituting } k) \\
&= O\left(\sqrt{\frac{d}{n} \cdot \log\frac{1}{\beta}} + \frac{d}{\varepsilon^2 n}\log^2\frac{1}{\delta\beta} \cdot \sqrt{\log\frac{n}{\beta}}\right).
\end{aligned}
$$

This completes the proof of the lemma. $\square$

We are now ready to state the sample complexity of Algorithm 2, putting together the lemmas above.

**Theorem 4.9** (Accuracy of $\mathcal{A}_{\varepsilon,\delta,\beta}^G(x)$). *There exists an absolute constant $C$ such that, for any $0 < \alpha, \beta, \varepsilon, \delta < 1$, mean $\mu$, and positive definite $\Sigma$, if $x \sim \mathcal{N}(\mu, \Sigma)^{\otimes 3n}$ and*

$$n \ge C\left(\frac{d}{\alpha^2}\log\frac{1}{\beta} + \frac{d}{\alpha\varepsilon^2}\log^3\frac{1}{\delta\beta} \cdot \log\frac{d\log(1/\delta\beta)}{\alpha\varepsilon}\right), \tag{29}$$

*then with probability* $1 - 3\beta$, *Algorithm* 2 *returns* $\mathcal{A}^G_{\varepsilon,\delta,\beta}(x) = \hat{\mu}$ *such that* $\|\hat{\mu} - \mu\|_\Sigma \leq \alpha$.

*Proof.* First, we argue that for this sample complexity, the algorithm does not return FAIL in line 2. Recall that the condition is

$$n = \Omega\left(\frac{k\lambda}{\varepsilon} \log \frac{1}{\delta}\right). \tag{30}$$

Substituting the terms $k = \frac{2}{\varepsilon} \log \frac{1}{\delta\beta} + 1$ and $\lambda = O(d \log(n/\beta))$ in condition (30), we have that

$$\frac{k\lambda}{\varepsilon} \log \frac{1}{\delta} = O\left(\frac{d}{\varepsilon^2} \log \frac{1}{\delta\beta} \cdot \log \frac{n}{\beta} \cdot \log \frac{1}{\delta}\right) = O\left(\frac{d}{\varepsilon^2} \log^3 \frac{1}{\delta\beta} \cdot \log n\right).$$

For some absolute constant $C$, we let

$$n \geq C\left(\frac{d}{\alpha^2} \log \frac{1}{\beta} + \frac{d}{\alpha\varepsilon^2} \log^3 \frac{1}{\delta\beta} \cdot \log \frac{d \log(1/\delta\beta)}{\alpha\varepsilon}\right). \tag{31}$$

By straightforward calculations and since $\alpha \leq 1$, we can see that the sample size of Eq. (31) above suffices for $n$ to satisfy condition (30), and so Algorithm 2 does not fail in line 2.

Since $\bar{x}$ is a permutation of $x$, it holds that $\bar{x} \sim \mathcal{N}(\mu, \Sigma)^{\otimes 3n}$ as well. Note that the number of samples in Eq. (31) satisfies $n = \Omega(d + \log(1/\beta))$. This implies that the assumptions of Lemma 4.7 are satisfied and thus it holds that, with probability $1 - \beta$, $\bar{x} \in \mathcal{G}(\lambda)$.

It follows that the Hamming distance of $\bar{x}$ from the good set in line 4 is $h = 0$. Since $r \sim \text{Lap}(1/\varepsilon)$, by concentration of the Laplace distribution, it holds that $|r| \leq \frac{1}{\varepsilon} \log \frac{1}{\beta}$ with probability $1 - \beta$. Thus, by union bound, with probability $1 - 2\beta$, $h + r \leq \frac{1}{\varepsilon} \log \frac{1}{\beta}$. It follows that, with probability $1 - 2\beta$, we do not fail in line 5 and we reach line 6, where the projection step leaves the data set unchanged, that is, $\tilde{x} = \bar{x}$, since $\bar{x} \in \mathcal{G}(\lambda)$.

So far, we have proven that with probability $1 - 2\beta$, Algorithm 2 does not fail in any step and returns $\hat{\mu} \sim \mathcal{N}(\mu_{\bar{x}}, C^2\Sigma_{\bar{x}})$ in line 7, where $\bar{x} \sim \mathcal{N}(\mu, \Sigma)^{\otimes 3n}$. Now, notice that the sample complexity stated in Eq. (31) satisfies the condition $n = \Omega(\max\{(d + \log(1/\beta)), k\lambda\})$ as well. Since $\bar{x} \sim \mathcal{N}(\mu, \Sigma)^{\otimes 3n}$, the assumptions of Lemma 4.8 are satisfied and therefore, by union bound, with probability at least $1 - 3\beta$, Algorithm 2 returns $\hat{\mu}$ such that

$$\|\hat{\mu} - \mu\|_\Sigma = O\left(\sqrt{\frac{d}{n} \cdot \log \frac{1}{\beta}} + \frac{d}{\varepsilon^2 n} \log^2 \frac{1}{\delta\beta} \cdot \sqrt{\log \frac{n}{\beta}}\right). \tag{32}$$

The proof is complete by observing that for the stated sample complexity and the right choice of constant $C$ in Eq. (31), the error in Eq. (32) is upper bounded so that $\|\hat{\mu} - \mu\|_\Sigma \leq \alpha$. □

## 4.2 Privacy Analysis

We state the privacy guarantee of our algorithm in Corollary 4.14. We consider two neighboring data sets $x, y$ and that they are "aligned," i.e. their Hamming distance is minimized and $D_H(x, y) \leq 1$. We show that due to the permutation step in line 3, it suffices to prove the privacy guarantee for this case (Lemma D.6).

The private check in line 5 of the algorithm ensures that for two data sets with Hamming distance 1, the probabilities of failing under $x, y$ are indistinguishable. If $x, y$ are far from the good set, then they both fail with high probability. On the other hand, if $x, y$ are close to the good set, then their projections $\tilde{x}, \tilde{y}$ are close to each other, i.e. $D_H(\tilde{x}, \tilde{y}) \leq k$. In particular, this implies that the estimators $\Sigma_{\tilde{x}}, \Sigma_{\tilde{y}}$ are "close" (in a sense established in Section 4.2.1) since they differ in at most $k$ terms and each term is bounded (because $\tilde{x}, \tilde{y} \in \mathcal{G}(\lambda)$).

Our main result is Theorem 4.13, which states that any two nearby and good data sets $\tilde{x}, \tilde{y}$ have empirical estimators that induce indistinguishable output distributions $\mathcal{N}(\mu_{\tilde{x}}, C^2\Sigma_{\tilde{x}})$ and $\mathcal{N}(\mu_{\tilde{y}}, C^2\Sigma_{\tilde{y}})$. The proof is broken into two parts:

1. First, we "change the mean" and show that $\mathcal{N}(\mu_{\tilde{x}}, C^2\Sigma_{\tilde{x}}) \approx_{\varepsilon,\delta} \mathcal{N}(\mu_{\tilde{y}}, C^2\Sigma_{\tilde{x}})$, which is equivalent to $\mathcal{N}(\Sigma_{\tilde{x}}^{-1/2}(\mu_{\tilde{x}} - \mu_{\tilde{y}}), C^2\mathbb{I}) \approx_{\varepsilon,\delta} \mathcal{N}(0, C^2\mathbb{I})$ (Lemma 4.19). This follows by an application of the Gaussian mechanism for the right choice of parameter $C$.

2. Second, we "change the covariance" and show that $\mathcal{N}(\mu_{\tilde{y}}, C^2\Sigma_{\tilde{x}}) \approx_{\varepsilon_1,\delta} \mathcal{N}(\mu_{\tilde{y}}, C^2\Sigma_{\tilde{y}})$, for $\varepsilon_1 = O\left(\frac{k\lambda}{n-k\lambda}\log\frac{1}{\delta}\right)$, which is equivalent to $\mathcal{N}(0, \Sigma_{\tilde{x}}) \approx_{\varepsilon_1,\delta} \mathcal{N}(0, \Sigma_{\tilde{y}})$ (Lemma 4.15). Notice that if we want $\varepsilon_1 \leq \varepsilon$, then it has to be the case that $n = \Omega\left(\frac{k\lambda}{\varepsilon}\log\frac{1}{\delta}\right) = \Omega\left(\frac{d}{\varepsilon^2}\cdot\text{polylog}\left(\frac{d\log(1/\beta\delta)}{\varepsilon}\right)\right)$, which is the condition in line 2 of Algorithm 2.

### 4.2.1 Implications of Goodness

Before directly addressing privacy, we state a few lemmas that follow from the goodness assumption. The proofs, provided in Appendix D, require only elementary linear algebra.

**Lemma 4.10.** *If $x \in \mathcal{G}(\lambda)$, then for any indices $i, j \in [3n]$ we have*

$$(x_i - x_j)^T \Sigma_x^{-1}(x_i - x_j) \leq 4\lambda.$$

*In particular, this applies to $u_i^T \Sigma_x^{-1} u_i$ for all $i \in [n]$, where $u_i = x_i - x_{i+n}$.*

**Lemma 4.11.** *Suppose $x, y \in \mathcal{G}(\lambda)$ and $D_H(x, y) \leq k$, with $2k\lambda < n$. For any vector $v$ we have*

$$v^T \Sigma_y^{-1} v \leq \frac{1}{1 - 2k\lambda/n}\cdot v^T \Sigma_x^{-1} v.$$

**Lemma 4.12.** *Suppose $x, y \in \mathcal{G}(\lambda)$ and $D_H(x, y) \leq k$, with $2k\lambda < n$. Then*

$$\|\Sigma_x^{-1/2}\Sigma_y\Sigma_x^{-1/2} - \mathbb{I}\|_{\text{tr}} \leq 2k\lambda\left(\frac{1}{n - 2k\lambda} + \frac{1}{n}\right)$$

$$\|\Sigma_y^{-1/2}\Sigma_x\Sigma_y^{-1/2} - \mathbb{I}\|_{\text{tr}} \leq 2k\lambda\left(\frac{1}{n - 2k\lambda} + \frac{1}{n}\right)$$

### 4.2.2 Proof of Differential Privacy

We are now ready to prove the privacy guarantees of Algorithm 2. Unlike the standard empirical estimators, our definition of $\Sigma_x$ is not invariant with respect to reordering the data. As a result, the covariance $\Sigma_y$ of an adjacent data set $y$ could differ in an arbitrary number of terms. To simplify our analysis, we establish indistinguishability for adjacent data sets that are "aligned," i.e., they have Hamming distance 1. Because of the data-order permutation step in Algorithm 2, we can extend this to apply more generally to adjacent data sets (interpreted as multisets) which differ in a single data point, as required by the standard definition of differential privacy (see Proposition D.6 in Appendix D).

The main result, Theorem 4.13, is that any two nearby, good data sets have empirical estimators that induce indistinguishable output distributions. With this in hand, overall privacy of Algorithm 2 follows from a standard calculation, included in Appendix D.2.

**Theorem 4.13.** *For any $\varepsilon, \delta, \lambda, k, n > 0$ such that*

$$n > 2k\lambda \quad \text{and} \quad \varepsilon \geq 10k\lambda\left(\frac{1}{n - 2k\lambda} + \frac{1}{n}\right)\log\frac{2}{\delta},$$

*set*

$$C^2 = \frac{32k^2}{\varepsilon^2 n^2}\cdot\frac{\lambda}{1 - 2k\lambda/n}\cdot\log\frac{1.25}{\delta}.$$

*For any data sets $x, y \in \mathcal{G}(\lambda)$ of size $3n$ such that $D_H(x, y) \leq k$, we have $\mathcal{N}(\mu_x, C^2\Sigma_x) \approx_{2\varepsilon,(1+e^\varepsilon)\delta} \mathcal{N}(\mu_y, C^2\Sigma_y)$.*

**Corollary 4.14** (Privacy of $\mathcal{A}^G_{\varepsilon,\delta,\beta}(x)$). *Algorithm 2 is $(3\varepsilon, e^\varepsilon(1 + e^\varepsilon)\delta)$-differentially private.*

Theorem 4.13 follows from the triangle inequality for indistinguishability[5] and Lemmas 4.15 and 4.19: the first establishes $\mathcal{N}(\mu_y, C^2\Sigma_y) \approx \mathcal{N}(\mu_y, C^2\Sigma_x)$ and the second gives us $\mathcal{N}(\mu_y, C^2\Sigma_x) \approx \mathcal{N}(\mu_x, C^2\Sigma_x)$.

**Lemma 4.15.** *Suppose $x, y \in \mathcal{G}(\lambda)$ and $D_H(x, y) \le k$. For any $\delta \in (0, 1)$, if $2k\lambda < n$ and*

$$\varepsilon \ge 10k\lambda\left(\frac{1}{n - 2k\lambda} + \frac{1}{n}\right)\log\frac{2}{\delta},$$

*then $\mathcal{N}(0, \Sigma_x) \approx_{\varepsilon,\delta} \mathcal{N}(0, \Sigma_y)$.*

Note that this implies indistinguishability for any bijection of these two distributions. In particular, we have $\mathcal{N}(\mu_y, C^2\Sigma_y) \approx_{\varepsilon,\delta} \mathcal{N}(\mu_y, C^2\Sigma_x)$. For this proof, we use the Hanson-Wright Inequality, stated in the next lemma (see [56] for this formulation).

**Lemma 4.16** (Hanson-Wright Inequality). *Let $u \sim \mathcal{N}(0, \mathbb{I})$ and $D \in \mathbb{R}^{d \times d}$. Then, with probability $1 - \beta$,*

$$\operatorname{tr}(D) - 2\|D\|_F\sqrt{\log\frac{2}{\beta}} \le u^T D u \le \operatorname{tr}(D) + 2\|D\|_F\sqrt{\log\frac{2}{\beta}} + 2\|D\|_2\log\frac{2}{\beta}.$$

*Proof of Lemma 4.15.* The privacy loss function is

$$
\begin{aligned}
f(w) &= \left|\log\frac{\Pr_{W \sim \mathcal{N}(0,\Sigma_x)}[W = w]}{\Pr_{W \sim \mathcal{N}(0,\Sigma_y)}[W = w]}\right| \\
&= \left|\log\left(\frac{|\Sigma_y|^{1/2}}{|\Sigma_x|^{1/2}}\exp\left\{-\frac{1}{2}w^T\Sigma_x^{-1}w + \frac{1}{2}w^T\Sigma_y^{-1}w\right\}\right)\right| \\
&\le \frac{1}{2}\left|w^T\left(\Sigma_y^{-1} - \Sigma_x^{-1}\right)w\right| + \frac{1}{2}\left|\log\frac{|\Sigma_y|}{|\Sigma_x|}\right|.
\end{aligned}
\tag{33}
$$

It suffices to prove that $\Pr_{w \sim \mathcal{N}(0,\Sigma_x)}[f(w) > \varepsilon] \le \delta$ and $\Pr_{w \sim \mathcal{N}(0,\Sigma_y)}[f(w) > \varepsilon] \le \delta$.

By Lemma 4.12, setting $\rho = 2k\lambda\left(\frac{1}{n-2k\lambda} + \frac{1}{n}\right)$, we have:

$$\|\Sigma_x^{-1/2}\Sigma_y\Sigma_x^{-1/2} - \mathbb{I}\|_{\operatorname{tr}} \le \rho$$
$$\|\Sigma_y^{-1/2}\Sigma_x\Sigma_y^{-1/2} - \mathbb{I}\|_{\operatorname{tr}} \le \rho$$

Now we will use the following facts, whose proofs follow by standard properties of the trace and are omitted.

**Fact 4.17.** *Let $A, B$ be two symmetric positive definite matrices. Then the following equalities hold.*

$$\operatorname{tr}\left(A^{-1/2}BA^{-1/2} - \mathbb{I}\right) = \operatorname{tr}\left(B^{1/2}A^{-1}B^{1/2} - \mathbb{I}\right) \quad and$$
$$\|A^{-1/2}BA^{-1/2} - \mathbb{I}\|_F = \|B^{1/2}A^{-1}B^{1/2} - \mathbb{I}\|_F.$$

**Fact 4.18.** *Let $|C|$ denote the determinant of a matrix $C$. Then $\operatorname{tr}(\mathbb{I} - C^{-1}) \le \log|C| \le \operatorname{tr}(C - \mathbb{I})$.*

By Fact 4.17 and since $\max\{|\operatorname{tr}(C)|, \|C\|_F\} \le \|C\|_{\operatorname{tr}}$ for any matrix $C$, this implies that

$$\max\left\{\left|\operatorname{tr}\left(\Sigma_y^{1/2}\Sigma_x^{-1}\Sigma_y^{1/2} - \mathbb{I}\right)\right|, \left\|\Sigma_y^{1/2}\Sigma_x^{-1}\Sigma_y^{1/2} - \mathbb{I}\right\|_F\right\} \le \rho \tag{34}$$

$$\max\left\{\left|\operatorname{tr}\left(\Sigma_x^{1/2}\Sigma_y^{-1}\Sigma_x^{1/2} - \mathbb{I}\right)\right|, \left\|\Sigma_x^{1/2}\Sigma_y^{-1}\Sigma_x^{1/2} - \mathbb{I}\right\|_F\right\} \le \rho \tag{35}$$

---

[5]For three distributions $P_1, P_2, P_3$, the definition of $(\varepsilon, \delta)$-indistinguishability tells us that if $P_1 \approx_{\varepsilon,\delta} P_2$ and $P_2 \approx_{\varepsilon,\delta} P_3$ then $P_1 \approx_{2\varepsilon,(1+e^\varepsilon)\delta} P_3$.

In addition, we observe that by applying Fact 4.18 once for $C = \Sigma_x^{-1}\Sigma_y$ and once for $C = \Sigma_y^{-1}\Sigma_x$ and using again the cyclic property of the trace, we can also bound the second term of Eq. (33) as

$$\left|\log\frac{|\Sigma_y|}{|\Sigma_x|}\right| \leq \rho. \tag{36}$$

We need a tail bound on $\left|w^T\left(\Sigma_y^{-1} - \Sigma_x^{-1}\right)w\right|$ under both distributions. Take $w \sim \mathcal{N}(0, \Sigma_x)$ or, alternatively, $u \sim \mathcal{N}(0, \mathbb{I})$ and $w = \Sigma_x^{1/2}u$. Using this, we have

$$
\begin{aligned}
\left|w^T\left(\Sigma_y^{-1} - \Sigma_x^{-1}\right)w\right| &= \left|(\Sigma_x^{1/2}u)^T\left(\Sigma_y^{-1} - \Sigma_x^{-1}\right)(\Sigma_x^{1/2}u)\right| \\
&= \left|u^T\left(\Sigma_x^{1/2}\Sigma_y^{-1}\Sigma_x^{1/2} - \mathbb{I}\right)u\right| \\
&= \left|u^T D u\right|,
\end{aligned}
\tag{37}
$$

defining $D$ as the "difference matrix."

Using the Hanson-Wright Inequality (Lemma 4.16), with probability at least $1 - \delta$,

$$|u^T D u| \leq |\operatorname{tr}(D)| + 2\|D\|_F\sqrt{\log(2/\delta)} + 2\|D\|_2\log(2/\delta).$$

It holds that $\|D\|_F \geq \|D\|_2$ for any matrix. So, with probability at least $1 - \delta$,

$$|u^T D u| \leq 5\log(2/\delta)\max\{|\operatorname{tr}(D)|, \|D\|_F\} \leq 5\log(2/\delta)\rho \tag{38}$$

where the latter holds by Eq. (34).

Combining Eq. (37), (38), and (36) in Eq. (33), with probability at least $1 - \delta$ under $w \sim \mathcal{N}(0, \Sigma_x)$,

$$f(w) \leq \frac{5}{2}\rho\log\frac{2}{\delta} + \frac{1}{2}\rho \leq 5\rho\log\frac{2}{\delta} = 10k\lambda\left(\frac{1}{n-2k\lambda} - \frac{1}{n}\right)\log\frac{2}{\delta} \leq \varepsilon.$$

Similarly, we bound the first term of Eq. (33) for $w \sim \mathcal{N}(0, \Sigma_y)$ using the same argument, where the "difference matrix" is now $D' = \Sigma_y^{1/2}\Sigma_x^{-1}\Sigma_y^{1/2} - \mathbb{I}$. $\qquad\square$

Our second lemma, about the indistinguishability of Gaussians with the same covariance and different means, follows from the analysis of the standard Gaussian mechanism and the application of our goodness assumption.

**Lemma 4.19.** *Suppose $x, y \in \mathcal{G}(\lambda)$ and $D_H(x,y) \leq k$, with $2k\lambda < n$. Set scaling parameter*

$$C^2 = \frac{32k^2}{\varepsilon^2 n^2} \cdot \frac{\lambda}{1 - 2k\lambda/n} \cdot \log\frac{1.25}{\delta}.$$

*Then $\mathcal{N}(\mu_y, C^2\Sigma_x) \approx_{\varepsilon,\delta} \mathcal{N}(\mu_x, C^2\Sigma_x)$.*

*Proof.* We have $\mathcal{N}(\mu_y, C^2\Sigma_x) \approx_{\varepsilon,\delta} \mathcal{N}(\mu_x, C^2\Sigma_x)$ iff $\mathcal{N}(\Sigma_x^{-1/2}(\mu_y - \mu_x), C^2\mathbb{I}) \approx_{\varepsilon,\delta} \mathcal{N}(0, C^2\mathbb{I})$, since translation and multiplication by an invertible matrix are bijections. By the standard analysis of the Gaussian mechanism (Lemma 2.7), if we can prove $\|\mu_y - \mu_x\|_{\Sigma_x} \leq \Delta_\mu$ and set $C \geq \Delta_\mu\varepsilon^{-1}\sqrt{2\log\frac{1.25}{\delta}}$, then this is $(\varepsilon, \delta)$-differentially private.

Let $S = \{i \in [n] : x_{i+2n} = y_{i+2n}\}$. We have

$$
\begin{aligned}
\left\|\mu_x - \mu_y\right\|_{\Sigma_x} &= \left\|\frac{1}{n}\sum_{i\in[n]}x_{i+2n} - \frac{1}{n}\sum_{i\in[n]}y_{i+2n}\right\|_{\Sigma_x} \\
&= \left\|\frac{1}{n}\sum_{i\in[n]\setminus S}x_{i+2n} - y_{i+2n}\right\|_{\Sigma_x} \\
&\leq \frac{1}{n}\sum_{i\in[n]\setminus S}\left\|x_{i+2n} - y_{i+2n}\right\|_{\Sigma_x}
\end{aligned}
$$

Pick any point $x_{j^*}$ for $j^* \in S$.

$$\|x_{i+2n} - y_{i+2n}\|_{\Sigma_x} \leq \|x_{i+2n} - x_{j^*}\|_{\Sigma_x} + \|y_{i+2n} - x_{j^*}\|_{\Sigma_x}.$$

By Lemma 4.10, the first term is at most $2\sqrt{\lambda}$. By Lemma 4.11, the second is at most $\frac{1}{\sqrt{1-2k\lambda/n}}\|y_{i+2n} - x_{j^*}\|_{\Sigma_y}$, the Mahalanobis distance under $\Sigma_y$. Applying Lemma 4.10 again, since $x_{j^*} \in y$, this is at most $\frac{2\sqrt{\lambda}}{\sqrt{1-2k\lambda/n}}$. $\quad\square$

## Acknowledgements

We thank the anonymous NeurIPS reviewers for useful suggestions on the presentation of this manuscript. Gavin Brown and Adam Smith were supported in part by NSF award CCF-1763786 as well as a Sloan Foundation research award. Marco Gaboardi was supported in part by NSF award CCF-2040222, CCF-1718220, CNS-1565365, and CNS-2040215. Jonathan Ullman and Lydia Zakynthinou were supported by NSF grants CCF-1750640, CNS-1816028, and CNS-1916020. Lydia Zakynthinou was also supported by a Facebook Fellowship.

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

# A   Linear Algebra Background

In this section, we present a short introduction to facts from linear algebra, which we use often in our proofs.

For any matrix $A$, we will denote by $\lambda_i(A)$ and by $\sigma_i(A)$ the $i$-th largest eigenvalue and singular value of $A$, respectively. We only consider real matrices $A \in \mathbb{R}^{d \times d}$.

**Proposition A.1** (Properties of eigenvalues and singular values). *For any real matrix $A \in \mathbb{R}^{d \times d}$,*

$$\sigma_i^2(A) = \lambda_i(A^T A) \quad and \quad \sum_{i=1}^{k} |\lambda_i(A)| \le \sum_{i=1}^{k} \sigma_i(A) \ \ \forall k \le d.$$

*If $A$ is symmetric, then $|\lambda_i(A)| = \sigma_i(A)$ for all $i \in [d]$.*

**Definition A.2** (Matrix norms). Let $A \in \mathbb{R}^{d \times d}$ be any square matrix.

- The *trace norm* (or *nuclear norm*) of $A$ is $\|A\|_{\mathrm{tr}} = \mathrm{tr}\left(\sqrt{A^T A}\right) = \sum_{i=1}^{d} \sigma_i(A)$.

- The *Frobenius norm* of $A$ is $\|A\|_F = \sqrt{\sum_{j=1}^{d} \sum_{i=1}^{d} |a_{i,j}|^2} = \sqrt{\mathrm{tr}(A^T A)} = \sqrt{\sum_{i=1}^{d} \sigma_i^2(A)}$.

- The *spectral norm* of $A$ is $\|A\|_2 = \sup\{\|Ax\|_2 : x \in \mathbb{R}^d \text{ s.t. } \|x\|_2 = 1\} = \sqrt{\lambda_1(A^T A)} = \sigma_1(A)$.

By straightforward comparison of the definitions above, $\|A\|_2 \le \|A\|_F \le \|A\|_{\mathrm{tr}}$.
We measure the error of our estimator using the Mahalanobis distance.

**Definition A.3** (Mahalanobis distance). For any vector $v \in \mathbb{R}^d$ and any positive definite matrix $\Sigma$, the *Mahalanobis distance* of $v$ with respect to $\Sigma$ is defined as $\|v\|_\Sigma = \|\Sigma^{-1/2} v\|_2$.

Also note that we can write $\|v\|_\Sigma^2 = v^T \Sigma^{-1} v$.

**Proposition A.4.** *For any vectors $u$ and $v$, $\|uv^T\|_2 \le u^T v$. Furthermore, for any vector $v$, $\mathrm{tr}(vv^T) = \|vv^T\|_{\mathrm{tr}} = \|vv^T\|_2 = v^T v$.*

# B  Subgaussian Data

In this section, we extend our analysis of Algorithm 2 to show that its guarantees hold even if the data are subgaussian, instead of the stricter Gaussian assumption used previously.

## B.1  Useful Facts and Definitions

To formalize our setting, let us first state useful definitions and concentration inequalities for subgaussian distributions and data sets.

**Definition B.1** (Subgaussian random variable). A random variable $v \in \mathbb{R}$ with mean $\mathbb{E}[v] = \mu$ is $\sigma^2$-*subgaussian* if $\mathbb{E}[e^{\lambda(v-\mu)}] \le e^{\lambda^2 \sigma^2/2}$ for all $\lambda \in \mathbb{R}$.

In this case, we write $v \sim \text{subG}(\sigma^2)$, slightly abusing notation, since $\text{subG}(\sigma^2)$ represents a family rather than a single distribution. We also write $P \in \text{subG}(\sigma^2)$ if $P$ is a subgaussian distribution with parameter $\sigma^2$. In the $d$-dimensional case, we define subgaussian distributions as follows. We write $P_{\mu,\Sigma}$ to denote that the mean and covariance of the distribution are $\mu$ and $\Sigma$, respectively.

**Definition B.2** (Subgaussian distribution). Let $P_{\mu,\Sigma}$ be a distribution over $\mathbb{R}^d$ with mean $\mu$ and covariance $\Sigma > 0$. For a constant $c > 0$, we say that $P_{\mu,\Sigma}$ is subgaussian with parameter $c\Sigma$, if for $v \sim P_{\mu,\Sigma}$ and all unit vectors $u$, the distribution of $v^T u$ is $c(u^T \Sigma u)$-subgaussian (as in Definition B.1). That is, for all $u \in \mathbb{R}^d$ such that $\|u\|_2 = 1$,
$$\mathbb{E}_{v \sim P_{\mu,\Sigma}}[e^{\lambda u^T(v-\mu)}] \le e^{c\lambda^2 (u^T \Sigma u)/2} \text{ for all } \lambda \in \mathbb{R}.$$

We write $P_{\mu,\Sigma} \in \text{subG}(c\Sigma)$. Intuitively, a distribution is subgaussian if it concentrates at least as well as a Gaussian along every univariate projection. We note that although Definition B.2 above is not the "textbook" definition (compare with [74], for example), is has appeared often in the relevant literature (see e.g. [59, 26]).

Concentration inequalities, analogous to those in Lemma 4.5 for Gaussian data, hold for subgaussian data:

**Lemma B.3** (Extension of Lemma 4.5). *Let $u_i$ be i.i.d. $d$-dimensional samples for $i \in [n]$ drawn from a distribution $P_{0,\mathbb{I}}$ with mean $\mu = 0$ and covariance $\Sigma = \mathbb{I}$, such that $P_{0,\mathbb{I}} \in \text{subG}(c\mathbb{I})$ for some constant $c > 0$. Define the estimator $\hat{\Sigma} = \frac{1}{n} \sum_{i=1}^n u_i u_i^T$. For every $\beta > 0$, the following conditions hold with probability $1 - \beta$:*

$$\left(1 - O\left(\sqrt{\frac{d + \log(1/\beta)}{n}}\right)\right) \cdot \mathbb{I} \le \hat{\Sigma} \le \left(1 + O\left(\sqrt{\frac{d + \log(1/\beta)}{n}}\right)\right) \cdot \mathbb{I} \tag{39}$$

$$\forall i \in [n] \quad \|u_i\|_2^2 \le O(d \log(n/\beta)) \tag{40}$$

Observe that if $P_{0,\Sigma}$, with mean $\mu = 0$ and covariance $\Sigma > 0$, is a $c\Sigma$-subgaussian distribution for some $c > 0$, then for any $x \sim P_{0,\Sigma}$ there exists $u = \Sigma^{-1/2} x$ with $u \sim P_{0,\mathbb{I}}$, where $P_{0,\mathbb{I}} \in \text{subG}(c\mathbb{I})$ with mean $\mu = 0$ and covariance $\mathbb{I}$. Using this observation and the lemma above, we have the following more general concentration facts for subgaussian distributions.

**Lemma B.4** (Extension of Lemma 4.6). *Let $x_i$ $\forall i \in [n]$ be i.i.d. $d$-dimensional samples from $P_{0,\Sigma}$, with mean $\mu = 0$ and covariance $\Sigma$, such that $P_{0,\Sigma} \in \text{subG}(c\Sigma)$ for some constant $c > 0$. Define the estimator $\hat{\Sigma}_x = \frac{1}{n} \sum_{i=1}^n x_i x_i^T$. For every $\beta > 0$, the following conditions hold with probability $1 - \beta$:*

$$\left(1 - O\left(\sqrt{\frac{d + \log(1/\beta)}{n}}\right)\right) \cdot \Sigma \le \hat{\Sigma}_x \le \left(1 + O\left(\sqrt{\frac{d + \log(1/\beta)}{n}}\right)\right) \cdot \Sigma \tag{41}$$

$$\forall i \in [n] \quad \|x_i\|_2^2 \le O(\lambda_1(\Sigma) \cdot d \log(n/\beta)) \tag{42}$$

We will also use the following standard concentration inequality for the empirical mean of a subgaussian data set.

**Lemma B.5** (Mean of Subgaussian Vectors). *Let $u_i$ be i.i.d. d-dimensional samples for $i \in [n]$ drawn from a distribution $P_{0,\mathbb{I}}$ with mean $\mu = 0$ and covariance $\Sigma = \mathbb{I}$, such that $P_{0,\mathbb{I}} \in \mathrm{subG}(c\mathbb{I})$ for some constant $c > 0$. For any $\beta > 0$, with probability at least $1 - \beta$,*

$$\left\| \frac{1}{n} \sum_{i=1}^{n} u_i \right\|_2 = O\left( \sqrt{\frac{d + \log(1/\beta)}{n}} \right).$$

## B.2 Guarantees of Algorithm 2 for Subgaussian Data

**Theorem B.6** (Privacy and Accuracy of the Empirically Rescaled Gaussian Mechanism for Subgaussian Data). *For any $\varepsilon > 0$, $0 < \delta < 1$, Algorithm 2 is $(3\varepsilon, e^\varepsilon(1 + e^\varepsilon)\delta)$-differentially private. There exists an absolute constant $C$ such that, for any $0 < \alpha, \beta, \varepsilon, \delta < 1$, mean $\mu$, and positive definite $\Sigma$, if $x \sim P_{\mu,\Sigma}^{\otimes n}$, where $P_{\mu,\Sigma} \in \mathrm{subG}(c\Sigma)$ for some constant $c > 0$, and*

$$n \geq C\left( \frac{d}{\alpha^2} \log \frac{1}{\beta} + \frac{d}{\alpha\varepsilon^2} \log^3 \frac{1}{\delta\beta} \cdot \log \frac{d \log(1/\delta\beta)}{\alpha\varepsilon} \right), \tag{43}$$

*then with probability at least $1 - 3\beta$, Algorithm 2 returns $\mathcal{A}_{\varepsilon,\delta,\beta}^G(x) = \hat{\mu}$ such that $\|\hat{\mu} - \mu\|_\Sigma \leq \alpha$.*

*Proof Sketch.* Notice first that the privacy guarantees of Algorithm 2 do not depend on the assumption that the data distribution is Gaussian. Therefore, the privacy analysis of Section 4.2 remains the same. The accuracy analysis follows the same steps, with two modifications: we need to prove that with high probability subgaussian data fall into the good set $\mathcal{G}(\lambda)$ with the same parameter $\lambda$ (Lemma B.7 below – an extension of Lemma 4.7) and that with high probability, for the given sample complexity, the error is upper bounded by $\alpha$ (Lemma B.8 below – an extension of Lemma 4.8). Plugging the new lemmas into the accuracy analysis of Algorithm 2 completes the proof of the theorem. □

**Lemma B.7** (Extension of Lemma 4.7). *Suppose that $x \sim P_{\mu,\Sigma}^{\otimes 3n}$, where $P_{\mu,\Sigma}$ is a distribution with mean $\mu$, covariance $\Sigma$, such that $P_{\mu,\Sigma} \in \mathrm{subG}(c\Sigma)$ for some constant $c > 0$. Let $n = \Omega(d + \log(1/\beta))$. There exists a $\lambda = O(d \log(n/\beta))$ such that, with probability at least $1 - \beta$ we have $x \in \mathcal{G}(\lambda)$.*

The proof of the lemma is omitted since it follows the same steps as the proof of Lemma 4.7, except that the use of the concentration properties of Gaussians stated in Lemma 4.6 is replaced by the use of the concentration properties of subgaussians stated in Lemma B.4.

**Lemma B.8** (Extension of Lemma 4.8). *Suppose that $x \sim P_{\mu,\Sigma}^{\otimes 3n}$, where $P_{\mu,\Sigma}$ is a distribution with mean $\mu$ and covariance $\Sigma$, such that $P_{\mu,\Sigma} \in \mathrm{subG}(c\Sigma)$ for some constant $c > 0$. Let $n = \Omega(\max\{(d + \log(1/\beta)), k\lambda\})$, where parameters $k, \lambda$ are set as in Algorithm 2. Then with probability at least $1 - \beta$, for $\hat{\mu} \sim \mathcal{N}(\mu_x, C^2\Sigma_x)$,*

$$\|\hat{\mu} - \mu\|_\Sigma = O\left( \sqrt{\frac{d}{n} \cdot \log \frac{1}{\beta}} + \frac{d}{\varepsilon^2 n} \log^2 \frac{1}{\delta\beta} \cdot \sqrt{\log \frac{n}{\beta}} \right).$$

*Proof Sketch.* By the triangle inequality, we have that

$$\|\hat{\mu} - \mu\|_\Sigma \leq \|\mu - \mu_x\|_\Sigma + \|\mu_x - \hat{\mu}\|_\Sigma. \tag{44}$$

The first term can be written as $\|\mu - \mu_x\|_\Sigma = \|\frac{1}{n} \sum_{i=1}^{n} \Sigma^{-1/2}(x_{i+2n} - \mu)\|_2 = \|\frac{1}{n} \sum_{j=1}^{n} u_i\|_2$, where $u_i \sim P_{0,\mathbb{I}} \ \forall i \in [n]$ are subgaussian vectors with mean 0, covariance $\mathbb{I}$, and $P_{0,\mathbb{I}} \in \mathrm{subG}(c\mathbb{I})$. By Lemma B.5, with probability at least $1 - \beta/2$, it holds that

$$\|\mu - \mu_x\|_\Sigma = O\left( \sqrt{\frac{d + \log \frac{1}{\beta}}{n}} \right). \tag{45}$$

The second term is bounded via the same steps as in the proof of Lemma 4.8, as the distribution of $\hat{\mu}$ has not changed (it is still drawn from a Gaussian with mean $\mu_x$ and covariance $C^2\Sigma_x$). This yields Eq. (28). Combining the latter with Eq. (45) via a union bound and following the same calculations as in the proof of Lemma 4.8 will complete the proof. □

# C  Finite Implementations of Our Algorithms

## C.1  Technical Tools

In this section we will give differentially private algorithms for estimating the largest and smallest eigenvalues of the covariance matrix $\Sigma$, denoted $\lambda_1$ and $\lambda_d$, as well as an enclosing box for the data $[-R, R]^d$. We start by describing a building block for both of these algorithms: the Stable Histogram of [11].

---

**Algorithm 3** StableHistogram$_{\varepsilon,\delta}(\{z_i\}, \{B_b\})$, from [11]

---

**Require:** Items $z_1, \ldots, z_n \in \mathcal{U}$. Bins $\{B_b\}_{b \in \mathbb{Z}}$. Privacy parameters $\varepsilon, \delta > 0$.
 1: **for** $b \in \mathbb{Z}$ **do**
 2:     $c_b \leftarrow |\{i : z_i \in B_b\}|$
 3: **for** $b$ with $c_b > 0$ **do**
 4:     $\tilde{c}_b \leftarrow c_b + \mathrm{Lap}(2/\varepsilon)$
 5: $\tau \leftarrow 1 + \frac{2 \log(1/\delta)}{\varepsilon}$
 6: **return** $\{(b, \tilde{c}_b) : b \in \mathbb{Z} \text{ and } \tilde{c}_b \geq \tau\}$.

---

We now state the guarantees of Stable Histogram, in a form which will be useful for our next steps.

**Lemma C.1** (Stable Histogram Guarantees). *StableHistogram$_{\varepsilon,\delta}$ is $(\varepsilon, \delta)$-differentially private. Let $z_1, \ldots, z_n$ be drawn i.i.d. from distribution $P$. Suppose that there exists $b \in \mathbb{Z}$ and a constant $\beta' < \frac{1}{4}$, such that $\Pr[z_i \notin B_{b-1} \cup B_b \cup B_{b+1}] \leq \beta'$ for any fixed $i \in [n]$. Let $b^* = \arg\max_b \tilde{c}_b$, where $\{(b, \tilde{c}_b)\} = \text{StableHistogram}_{\varepsilon,\delta}(z_1, \ldots, z_n)$. There exists a constant $C > 0$ such that, for all $0 < \varepsilon, \beta, \delta < 1$, if*

$$n \geq \frac{C}{\varepsilon} \log \frac{1}{\beta\delta},$$

*then with probability at least $1 - \beta$, $b^* \in \{b - 1, b, b + 1\}$.*

A proof of the privacy guarantee can be found in [72, Theorem 3.5]. A slightly larger (by logarithmic factors) sample complexity guarantee than the one stated above can be proven in a straightforward way, using intermediate results of the proof of [53, Lemma 2.3]. However, we provide a proof of the tighter sample complexity bound stated here, for completeness.

*Proof.* Note that the $z_i$ are independent. There are at most 3 "good bins" $b-1, b, b+1$ and $\Pr[z_i \in \text{good bins}] \geq 1 - \beta'$. There must be a heaviest good bin, which we call the "best bin" $b_1$, such that $\Pr[z_i \in B_{b_1}] \geq \frac{1-\beta'}{3}$. The bad bins collectively satisfy $\Pr[z_i \in \text{bad bins}] \leq \beta'$.

Let random variable $X_{\text{best}}$ be the number of items that fall into the best bin and $X_{\text{bad}}$ be the number of items that fall into *any* of the bad bins. Since both these random variables are sums of independent $0 - 1$ trials, we apply Chernoff bounds [62, Theorems 4.4, 4.5]. We have $\mathbb{E}[X_{\text{best}}] \geq \frac{1-\beta'}{3}n$ and $\mathbb{E}[X_{\text{bad}}] \leq \beta'n$. Introduce constants $\gamma_1, \gamma_2 > 0$ so that

$$\beta' + \gamma_1 < \frac{1 - \beta'}{3} - \gamma_2.$$

Then we can bound

$$\Pr[X_{\text{bad}} \geq n(\beta' + \gamma_1)] = \Pr[X_{\text{bad}} \geq \beta'n(1 + \gamma_1/\beta')] \leq \exp\left\{-\frac{\gamma_1^2 n}{3\beta'}\right\} \tag{46}$$

and

$$\Pr\left[X_{\text{best}} \leq m\left(\frac{1 - \beta'}{3} - \gamma_2\right)\right] = \Pr\left[X_{\text{best}} \geq \frac{(1 - \beta')n}{3}(1 - 3\gamma_2/(1 - \beta'))\right] \tag{47}$$

$$\leq \exp\left\{-\frac{3\gamma_2^2 n}{2(1 - \beta')}\right\} \tag{48}$$

Conditioned on the best bin $b_1$ receiving sufficiently many items, we need to ensure that its noisy count is (i) not suppressed and (ii) higher than that of any bad bin. Introduce a third constant $\gamma_3 > 0$ and define random variable $Z \sim \text{Lap}(1/\varepsilon)$.

$$\Pr\left[X_{\text{best}} + Z \leq m\left(\frac{1-\beta'}{3} - \gamma_3\right)\Big| X_{\text{best}} \geq n\left(\frac{1-\beta'}{3} - \gamma_2\right)\right] \leq \Pr[Z \leq n(\gamma_2 - \gamma_3)] \tag{49}$$

$$= \Pr[Z \geq n(\gamma_3 - \gamma_2)] \qquad \text{(flip the signs)}$$

$$\leq \frac{1}{2}\exp\{-\varepsilon n(\gamma_3 - \gamma_2)\}. \tag{50}$$

To avoid suppression, we require $n\left(\frac{1-\beta'}{3} - \gamma_3\right) > 1 + \frac{\log(1/\delta)}{\varepsilon}$. Since $\beta'$ and $\gamma_3$ are constants, this means $n = \Omega(\log(1/\delta)/\varepsilon)$. Similarly, for any single bad bin we must control

$$\Pr[Z \geq n(\gamma_3 - \gamma_1)] \leq \frac{1}{2}\exp\{-\varepsilon n(\gamma_3 - \gamma_1)\}. \tag{51}$$

We do not mind if the bad bins get suppressed. We will take a union bound over the (no more than) $n$ bad bins.

We want to bound the probability that $b^* = \arg\max_b \tilde{c}_b$ belongs in any of the bad bins. Putting the pieces together, we need to apply the union bound over the following bad events: (i) the best bin fails to receive enough items, (ii) the bad bins (collectively) receive too many items, (iii) too much (negative) noise is added to the best bin, and (iv) too much (positive) noise is added to *any* of the bad bins that received an item.

$$\Pr[b^* \notin \{b - 1, b, b + 1\}] \leq e^{-\frac{\gamma_1^2 n}{3\beta'}} + e^{-\frac{3\gamma_2^2 n}{2(1-\beta')}} + \frac{1}{2}e^{-\varepsilon n(\gamma_3 - \gamma_2)} + \frac{n}{2}e^{-\varepsilon n(\gamma_3 - \gamma_1)}. \tag{52}$$

With $\beta' < \frac{1}{4}$, we can take $\gamma_1, \gamma_2$, and $\gamma_3$ to be constants. And, if we set $\gamma_3 - \gamma_1 > \gamma_3 - \gamma_2$ (i.e. $\gamma_2 > \gamma_1$) then asymptotically we don't have to pay for the union bound over bad bins and we get $\Pr[b^* \notin \{b - 1, b, b + 1\}] = O(e^{-c\varepsilon n})$ for some constant $c$. For this to be less than $\beta$, we need $n = \Omega(\log(1/\beta)/\varepsilon)$. We also had $n = \Omega(\log(1/\delta)/\varepsilon)$, so we require that $n = \Omega(\log(1/\beta\delta)/\varepsilon)$. $\qquad \square$

### C.1.1 Private Eigenvalue Estimation

We now give an $(\varepsilon, \delta)$-differentially private algorithm, based on the well-known *Sample-and-Aggregate* framework [63]. We denote by $\lambda_k(A)$ the $k$-th largest eigenvalue of matrix $A$.

---

**Algorithm 4** Private Eigenvalue Estimation via Sample and Aggregate: $\text{Eigen}_{\varepsilon,\delta,\beta}(x, k)$

---

**Require:** Data set $x = (x_1, \ldots, x_n)^T \in \mathbb{R}^{n \times d}$. Index $k \in [d]$. Privacy parameters $\varepsilon, \delta > 0$. Failure probability $\beta > 0$.

1: Initialize $m \leftarrow \Omega(\log(1/\delta\beta)/\varepsilon)$.
2: **for** $i \in [m]$ **do**
3: $\quad \hat{\Sigma} \leftarrow \frac{m}{n}\sum_{j=1}^{n/m}(x_{\frac{n}{m}(i-1)+j})(x_{\frac{n}{m}(i-1)+j})^T$ $\qquad\qquad\qquad$ ▷ empirical covariance of block $i \in [m]$
4: $\quad \hat{\lambda}_k^{(i)} \leftarrow \lambda_k(\hat{\Sigma})$
5: $\quad z_i \leftarrow \text{Round}(\hat{\lambda}_k^{(i)})$, rounded down to the nearest $2^q$ for $q \in \mathbb{Z}$
6: $\{(b, \tilde{c}_b)\} \leftarrow \text{StableHistogram}_{\varepsilon,\delta}(\{z_i\}, \{B_b\})$ for bins $B_b = [2^b, 2^{b+1})$.
7: $b^* \leftarrow \arg\max_b \tilde{c}_b$
8: **return** $2^{b^*}$

---

**Lemma C.2** (Private Estimate of Smallest/Largest Eigenvalue). *Algorithm 4 is $(\varepsilon, \delta)$-differentially private. Suppose $x$ is drawn i.i.d. from a distribution $P_{0,\Sigma}$ with mean 0, covariance $\Sigma$, and that $P \in \text{subG}(c\Sigma)$ for constant $c > 0$. There exists a constant $C > 0$ such that for any $0 < \varepsilon, \delta, \beta < 1$, $k \in [d]$, if*

$$n \geq C\frac{d}{\varepsilon}\log\left(\frac{1}{\delta\beta}\right), \tag{53}$$

*with probability at least $1 - \beta$, Algorithm 4 returns an estimate $\hat{\lambda}_k$ such that $\frac{1}{4}\lambda_k(\Sigma) \le \hat{\lambda}_k \le 4\lambda_k(\Sigma)$.*

*Proof.* The privacy guarantee is inherited by the guarantee of the Stable Histogram (Lemma C.1). From Lemma B.4, if $n/m$ (the number of samples in each block) is $\Omega(d)$ then with probability $1 - \beta'$ for a constant $\beta' < \frac{1}{4}$, we get $\frac{1}{2}\lambda_k(\Sigma) \le \hat{\lambda}_k^{(i)} \le 2\lambda_k(\Sigma)$ for any fixed $i \in [m]$. Let $\lambda_k(\Sigma) \in B_b$, that is, $B_b$ is the bin that the rounding of the true eigenvalue would fall into. By the previous guarantee, if $n/m = \Omega(d)$, we can write for any fixed $i \in [m]$,

$$\Pr[z_i \notin B_{b-1} \cup B_b \cup B_{b+1}] \le \beta'.$$

The hypotheses of Lemma C.1 are then satisfied, and it follows that, there exists a constant $C > 0$ such that if $m \ge \frac{C}{\varepsilon} \log \frac{1}{\beta\delta}$, then

$$\Pr[b^* \notin \{b - 1, b, b + 1\}] \le \beta.$$

Combining the conditions on the number of samples, if

$$n = \Omega(dm) = \Omega\left(\frac{d}{\varepsilon} \log \frac{1}{\beta\delta}\right),$$

then with probability at least $1 - \beta$, Algorithm 4 returns $\hat{\lambda}_k \in B_{b-1} \cup B_b \cup B_{b+1}$. Equivalently, with probability $1 - \beta$, it returns $\frac{1}{4}\lambda_k(\Sigma) \le \hat{\lambda}_k \le 4\lambda_k(\Sigma)$. □

### C.1.2 Private Range Estimation

---

**Algorithm 5** Private Range Estimation, from [53]: $\text{Range}_{\varepsilon,\delta,\beta}(x, \sigma^2)$

---

**Require:** Data set $x = (x_1, \ldots, x_n)^T \in \mathbb{R}^{n \times d}$. Privacy parameters $\varepsilon, \delta > 0$. Failure probability $\beta > 0$. Variance upper bound $\sigma^2$.

1: **for** $j \in [d]$ **do**
2:      $z_i \leftarrow x_{i,j}$ for all $i \in [n]$              ▷ Choose the $j$-th coordinate from each sample $i \in [n]$
3:      $\{(b, \tilde{c}_b)\} \leftarrow \text{StableHistogram}_{\frac{\varepsilon}{d}, \frac{\delta}{d}}(\{z_i\}, \{B_b\})$ for bins $B_b = [3\sigma b, 3\sigma(b + 1))$.
4:      $b_j^* \leftarrow \arg\max_b \tilde{c}_b$
5:      $X_{min}^j \leftarrow 3\sigma b_j^* - 11\sigma \log \frac{nd}{\beta}$
6:      $X_{max}^j \leftarrow 3\sigma b_j^* + 11\sigma \log \frac{nd}{\beta}$
7: **return** $\{(X_{min}^j, X_{max}^j)\}_{j \in [d]}$

---

The algorithm above follows a standard approach for range estimation of univariate Gaussian data sets, applied $d$ times—one for each coordinate $i \in [d]$. In particular, Karwa and Vadhan [53] prove the guarantees of the algorithm for Gaussian data sets. Liu et al. [58] also prove its guarantees for subgaussian data sets with identity covariance and corruptions. Since neither of the two covers our exact case, we provide a modification of their proofs below.

**Lemma C.3** (Private Range Estimate). *Algorithm 5 is $(\varepsilon, \delta)$-differentially private. Suppose $x$ is drawn i.i.d. from a distribution $P_{\mu,\Sigma}$ with mean $\mu$ and covariance $\Sigma$, and that for every coordinate $j \in [d]$ if $z \sim P_{\mu,\Sigma}$ then $z_j$ is $\sigma^2$-subgaussian. There exists a constant $C > 0$ such that for any $0 < \varepsilon, \delta, \beta < 1$, if*

$$n \ge C\frac{d}{\varepsilon} \log\left(\frac{d}{\delta\beta}\right), \tag{54}$$

*with probability at least $1 - 2\beta$, Algorithm 5 returns an estimate $\{(X_{min}^j, X_{max}^j)\}_{j \in [d]}$ such that for all $i \in [n]$, $j \in [d]$, $x_{i,j} \in [X_{min}^j, X_{max}^j]$.*

*Proof.* The privacy guarantee is inherited by the Stable Histogram algorithm, via composition (Lemma 2.3).[6] By an equivalent definition of $\sigma^2$-subgaussian random variables, we have that for all $j \in [d]$,

$$\Pr[|x_{i,j} - \mu_j| > t] \leq 2e^{-t^2/2\sigma^2}. \tag{55}$$

Setting $t = 3\sigma$, we have that $\Pr[|x_{i,j} - \mu_j| > 3\sigma] \leq \beta'$, for $\beta' = 0.03 < 1/4$. Suppose $\mu_j \in B_b$, for some bin $b$. Then, we have that $\Pr[z_i \notin B_{b-1} \cup B_b \cup B_{b+1}] \leq \beta'$.

Therefore, the hypothesis of Lemma C.1 is satisfied and so, if $n = \Omega(\frac{d}{\varepsilon} \log \frac{d}{\delta\beta})$, with probability $1 - \beta/d$, we return $b_j^*$ such that $b_j^* \in \{b-1, b, b+1\}$. Equivalently, with probability $1 - \beta$, for all $j \in [d]$ simultaneously,

$$|\mu_j - 3\sigma b_j^*| \leq 9\sigma. \tag{56}$$

By the bound on subgaussian tails of Eq. (55) and a union bound, with probability $1 - \beta$, for all $i \in [n], j \in [d]$,

$$|x_{i,j} - \mu_j| \leq \sqrt{2\sigma^2 \log \frac{nd}{\beta}} \leq 2\sigma \log \frac{nd}{\beta} \tag{57}$$

Combining Eq. 57 and 56, with probability at least $1 - 2\beta$, for all $i \in [n], j \in [d]$,

$$x_{i,j} \in [3\sigma b_j^* - 11\sigma \log \frac{nd}{\beta}, 3\sigma b_j^* + 11\sigma \log \frac{nd}{\beta}].$$

$\square$

## C.2 A Finite Implementation of Algorithm 1

We modify Algorithm 1 (using our private eigenvalue and bounding-box estimates) by running the original algorithm with the data space $\mathbb{R}^d$ replaced by a finite grid of points $Q_{\alpha'}$. For simplicity, we work with data sets of size $2n$.

---

**Algorithm 6** Finite Implementation of $\mathcal{A}_{\varepsilon,\delta,t}^E(x)$

---

**Require:** Data set $x = (x_1, \ldots, x_{2n})^T \in \mathbb{R}^{2n \times d}$. Privacy parameters: $\varepsilon, \delta > 0$. Accuracy parameters: $\alpha, \beta > 0$.

**Stage 1: Range estimates**
1: Construct data set $u \in \mathbb{R}^{n \times d}$ where $u_i = (x_i - x_{i+n})/\sqrt{2}$, $i \in [n]$.
2: $\hat{\lambda}_1 \leftarrow \text{Eigen}_{\varepsilon,\delta,\beta}(u, 1)$                        ▷ private estimate of largest eigenvalue
3: $\hat{\lambda}_d \leftarrow \text{Eigen}_{\varepsilon,\delta,\beta}(u, d)$                       ▷ private estimate of smallest eigenvalue
4: $\sigma^2 \leftarrow 4\hat{\lambda}_1$                                 ▷ upper bound on variance in every direction
5: $\{X_{\min}^j, X_{\max}^j\}_{j \in [d]} \leftarrow \text{Range}_{\varepsilon,\delta,\beta}(x, \sigma^2)$
6: Set

$$\alpha' \leftarrow O\left(\frac{\alpha\sqrt{\hat{\lambda}_d}}{d}\right). \tag{58}$$

7: $R \leftarrow \alpha' + \max_j \max\{|X_{\max}^j|, |X_{\min}^j|\}$

**Stage 2: Discretize**
8: $Q_{\alpha'} \leftarrow \alpha'$-fine grid over $[-R, R]^d$.
9: For all $i \in [n]$, let $x_i^\Delta = \arg\min_{p \in Q_{\alpha'}} \|p - x_i\|_1$.

**Stage 3: Run the algorithm**
10: Run $\mathcal{A}_{\varepsilon,\delta,t}^E(x^\Delta)$.

---

[6]Note that by using advanced composition [36], we could have set the privacy parameter of StableHistogram to $\approx \varepsilon/\sqrt{d}$ but since this is not the sample complexity bottleneck, we did not.

**Privacy.** Since the discretization process doesn't affect the privacy analysis of Algorithm 1, the overall privacy follows from composition and the privacy analyses in C.1.

**Computation.** Tukey depth can be computed in time $\tilde{O}(n^d)$ [57]. Since we can run the restricted exponential mechanism over the grid $Q_{\alpha'}$, it remains to describe how, given $x^{\Delta}$, we can compute the distance to the set $\text{UNSAFE}_{(\varepsilon,\delta,t)}$. First note that, given two data sets $y, y' \subset Q_{\alpha'}^n$, we can check whether $\mathcal{M}_{\varepsilon,t}(y) \approx_{\varepsilon,\delta} \mathcal{M}_{\varepsilon,t}(y')$ by computing the distributions explicitly. Thus, by iterating over all neighbors of any data set $y$, we can check if $y \in \text{UNSAFE}_{(\varepsilon,\delta,t)}$. With this, computing the distance to $\text{UNSAFE}_{(\varepsilon,\delta,t)}$ requires iterating over all data sets in $Q_{\alpha'}^n$, which are at most $\left(\frac{2R}{\alpha'}\right)^{dn} = \tilde{O}\left(\frac{d(\|\mu\|_{\infty}/\sqrt{\lambda_d}+\sqrt{\kappa})}{\alpha}\right)^{dn}$, where $\kappa = \lambda_1/\lambda_d$ is the condition number of the covariance matrix $\Sigma$.

**Accuracy.** It remains to show that this algorithm provides an accurate estimate of $\mu$ when the data is Gaussian. We will show that the (old) error from uniform convergence and the (new) error from discretization can be grouped together, and that the $\alpha'$ we pick results in negligible error from discretization.

Fix $Q_{\alpha'}$ and let $P_{\Delta}$ be the distribution generated by snapping samples from the Gaussian $\mathcal{N}(\mu, \Sigma)$ to that grid. Since our uniform convergence argument holds for any distribution, with probability $1 - \beta$ over the choice of $x$ we have, for all $y$ within our bounding box, that $|T_x(y) - T_{P_{\Delta}}(y)| \leq \alpha_1$, using the same "typicality" parameter as in the main argument. We now relate $T_P(y)$ to $T_{P_{\Delta}}(y)$ for all $y$ within the bounding box.

**Lemma C.4.** *Let $Q_{\alpha'}$ be an $\alpha'$-fine grid over $[-R, R]^d$. Let $P = \mathcal{N}(\mu, \Sigma)$ be any Gaussian and let $P_{\Delta}$ be the distribution resulting from drawing from $P$ and then discretizing according to $Q_{\alpha'}$. Assume $\hat{\lambda}_d \leq \frac{1}{4}\lambda_d(\Sigma)$. For any point $y \in [-R, R]^d$ and any $\alpha_3 > 0$, if $\alpha' \leq c\sqrt{\frac{\hat{\lambda}_d}{d}}\alpha_3$ for some specific constant $c$, then*

$$|T_P(y) - T_{P_{\Delta}}(y)| \leq \alpha_3.$$

*Proof.* Let $X \sim \mathcal{N}(\mu, \Sigma)$ and let $\gamma$ be the "discretization random variable," so $X + \gamma \sim P_{\Delta}$.

Pick a vector $u$ such that $\|u\|_2 = 1$. After projecting onto $u$, we have a univariate random variable: $X^T u \sim \mathcal{N}(\mu^T u, u^T \Sigma u)$. Since $\|\gamma\|_2 \leq \frac{\sqrt{d}\alpha'}{2}$ and $\|u\|_2 = 1$, by Cauchy-Schwarz we have $\|\gamma^T u\|_2 \leq \frac{\sqrt{d}\alpha'}{2}$ as well.

The discretization can only affect the result when $X$ is close to the hyperplane, we have

$$\left|\Pr[X^T u \geq y^T u] - \Pr[(X + \gamma)^T u \geq y^T u]\right| \leq 2\Pr[X^T u \in y^T u \pm \sqrt{d}\alpha'/2] \tag{59}$$

$$\leq 2 \cdot \frac{1}{\sqrt{2\pi u^T \Sigma u}} \cdot \frac{\sqrt{d}\alpha'}{2}. \tag{60}$$

Since $\|u\|_2 = 1$, we have $\frac{1}{4}\hat{\lambda}_d \leq \lambda_d(\Sigma) \leq u^T \Sigma u$. Setting $\alpha'$ as in the lemma statement for a constant $c = \sqrt{2/\pi}$ makes this value at most $\alpha_3$.

Since the expected Tukey depth is defined as a minimum over all $u$, we are done. $\square$

We will thus be able to bound the volume ratios with an analog of Lemma 3.6.

**Lemma C.5** (Analog of Lemma 3.6). *Suppose for all $y \in [-R, R]^d$ that $|T_x(y) - T_{P_{\Delta}}(y)| \leq \alpha_1$ and $|T_{P_{\Delta}}(y) - T_P(y)| \leq \alpha_3$. Then, for all $p, q \in [0, 1/2]$,*

$$\frac{\text{Vol}(\mathcal{Y}_{np,x})}{\text{Vol}(\mathcal{Y}_{nq,x})} \leq \left(\frac{\Phi^{-1}(1 - p + \alpha_1 + \alpha_3)}{\Phi^{-1}(1 - q - \alpha_1 - \alpha_3)}\right)^d.$$

*Proof.* Applying the triangle inequality, we have $|T_x(y) - T_P(y)| \leq \alpha_1 + \alpha_3$ for all $y \in [-R, R]^d$.

To upper bound $\text{Vol}(\mathcal{Y}_{np,x})$, observe that $T_x(y) \geq p$ implies $T_P(y) \geq p - \alpha_1 - \alpha_3$, so by Lemma 3.3 we have $\|y - \mu\|_{\Sigma} \leq \Phi^{-1}(1 - p + \alpha_1 + \alpha_3)$. To lower bound $\text{Vol}(\mathcal{Y}_{nq,x})$, observe that $\|y - \mu\|_{\Sigma} \leq \Phi^{-1}(1 - q - \alpha_1 - \alpha_3)$ implies $T_P(y) \geq q + \alpha_1 + \alpha_3$, and thus $T_x(y) \geq q$.

Recalling that $\mathcal{B}_r$ denotes the Mahalanobis ball of radius $r$, we have

$$\frac{\text{Vol}(\mathcal{Y}_{np,x})}{\text{Vol}(\mathcal{Y}_{nq,x})} \leq \frac{\text{Vol}(\mathcal{B}_{\Phi^{-1}(1-p+\alpha_1+\alpha_3)})}{\text{Vol}(\mathcal{B}_{\Phi^{-1}(1-q-\alpha_1-\alpha_3)})} = \left(\frac{\Phi^{-1}(1 - p + \alpha_1 + \alpha_3)}{\Phi^{-1}(1 - q - \alpha_1 - \alpha_3)}\right)^d.$$

$\square$

The earlier version of this lemma had $\pm \alpha_1$ where we have $\pm(\alpha_1 + \alpha_3)$. Therefore, if $\alpha_1$ and $\alpha_3$ are sufficiently small, the proofs of the following lemmas go through with the exact same arguments.

**Lemma C.6** (Analog of Lemma 3.9). *Assume that for all $y \in [-R, R]^d$, $|T_x(y) - T_P(y)| \le \alpha_1 + \alpha_3$ with $\alpha_1 + \alpha_3 \le \frac{1}{10}$. There exists a constant $c$ such that, for any $\beta, \delta, \varepsilon > 0$ with $\varepsilon \le 1$ and $\delta \le \frac{1}{2}$, if $n \ge c\left(\frac{d + \log(1/\beta\delta)}{\varepsilon}\right)$ then $x$ is $\frac{\log(1/2\beta\delta)}{\varepsilon}$-far from $\mathrm{UNSAFE}_{(\varepsilon, \delta, n/4)}$.*

**Lemma C.7** (Analog of Lemma 3.10). *Assume that for all $y \in [-R, R]^d$ that $|T_x(y) - T_P(y)| \le \alpha_1 + \alpha_3$ with $\alpha_1 + \alpha_3 \le \frac{1}{10}$. For any $\beta > 0$ and $\alpha_2 \ge 2(\alpha_1 + \alpha_3)$, we have, for some constant $c$,*

$$\Pr_{y \sim \mathcal{M}_{n/4}(x)}\left[T_x(y) < \frac{1}{2} - \alpha_2\right] \le \left(\frac{c}{\alpha_2 - 2(\alpha_1 + \alpha_3)}\right)^d e^{-\alpha_2 n \varepsilon/4}. \tag{61}$$

Furthermore, discretizing with $\alpha' = \frac{\sqrt{\hat{\lambda}_d}\alpha}{d}$ instead of $O\left(\frac{\sqrt{\hat{\lambda}_d}\alpha}{\sqrt{d}}\right)$ will allow us to take $\alpha_3 = o(\alpha)$, so the discretization error does not affect the final sample complexity or accuracy. The only change is another additive $3\beta$ probability of failure, since (when the data is Gaussian) our bounding box may fail to contain all data points or we may have poor eigenvalue estimates.

**Theorem C.8** (Analog of Theorem 3.2). *There exists an absolute constant $C$ such that, for any $0 < \alpha, \beta, \varepsilon < 1$, $0 < \delta \le \frac{1}{2}$, mean $\mu$, and positive definite $\Sigma$, if $x \sim \mathcal{N}(\mu, \Sigma)^{\otimes n}$ and*

$$n \ge C\left(\frac{d + \log(1/\beta)}{\alpha^2} + \frac{d\log(1/\alpha) + \log(1/\beta)}{\alpha\varepsilon} + \frac{\log(1/\delta)}{\varepsilon}\right), \tag{62}$$

*then with probability at least $1 - 6\beta$, Algorithm 6 returns $\hat{\mu}$ such that $\|\hat{\mu} - \mu\|_\Sigma \le \alpha$.*

## C.3 A Finite Implementation of Algorithm 2

The finite implementation requires that we know the target accuracy $\alpha$ as well as an upper bound for the constant that goes into the subgaussian parameter $c_s$; note that this not the case for Gaussian data, as $c_s = 1$.

---

**Algorithm 7** Finite Implementation of $\mathcal{A}^G_{\varepsilon,\delta,\beta}(x)$

---

**Require:** Data set $x = (x_1, \ldots, x_{3n})^T \in \mathbb{R}^{3n \times d}$. Privacy parameters: $\varepsilon, \delta > 0$. Accuracy parameters: $\alpha, \beta > 0$. Subgaussian constant $c_s$.

**Stage 1: Range estimates**
1: Construct data set $u \in \mathbb{R}^{n \times d}$ where $u_i = (x_i - x_{i+n})/\sqrt{2}$, $i \in [n]$.
2: $\hat{\lambda}_1 \leftarrow \text{Eigen}_{\varepsilon,\delta,\beta}(u, 1)$                                    ▷ private estimate of largest eigenvalue
3: $\hat{\lambda}_d \leftarrow \text{Eigen}_{\varepsilon,\delta,\beta}(u, d)$                                   ▷ private estimate of smallest eigenvalue
4: $\sigma^2 \leftarrow 4c_s \hat{\lambda}_1$                                          ▷ upper bound on variance in every direction
5: $\{X^j_{\min}, X^j_{\max}\}_{j \in [d]} \leftarrow \text{Range}_{\varepsilon,\delta,\beta}(x, \sigma^2)$
6: Set

$$\alpha' \leftarrow O\left(\alpha \cdot \min\left\{\frac{\hat{\lambda}_d}{\hat{\lambda}_1} \cdot \frac{1}{d^{3/2} \log(n/\beta)}, \sqrt{\frac{\hat{\lambda}_d}{d}}\right\}\right). \tag{63}$$

7: $R \leftarrow \alpha' + \max_j \max\{|X^j_{\max}|, |X^j_{\min}|\}$

**Stage 2: Discretize**
8: $Q_{\alpha'} \leftarrow \alpha'$-fine grid over $[-R, R]^d$.
9: For all $i \in [n]$, let $x^\Delta_i = \arg\min_{p \in Q_{\alpha'}} \|p - x_i\|_1$.

**Stage 3: Run the algorithm.**
10: Run $\mathcal{A}^G_{\varepsilon,\delta,\beta}(x^\Delta)$.

---

Having constructed this grid, the projection step of Algorithm $\mathcal{A}^G_{\varepsilon,\delta,\beta}$ searches over all "good" data sets of size $3n$ whose data points belong on the grid $Q_{\alpha'}$, that is, line 6 of $\mathcal{A}^G_{\varepsilon,\delta,\beta}$ (Algorithm 2) is replaced by $\tilde{x} \leftarrow \arg\min_{z \in \mathcal{G}(\lambda) \cap Q_{\alpha'}} D_H(\bar{x}, z)$.

**Privacy.** Since the discretization process doesn't affect the privacy analysis of Algorithm 2, the overall privacy follows from composition and the privacy analysis in C.1.

**Computation.** The bottleneck in the algorithm above is the projection step, which is searching over all data sets on the grid $Q_{\alpha'}$, checking for each whether it is in the good set $\mathcal{G}(\lambda)$, and calculating its Hamming distance to $x$. For each data set, both these operations have running time polynomial in $d$ and $n$. However, the number of data sets in the grid is roughly $\left(\frac{2R}{\alpha'}\right)^{3dn} = \left(\frac{d\kappa(\|\mu\|_\infty + \sqrt{\lambda_1})}{\alpha}\right)^{O(dn)}$, where $\kappa = \lambda_1/\lambda_d$ is the condition number of the covariance matrix, making this algorithm computationally inefficient.

**Accuracy.** It suffices to show that the discretized data set is in the good set (Lemma C.11) and that the discretization adds negligible error (Lemma C.12).

First, observe that discretization (which happens coordinate-wise) has a limited effect in $\ell_2$ norm. For each $i \in [3n]$, let $x^\Delta_i = x_i + \gamma_i$. We snap each coordinate of $x_i$ to the nearest integer multiple of $\alpha'$, so $\|\gamma_i\|_\infty \le \alpha'/2$, which implies $\|\gamma_i\|^2_2 \le d(\alpha'/2)^2$ and thus $\|\gamma_i\|_2 \le \frac{\sqrt{d}\alpha'}{2}$. We now show that the discretized empirical covariance matrix is a close approximation to the original. We gather the following assumptions which we later show hold with high probability.

**Assumption C.9.** Suppose all the following conditions hold:

1. Our estimates $\hat{\lambda}_1, \hat{\lambda}_d$ have constants $c_1, c_2$ such that $c_1 \hat{\lambda}_1 \ge \lambda_1(\Sigma)$ and $c_2 \lambda_d(\Sigma) \le \hat{\lambda}_d$.

2. Our estimate $\hat{\lambda}_d$ has constant $c_3$ such that $\hat{\lambda}_d \le c_3 \lambda_d(\Sigma_x)$.

3. For all $i \in [3n]$, we have $\|x_i\|_\infty \le R$.

4. For all $i \in [3n]$, we have $\|x_i - \mu\|_2 \le c_4 \lambda_1(\Sigma) d \log(n/\beta)$ for some constant $c_4$.

**Lemma C.10** (Covariance after discretization). *Suppose Assumption C.9 holds. Then* $(1 - c_5)\Sigma_x \preceq \Sigma_{x^\Delta} \preceq (1 + c_5)\Sigma_x$ *for some constant* $c_5 \in (0, 1)$.

*Proof.* Write $\Sigma_{x^\Delta} = \Sigma_x + A$. We want to prove

$$- c_5 \Sigma_x \preceq A \preceq c_5 \Sigma_x, \tag{64}$$

for which it suffices to prove $\|A\|_2 \leq c_5 \lambda_d(\Sigma_x)$.

Let $u_i = x_i - x_{i+n}$ be the vectors that make up the empirical covariance, and let $u_i' = u_i + g_i$ be the discretized version. We have $g_i = \gamma_i - \gamma_{i+n}$ and thus $\|g_i\|_2 \leq 2\|\gamma_i\|_2 \leq \sqrt{d}\alpha'$. Then

$$
\begin{aligned}
A = \Sigma_{x^\Delta} - \Sigma_x &= \left(\frac{1}{2n}\sum_{i=1}^{n}(u_i + g_i)(u_i + g_i)^T\right) - \left(\frac{1}{2n}\sum_{i=1}^{n}u_i u_i^T\right) \\
&= \left(\frac{1}{2n}\sum_{i=1}^{n}u_i u_i^T + g_i g_i^T + g_i u_i^T + u_i g_i^T\right) - \left(\frac{1}{2n}\sum_{i=1}^{n}u_i u_i^T\right) \\
&= \left(\frac{1}{2n}\sum_{i=1}^{n}g_i g_i^T + g_i u_i^T + u_i g_i^T\right)
\end{aligned}
$$

Using the triangle inequality (and implicitly considering the maximum over $i$), we apply Fact A.4 to bound the spectral norm.

$$
\begin{aligned}
\|A\|_2 &\leq \frac{1}{2}\left(\|g_i g_i^T\|_2 + \|g_i u_i^T\|_2 + \|u_i g_i^T\|_2\right) \\
&\leq \frac{1}{2}\left(\|g_i\|_2^2 + 2\|g_i\|_2\|u_i^T\|_2\right) \\
&\leq \frac{d(\alpha')^2 + 4c_4 d^{3/2}\alpha' \lambda_1(\Sigma)\log(n/\beta)}{2}. & \text{(by assumption)}
\end{aligned}
$$

Since $\alpha' \leq 1$, use $(\alpha')^2 \leq \alpha'$ and simplify the upper bound to

$$\|A\|_2 \leq 3c_4 d^{3/2}\lambda_1(\Sigma)\log(n/\beta) \cdot \alpha'. \tag{65}$$

By our setting of $\alpha'$,

$$\alpha' \leq \frac{1}{3c_1 c_3 c_4} \cdot \frac{\hat{\lambda}_d}{\hat{\lambda}_1} \cdot \frac{\alpha}{d^{3/2}\log(n/\beta)}. \tag{66}$$

By assumption on our estimates for $\lambda_1(\Sigma)$ and $\lambda_d(\Sigma_x)$, and replacing the above $\alpha'$ in Eq. (65), we have that $\|A\|_2 = O(\lambda_d(\Sigma_x))$, so there exists indeed a $c_5$ such that $\|A\|_2 \leq c_5 \lambda_d(\Sigma_x)$. □

**Lemma C.11** (Analog of Lemma 4.7 and Lemma B.7). *Suppose Assumption C.9 holds. If* $x \in \mathcal{G}(\lambda)$, *then* $x^\Delta \in \mathcal{G}(\lambda')$ *for some* $\lambda' = O(\lambda)$.

*Proof.* Attack the definition of goodness directly. For all $i$,

$$
\begin{aligned}
\|x_i^\Delta - \mu_{x^\Delta}\|_{\Sigma_{x^\Delta}} &\leq c_6 \|x_i^\Delta - \mu_{x^\Delta}\|_{\Sigma_x} & \text{(by Proposition 4.4 for } c_6 = 1/\sqrt{1 - c_5}) \\
&= c_6 \|x_i^\Delta - x_i + x_i - \mu_{x^\Delta} + \mu_x - \mu_x\|_{\Sigma_x} \\
&= c_6 \|(\gamma_i) + (x_i - \mu_x) + (\mu_x - \mu_{x^\Delta})\|_{\Sigma_x} \\
&\leq c_6 \|x_i - \mu_x\|_{\Sigma_x} + c_6 \|\gamma_i\|_{\Sigma_x} + c_6 \|\mu_x - \mu_{x^\Delta}\|_{\Sigma_x}.
\end{aligned}
$$

The first term is bounded by $c_6 \lambda$, by our assumption that $x \in \mathcal{G}(\lambda)$. The second term we can bound because the $\gamma_i$'s have small $\ell_2$ norm. The third term is simply an average of the $\gamma_i$'s, so it will be bounded in the same manner. We have

$$\|\gamma_i\|_{\Sigma_x} \leq \frac{1}{\sqrt{\lambda_d(\Sigma_x)}} \cdot \|\gamma_i\|_2 \leq \sqrt{\frac{c_3}{\hat{\lambda}_d}} \cdot \frac{\sqrt{d}\alpha'}{2}. \tag{67}$$

Together, then, we have for all $i$ that

$$\|x_i^\Delta - \mu_{x^\Delta}\|_{\Sigma_{x^\Delta}} \le c_6\lambda + \frac{c_6\sqrt{c_3 d}}{\sqrt{\hat\lambda_d}} \cdot \alpha'. \tag{68}$$

By our setting of $\alpha'$, the second term is $O(1)$, thus $\lambda' = O(\lambda)$. □

The following lemma bounds the error of the estimator for input $x^\Delta$.

**Lemma C.12** (Analog of Lemma 4.8 and Lemma B.8). *Suppose Assumption C.9 holds. Suppose that $x \sim P_{\mu,\Sigma}^{\otimes 3n}$, where $P_{\mu,\Sigma}$ is a distribution with mean $\mu$, covariance $\Sigma$, such that $P_{\mu,\Sigma} \in \mathrm{subG}(c_s\Sigma)$ for some constant $c_s > 0$. Let $n = \Omega(\max\{(d + \log(1/\beta)), k\lambda\})$, where parameters $k, \lambda$ are set as in Algorithm 2. Then with probability at least $1 - \beta$, for $\hat\mu \sim \mathcal{N}(\mu_{x^\Delta}, C^2\Sigma_{x^\Delta})$,*

$$\|\hat\mu - \mu\|_\Sigma = O\left(\sqrt{\frac{d}{n} \cdot \log\frac{1}{\beta}} + \frac{d}{\varepsilon^2 n}\log^2\frac{1}{\delta\beta} \cdot \sqrt{\log\frac{n}{\beta}} + \alpha\right).$$

*Proof Sketch.* Because the error of discretization is negligible, the proof of this lemma is almost identical to that of Lemma 4.8 (and of its extension to subgaussian data, Lemma B.8). To see this, apply the triangle inequality:

$$\|\hat\mu - \mu\|_\Sigma \le \|\hat\mu - \mu_{x^\Delta}\|_\Sigma + \|\mu_{x^\Delta} - \mu_x\|_\Sigma + \|\mu_x - \mu\|_\Sigma. \tag{69}$$

By our assumption, Lemma C.11 implies that $\exists c_5$ such that $(1 - c_5)\Sigma_x \preceq \Sigma_{x^\Delta} \preceq (1 + c_5)\Sigma_x$. By Proposition 4.4, the first term is then $\|\hat\mu - \mu_{x^\Delta}\|_\Sigma = O(\|\hat\mu - \mu_{x^\Delta}\|_{\Sigma_{x^\Delta}})$. The analysis of this term, and that of the third, are independent of the discretization process. They follow by mean concentration of Gaussian and subgaussian data sets respectively and are included in the proof of Lemma 4.8 and Lemma B.8. The middle term is bounded as follows

$$\|\mu_{x^\Delta} - \mu_x\|_\Sigma \le (\lambda_d(\Sigma))^{-1/2}\|\mu_{x^\Delta} - \mu_x\|_2 \tag{70}$$

$$\le \sqrt{\frac{c_2}{\hat\lambda_d}}\frac{\sqrt{d}\alpha'}{2} \tag{71}$$

$$= O(\alpha). \tag{72}$$

Therefore, this incurs only a constant factor increase in the error bound. □

We now state the accuracy guarantees of our finite implementation.

**Theorem C.13** (Accuracy of Algorithm 7). *There exists an absolute constant $C$ such that, for any $0 < \alpha, \beta, \varepsilon, \delta < 1$, mean $\mu$, and positive definite $\Sigma$, if $x \sim P_{\mu,\Sigma}^{\otimes n}$, where $P_{\mu,\Sigma} \in \mathrm{subG}(c_s\Sigma)$ for some constant $c_s > 0$, and*

$$n \ge C\left(\frac{d}{\alpha^2}\log\frac{1}{\beta} + \frac{d}{\alpha\varepsilon^2}\log^3\frac{1}{\delta\beta} \cdot \log\frac{d\log(1/\delta\beta)}{\alpha\varepsilon}\right), \tag{73}$$

*then with probability at least $1 - 7\beta$, Algorithm 2 returns $\mathcal{A}_{\varepsilon,\delta,\beta}^G(x) = \hat\mu$ such that $\|\hat\mu - \mu\|_\Sigma \le \alpha$.*

The proof of the theorem follows exactly the same steps as its counterparts for Gaussian and subgaussian distributions in Sections 4 and B respectively, combined with the analogous lemmas above. It remains to argue that Assumption C.9 holds with probability at least $1 - 4\beta$, and then the theorem would follow by a union bound.

Note that if $x \sim P_{\mu,\Sigma}$ where $P_{\mu,\Sigma}$ has mean $\mu$, covariance $\Sigma$ and is subgaussian with parameter $c_s\Sigma$, then every coordinate is also subgaussian with parameter $c_s\lambda_1(\Sigma)$. By the guarantees of $\mathrm{Eigen}_{\varepsilon,\delta,\beta}$ (Lemma C.2), with probability $1 - 2\beta$, the eigenvalue estimates are good approximations of the true eigenvalues, that is, $\frac{\lambda_1(\Sigma)}{4} \le \hat\lambda_1 \le 4\lambda_1(\Sigma)$ and $\frac{\lambda_d(\Sigma)}{4} \le \hat\lambda_1 \le 4\lambda_d(\Sigma)$. By substituting this bound, it follows that in every coordinate $x$

is $4c_s\hat{\lambda}_1$-subgaussian. Applying the guarantees of $\text{Range}_{\varepsilon,\delta,\beta}$ (Lemma C.3) and by union bound and our choice of $R$, we have that with probability at least $1-3\beta$, the size of the $d$-dimensional box that encloses our grid is set so that all points $x_i$ of the original data set as well as all points $x_i^\Delta$ of the discretized dataset belong in the box, that is, $\|x_i\|_\infty \le R$ and $\|x_i^\Delta\|_\infty \le R$. Therefore, with probability at least $1-3\beta$, item 1 and 3 of Assumption C.9 hold (for $c_1 = 4$ and $c_2 = 1/4$).

Moreover, by Lemma B.4, with probability $1-\beta$, we have that for all $i \in [3n]$, $\|x_i - \mu\|_2 \le c_4\lambda_1(\Sigma)d\log(n/\beta)$ for some constant $c_4$ and that if $n = \Omega(d + \log(1/\beta))$, then $\lambda_d(\Sigma) = \Theta(\lambda_d(\Sigma_x))$. By the latter and since $\hat{\lambda}_d \le 4\lambda_d(\Sigma)$, we have that for some constant $c_3$, $\hat{\lambda}_d \le \frac{1}{c_3}\lambda_d(\Sigma_x)$. Since for the stated sample complexity $n$ satisfies this condition, we have that items 2 and 4 of Assumption C.9 hold as well.

# D  Additional Proofs

## D.1  Tukey-Depth Mechanism

The mechanism fits into the well-known propose-test-release framework of [30]; privacy follows from a standard calculation. We include it here for completeness.

**Proposition D.1.** *Algorithm 1 is $(2\varepsilon, e^\varepsilon\delta)$-differentially private.*

*Proof.* Take adjacent $x, x'$ and fix some subset $B \subseteq \mathcal{Y} \cup \{\text{FAIL}\}$. As shorthand, let $F = \{\text{FAIL}\}$ and write $\mathcal{A}$ in place of $\mathcal{A}_{\varepsilon,\delta,t}^E$.

We proceed by cases. Suppose first that $\mathcal{M}_{\varepsilon,t}(x) \not\approx_{\varepsilon,\delta} \mathcal{M}_{\varepsilon,t}(x')$, so running the restricted sampler may reveal too much. Then both $x, x' \in \text{UNSAFE}$, and for both we compute distance $h = 0$ to unsafety. Thus

$$\Pr[\mathcal{A}(x) \in B] = \Pr[\mathcal{A}(x) \in B \cap F] + \Pr[\mathcal{A}(x) \in B \setminus F]$$
$$\le \Pr[\mathcal{A}(x) \in B \cap F] + \Pr[\mathcal{A}(x) \notin F]$$
$$\le \Pr[\mathcal{A}(x') \in B] + \Pr[\mathcal{A}(x') \notin F],$$

where the last line follows from the facts that both $x$ and $x'$ have the same probability of failing and that $B \cap F \subseteq B$. The threshold $\frac{\log(1/2\delta)}{\varepsilon}$ is set so that the probability a Laplace random variable $\text{Lap}(1/\varepsilon)$ exceeds it is $\Pr[\mathcal{A}(x) \notin F] = \delta$.

Now suppose $\mathcal{M}_{\varepsilon,t}(x) \approx_{\varepsilon,\delta} \mathcal{M}_{\varepsilon,t}(x')$. Since $x$ and $x'$ are adjacent, the distances-to-unsafety we compute under $x$ and $x'$ can differ by at most 1, so the probability of failing can differ by at most a factor of $e^\varepsilon$. We break down the probability similarly:

$$\Pr[\mathcal{A}(x) \in B] = \Pr[\mathcal{A}(x) \in B \cap F] + \Pr[\mathcal{A}(x) \in B \setminus F]$$
$$= \Pr[\mathcal{A}(x) \in B \mid \mathcal{A}(x) \in F]\Pr[\mathcal{A}(x) \in F]$$
$$\quad + \Pr[\mathcal{A}(x) \in B \mid \mathcal{A}(x) \notin F]\Pr[\mathcal{A}(x) \notin F]$$
$$\le e^\varepsilon \bigg(\Pr[\mathcal{A}(x) \in B \mid \mathcal{A}(x) \in F]\Pr[\mathcal{A}(x') \in F]$$
$$\quad + \Pr[\mathcal{A}(x) \in B \mid \mathcal{A}(x) \notin F]\Pr[\mathcal{A}(x') \notin F]\bigg).$$

Since $B$ either contains $\text{FAIL}$ or it doesn't, we have $\Pr[\mathcal{A}(x) \in B \mid \mathcal{A}(x) \in F] = \Pr[\mathcal{A}(x') \in B \mid \mathcal{A}(x') \in F]$. Furthermore, since not failing means we run $\mathcal{M}_{\varepsilon,t}(x)$, we have

$$\Pr[\mathcal{A}(x) \in B] \le e^\varepsilon\big(\Pr[\mathcal{A}(x') \in B \cap F] + \Pr[\mathcal{M}_{\varepsilon,t}(x) \in B]\Pr[\mathcal{A}(x') \notin F]\big)$$
$$\le e^\varepsilon\big(\Pr[\mathcal{A}(x') \in B \cap F] + \big(e^\varepsilon\Pr[\mathcal{M}_{\varepsilon,t}(x') \in B] + \delta\big)\Pr[\mathcal{A}(x') \notin F]\big),$$

applying our assumption that $\mathcal{M}_{\varepsilon,t}(x) \approx_{\varepsilon,\delta} \mathcal{M}_{\varepsilon,t}(x')$. To finish the proof, we simplify:

$$\Pr[\mathcal{A}(x) \in B] \le e^\varepsilon\Pr[\mathcal{A}(x') \in B \cap F] + e^\varepsilon e^\varepsilon\Pr[\mathcal{M}_{\varepsilon,t}(x') \in B]\Pr[\mathcal{A}(x') \notin F] + e^\varepsilon\delta\Pr[\mathcal{A}(x') \notin F]$$
$$= e^\varepsilon\Pr[\mathcal{A}(x') \in B \cap F] + e^{2\varepsilon}\Pr[\mathcal{A}(x') \in B \setminus F] + e^\varepsilon\delta\Pr[\mathcal{A}(x') \notin F]$$
$$\le e^{2\varepsilon}\Pr[\mathcal{A}(x') \in B] + e^\varepsilon\delta.$$

Since $e^\varepsilon \geq 1$, these parameters are also an upper bound for the first case. The fact that $\Pr[\mathcal{A}(x') \in B] \leq e^{2\varepsilon}\Pr[\mathcal{A}(x) \in B] + e^\varepsilon\delta$ follows by an identical argument. $\qquad\square$

**Proposition D.2** (Restatement of Proposition 3.3). *For any $\mu, y \in \mathbb{R}^d$ and positive definite $\Sigma$, $T_{\mathcal{N}(\mu,\Sigma)}(y) = T_P(y) = \Phi(-\|y - \mu\|_\Sigma)$.*

*Proof.* If $y = \mu$, by the symmetry of the Gaussian, $T_P(y) = \frac{1}{2} = \Phi(0)$. So consider $y \neq \mu$.

We first calculate for a given $u$, and then take the minimum. If $u = 0$, then $\Pr[X^T u \geq y^T u] = 1$, so assume nonzero $u$. We lower bound $\Pr[X^T u > y^T u]$, where $X \sim P = \mathcal{N}(\mu, \Sigma)$, and begin by rewriting the random variable to be drawn from $\mathcal{N}(0, \mathbb{I})$:

$$\Pr_{X \sim P}[X^T u > y^T u] = \Pr_{Z \sim \mathcal{N}(0,\mathbb{I})}\left[(\Sigma^{1/2}Z + \mu)^T u > y^T u\right].$$

We move terms to the right, multiply by $\Sigma^{-1/2}\Sigma^{1/2}$, and normalize by $\|\Sigma^{1/2}u\|_2$:

$$\Pr_{Z \sim \mathcal{N}(0,\mathbb{I})}\left[(\Sigma^{1/2}Z + \mu)^T u > y^T u\right] = \Pr_{Z \sim \mathcal{N}(0,\mathbb{I})}\left[(\Sigma^{1/2}Z)^T u > (y - \mu)^T u\right]$$

$$= \Pr_{Z \sim \mathcal{N}(0,\mathbb{I})}\left[Z^T(\Sigma^{1/2}u) > (\Sigma^{-1/2}(y - \mu))^T(\Sigma^{1/2}u)\right]]$$

$$= \Pr_{Z \sim \mathcal{N}(0,\mathbb{I})}\left[Z^T(\Sigma^{1/2}u)/\|\Sigma^{1/2}u\|_2 > (\Sigma^{-1/2}(y - \mu))^T(\Sigma^{1/2}u)/\|\Sigma^{1/2}u\|_2\right].$$

Let $u' = \Sigma^{1/2}u/\|\Sigma^{1/2}u\|_2$, and recall that, since $u'$ is a unit vector, $-Z^T u' \sim \mathcal{N}(0, 1)$. We have

$$\Pr_{Z \sim \mathcal{N}(0,\mathbb{I})}\left[Z^T u' > (\Sigma^{-1/2}(y - \mu))^T u'\right] = \Pr_{Z_1 \sim \mathcal{N}(0,1)}\left[Z_1 < -(\Sigma^{-1/2}(y - \mu))^T u'\right]$$

$$= \Phi(-(\Sigma^{-1/2}(y - \mu))^T u').$$

Since $\Phi$ is an increasing function, the above term is minimized when $u' = \frac{\Sigma^{-1/2}(y-\mu)}{\|\Sigma^{-1/2}(y-\mu)\|_2}$, that is, $u$ is a rescaling of $\Sigma^{-1}(y - \mu)$. With this value of $u'$, we see that $T_P(y) = \Phi(-\|y - \mu\|_\Sigma)$. Since this is strictly less than $\frac{1}{2}$ for $y \neq \mu$, our exclusion of $u = 0$ did not affect the outcome. $\qquad\square$

## D.2 Empirically Rescaled Gaussian Mechanism

### D.2.1 Implications of Goodness

**Lemma D.3** (Restatement of Lemma 4.10). *If $x \in \mathcal{G}(\lambda)$, for any indices $i, j \in [3n]$,*

$$(x_i - x_j)^T \Sigma_x^{-1}(x_i - x_j) \leq 4\lambda.$$

*In particular, this applies to $u_i^T \Sigma_x^{-1} u_i$ for all $i \in [n]$, where $u_i = x_i - x_{i+n}$.*

*Proof.* Fix $i, j \in [2n]$. Since $x \in \mathcal{G}(\lambda)$, $\|x_i - \mu_x\|_{\Sigma_x} \leq \sqrt{\lambda}$ and $\|x_j - \mu_x\|_{\Sigma_x} \leq \sqrt{\lambda}$. It holds that

$$(x_i - x_j)^T \Sigma_x^{-1}(x_i - x_j) = \|x_i - x_j\|_{\Sigma_x}^2$$

$$= \|(x_i - \mu_x) - (x_j - \mu_x)\|_{\Sigma_x}^2$$

$$\leq \left(\|x_i - \mu_x\|_{\Sigma_x} + \|x_j - \mu_x\|_{\Sigma_x}\right)^2 \qquad \text{(by triangle inequality)}$$

$$\leq (2\sqrt{\lambda})^2 = 4\lambda$$

This concludes the proof of the lemma. $\qquad\square$

**Lemma D.4** (Restatement of Lemma 4.11). *Suppose $x, y \in \mathcal{G}(\lambda)$ and $D_H(x, y) \leq k$, with $2k\lambda < n$. For any vector $v$ we have*

$$v^T \Sigma_y^{-1} v \leq \frac{1}{1 - 2k\lambda/n} \cdot v^T \Sigma_x^{-1} v.$$

*Proof.* Define the matching paired indices $S = \{i \in [n] : x_i = y_i \text{ and } x_{i+n} = y_{i+1}\}$. We have $|S| \geq n - k$. Define $\Sigma_{x_S} = \frac{1}{2n} \sum_{i \in S} (x_i - x_{i+n})(x_i - x_{i+n})^T$. Note that we normalize by $\frac{1}{2n}$ instead of $\frac{1}{2|S|}$. We will upper bound $v^T \Sigma_{x_S}^{-1} v$. This will finish the proof, since $v^T \Sigma_y^{-1} v \leq v^T \Sigma_{x_S}^{-1} v$. To see this fact, note that $\Sigma_y \succeq \Sigma_{x_S}$, since $\Sigma_y$ is $\Sigma_{x_S}$ plus a positive semidefinite matrix. So $\Sigma_y^{-1} \preceq \Sigma_{x_S}^{-1}$ [46, Cor 7.7.4.a].

Set $u_i = x_i - x_{i+n}$ and write

$$\Sigma_x = \Sigma_{x_S} + \frac{1}{2n} \sum_{i \in [n] \setminus S} u_i u_i^T.$$

Conjugating by $\Sigma_x^{-1/2}$ on both sides, we have

$$\mathbb{I} = \Sigma_x^{-1/2} \Sigma_{x_S} \Sigma_x^{-1/2} + \frac{1}{2n} \sum_{i \in [n] \setminus S} \left(\Sigma_x^{-1/2} u_i\right)\left(\Sigma_x^{-1/2} u_i\right)^T \tag{74}$$

$$= \Sigma_x^{-1/2} \Sigma_{x_S} \Sigma_x^{-1/2} + \frac{1}{2n} A \tag{75}$$

defining matrix $A$ as the sum of the second term. By the triangle inequality,

$$\|A\|_2 \leq k \cdot \max_{i \in [n] \setminus S} u_i^T \Sigma_x^{-1} u_i \leq 4k\lambda,$$

where the last inequality holds by the assumption of goodness and Lemma 4.10. By assumption, $2k\lambda < n$, so $\|A\|_2 < 2n$, which implies that $\mathbb{I} - \frac{1}{2n} A$ is positive definite and thus invertible. This and Eq. (75) imply that $\Sigma_{x_S}$ is also invertible. Rearranging and taking the inverse gives us

$$\Sigma_x^{1/2} \Sigma_{x_S}^{-1} \Sigma_x^{1/2} = \left(\mathbb{I} - \frac{1}{2n} A\right)^{-1}.$$

The operator norm of the above matrix is at most $\frac{1}{1 - 2k\lambda/n}$. We can use this to bound $v^T \Sigma_{x_S}^{-1} v$:

$$v^T \Sigma_{x_S} v = \left(\Sigma_x^{-1/2} v\right)^T \left(\Sigma_x^{1/2} \Sigma_{x_S}^{-1} \Sigma_x^{1/2}\right)\left(\Sigma_x^{-1/2} v\right)$$

$$\leq \left\|\Sigma_x^{1/2} \Sigma_{x_S}^{-1} \Sigma_x^{1/2}\right\|_2 \cdot \left\|\Sigma_x^{-1/2} v\right\|_2^2$$

$$\leq \frac{1}{1 - 2k\lambda/n} \cdot v^T \Sigma_x^{-1} v.$$

This completes the proof. □

**Lemma D.5** (Restatement of Lemma 4.12). *Suppose $x, y \in \mathcal{G}(\lambda)$ and $D_H(x, y) \leq k$, with $2k\lambda < n$. Then*

$$\left\|\Sigma_x^{-1/2} \Sigma_y \Sigma_x^{-1/2} - \mathbb{I}\right\|_{\text{tr}} \leq 2k\lambda\left(\frac{1}{n - 2k\lambda} + \frac{1}{n}\right)$$

$$\left\|\Sigma_y^{-1/2} \Sigma_x \Sigma_y^{-1/2} - \mathbb{I}\right\|_{\text{tr}} \leq 2k\lambda\left(\frac{1}{n - 2k\lambda} + \frac{1}{n}\right)$$

*Proof.* Define the indices of agreement: $S = \{i \in [n] : x_i = y_i \text{ and } x_{i+n} = y_{i+n}\}$. Since $D_H(x, y) \leq k$, it holds that $|S| \geq n - k > n(1 - 1/2\lambda)$, where the last inequality holds by assumption. Recall $u_i = x_i - x_{i+n}$ and define $v_i = y_i - y_{i+n}$. We can write, defining matrix $A$,

$$\Sigma_y = \Sigma_x + \frac{1}{2n} \sum_{i \in [n] \setminus S} v_i v_i^T - \frac{1}{2n} \sum_{i \in [n] \setminus S} u_i u_i^T \stackrel{\text{def}}{=} \Sigma_x + A.$$

Conjugating by $\Sigma_x^{-1/2}$ and subtracting $\mathbb{I}$ from both sides, we get

$$\Sigma_x^{-1/2}\Sigma_y\Sigma_x^{-1/2} - \mathbb{I} = \Sigma_x^{-1/2}(\Sigma_x + A)\Sigma_x^{-1/2} - \mathbb{I}$$

$$= \Sigma_x^{-1/2}A\Sigma_x^{-1/2}$$

$$= \frac{1}{2n}\sum_{i\in[n]\setminus S}\left(\Sigma_x^{-1/2}v_i\right)\left(\Sigma_x^{-1/2}v_i\right)^T - \frac{1}{2n}\sum_{i\in[n]\setminus S}\left(\Sigma_x^{-1/2}u_i\right)\left(\Sigma_x^{-1/2}u_i\right)^T.$$

Since the trace norm satisfies the triangle inequality, we have

$$\|\Sigma_x^{-1/2}\Sigma_y\Sigma_x^{-1/2} - \mathbb{I}\|_{\mathrm{tr}} \leq \frac{1}{2n}\sum_{i\in[n]\setminus S}\left[\left\|\left(\Sigma_x^{-1/2}v_i\right)\left(\Sigma_x^{-1/2}v_i\right)^T\right\|_{\mathrm{tr}} + \left\|\left(\Sigma_x^{-1/2}u_i\right)\left(\Sigma_x^{-1/2}u_i\right)^T\right\|_{\mathrm{tr}}\right].$$

Each term in these sums is an outer product of the form $(\Sigma_x^{-1/2}v)(\Sigma_x^{-1/2}v)^T$. Since for every vector $v$, $\|vv^T\|_{\mathrm{tr}} = v^Tv$ (see Proposition A.4), we can write

$$\|\Sigma_x^{-1/2}\Sigma_y\Sigma_x^{-1/2} - \mathbb{I}\|_{\mathrm{tr}} \leq \frac{1}{2n}\sum_{i\in[n]\setminus S} v_i^T\Sigma_x^{-1}v_i + u_i^T\Sigma_x^{-1}u_i.$$

By Lemma 4.10, for all $i \in [n]\setminus S$ we have $u_i^T\Sigma_x^{-1}u_i \leq 4\lambda$. By Lemmas 4.10 and 4.11, for all $i$ we have

$$v_i^T\Sigma_x^{-1}v_i \leq \frac{1}{1 - 2k\lambda/n}v_i^T\Sigma_y^{-1}v_i \leq \frac{4\lambda}{1 - 2k\lambda/n}.$$

Combining these establishes the first inequality. The second holds by a symmetrical argument. $\qquad\square$

### D.2.2  Privacy analysis

**Proposition D.6** (Coupling and Data Order). *Suppose we have a mechanism $\mathcal{M} = \mathcal{A} \circ \mathcal{P}$, where $\mathcal{P}$ randomly permutes our data and $\mathcal{A}$ has the following privacy guarantee: for any two data sets $\bar{x}$ and $\bar{y}$ with $D_H(\bar{x}, \bar{y}) \leq \xi$ and any $O \subseteq \mathrm{Range}(\mathcal{M}) = \mathrm{Range}(\mathcal{A})$,*

$$\Pr[\mathcal{A}(\bar{x}) \in O] \leq e^\varepsilon \Pr[\mathcal{A}(\bar{y}) \in O] + \delta.$$

*Then, for any $x$ and $y$ which differ in at most $\xi$ points,*

$$\Pr[\mathcal{M}(x) \in O] \leq e^\varepsilon \Pr[\mathcal{M}(y) \in O] + \delta.$$

*In other words, if $\mathcal{A}$ is $(\varepsilon, \delta)$-differentially private under the stricter Hamming distance adjacency, then $\mathcal{M}$ is $(\varepsilon, \delta)$-differentially private under the symmetric difference notion of adjacency.*

*Proof.* Let $S_m$ be the set of permutations on $m$. For any $x$ and $y$ which differ in $\xi$ points, let $\sigma^*$ be an "aligning" permutation, such that $D_H(x, \sigma^*(y)) = \xi$. So we can write

$$\Pr[\mathcal{M}(x) \in O] = \sum_{\sigma \in S_m} \frac{1}{m!} \cdot \Pr[\mathcal{A}(\sigma(x)) \in O]$$

$$\leq \sum_{\sigma \in S_m} \frac{1}{m!} \cdot (e^\varepsilon \Pr[\mathcal{A}(\sigma(\sigma^*(y))) \in O] + \delta),$$

since $D_H(\sigma(x), \sigma(\sigma^*(y))) = \xi$. Furthermore, note that $f(\sigma) \overset{\text{def}}{=} \sigma(\sigma^*)$ is a bijection from $S_m$ to itself, so we can rewrite this sum as over a reordering of $S_m$:

$$\Pr[\mathcal{M}(x) \in O] \leq \sum_{\sigma' \in S_m}\left[\frac{1}{m!} \cdot (e^\varepsilon \Pr[\mathcal{A}(\sigma'(y)) \in O] + \delta)\right] \leq e^\varepsilon \Pr[\mathcal{M}(y) \in O] + \delta.$$

This completes the proof of the lemma. $\qquad\square$

**Corollary D.7** (Restatement of Corollary 4.14). *Algorithm 2 is $(3\varepsilon, e^{\varepsilon}(1 + e^{\varepsilon})\delta)$-differentially private.*

We show that, if $x$ and $y$ are far from $\mathcal{G}(\lambda)$, then with high probability the algorithm fails. If they are close to $\mathcal{G}(\lambda)$, then by Theorem 4.13 the output distributions are indistinguishable.

*Proof.* Take adjacent data sets $x$ and $y$ and some output event $E \in \mathbb{R}^d \cup \{\texttt{FAIL}\}$. As shorthand, let $F = \{\texttt{FAIL}\}$ and write $\mathcal{A}$ in place of $\mathcal{A}^G_{\varepsilon,\delta,\beta}$. Recall that, by Lemma D.6, it suffices to prove privacy for $x, y$ with Hamming distance 1.

For the first case, assume that $\max_{z \in \{x,y\}} D_H(z, \mathcal{G}(\lambda)) \geq \frac{\log(1/\delta\beta)}{\varepsilon} + 1$, so we know both $D_H(y, \mathcal{G}(\lambda)) \geq \frac{\log(1/\delta\beta)}{\varepsilon}$ and $D_H(x, \mathcal{G}(\lambda)) \geq \frac{\log(1/\delta\beta)}{\varepsilon}$. Then, by the CDF of the Laplace distribution and the fact that we set our threshold to $\frac{\log(1/\beta)}{\varepsilon}$, under both $x$ and $y$ we have $\Pr[\texttt{FAIL}] \geq 1 - \delta$. Thus

$$
\begin{aligned}
\Pr[\mathcal{A}(x) \in E] &= \Pr[\mathcal{A}(x) \in E \mid \mathcal{A}(x) \in F]\Pr[\mathcal{A}(x) \in F] + \Pr[\mathcal{A}(x) \in E \mid \mathcal{A}(x) \notin F]\Pr[\mathcal{A}(x) \notin F] \\
&= \Pr[\mathcal{A}(y) \in E \mid \mathcal{A}(y) \in F]\Pr[\mathcal{A}(x) \in F] + \Pr[\mathcal{A}(x) \in E \mid \mathcal{A}(x) \notin F]\Pr[\mathcal{A}(x) \notin F] \\
&\leq \Pr[\mathcal{A}(y) \in E \mid \mathcal{A}(y) \in F](e^{\varepsilon}\Pr[\mathcal{A}(y) \in F]) + 1 \cdot \delta,
\end{aligned}
$$

where in the first line we used the fact that $E$ either contains $\texttt{FAIL}$ or it does not, and in the second line we used (twice) the CDF of the Laplace distribution. Since

$$
\Pr[\mathcal{A}(y) \in E \mid \mathcal{A}(y) \in F]\Pr[\mathcal{A}(y) \in F] = \Pr[\mathcal{A}(y) \in E \cap F] \leq \Pr[\mathcal{A}(y) \in E],
$$

we have our $(\varepsilon, \delta)$ upper bound for $\Pr[\mathcal{A}(x) \in E]$. The upper bound for $\Pr[\mathcal{A}(y) \in E]$ follows from an identical argument. This finishes the first case.

For the second case, assume that $\max_{z \in \{x,y\}} D_H(z, \mathcal{G}(\lambda)) \leq \frac{\log(1/\delta\beta)}{\varepsilon}$, so writing $\tilde{x}, \tilde{y}$ for the projections into $\mathcal{G}(\lambda)$,

$$
D_H(\tilde{x}, \tilde{y}) \leq D_H(\tilde{x}, x) + D_H(x, y) + D_H(y, \tilde{y}) \leq \frac{2\log(1/\delta\beta)}{\varepsilon} + 1.
$$

Recall that, if $\mathcal{A}(x) \neq \texttt{FAIL}$, then the algorithm samples from $\mathcal{N}(\mu_{\tilde{x}}, C^2\Sigma_{\tilde{x}})$, and analogously for $\mathcal{A}(y)$. By Theorem 4.13, for any $\tilde{x}, \tilde{y} \in \mathcal{G}(\lambda)$ such that $D_H(\tilde{x}, \tilde{y}) \leq k$, if

$$
n > 2k\lambda \quad \text{and} \quad \varepsilon \geq 10k\lambda\left(\frac{1}{n - 2k\lambda} + \frac{1}{n}\right)\log\frac{2}{\delta}, \tag{76}
$$

then $\mathcal{N}(\mu_{\tilde{x}}, C^2\Sigma_{\tilde{x}}) \approx_{2\varepsilon, (1+e^{\varepsilon})\delta} \mathcal{N}(\mu_{\tilde{y}}, C^2\Sigma_{\tilde{y}})$. We can assume the conditions in (76) are satisfied, since otherwise the algorithm immediately aborts. Write $u_x \sim \mathcal{N}(\mu_{\tilde{x}}, C^2\Sigma_{\tilde{x}})$ and $u_y \sim \mathcal{N}(\mu_{\tilde{y}}, C^2\Sigma_{\tilde{y}})$ We have

$$
\begin{aligned}
\Pr[\mathcal{A}(x) \in E] &= \Pr[\mathcal{A}(x) \in E \mid \mathcal{A}(x) \in F]\Pr[\mathcal{A}(x) \in F] + \Pr[\mathcal{A}(x) \in E \mid \mathcal{A}(x) \notin F]\Pr[\mathcal{A}(x) \notin F] \\
&\leq e^{\varepsilon}\Pr[\mathcal{A}(x) \in E \mid \mathcal{A}(x) \in F]\Pr[\mathcal{A}(y) \in F] + e^{\varepsilon}\Pr[\mathcal{A}(x) \in E \mid \mathcal{A}(x) \notin F]\Pr[\mathcal{A}(y) \notin F] \\
&= e^{\varepsilon}(\Pr[\mathcal{A}(y) \in E \mid \mathcal{A}(y) \in F]\Pr[\mathcal{A}(y) \in F] + \Pr[u_x \in E]\Pr[\mathcal{A}(y) \notin F]) \\
&\leq e^{\varepsilon}\left(\Pr[\mathcal{A}(y) \in E \mid \mathcal{A}(y) \in F]\Pr[\mathcal{A}(y) \in F] + \left(e^{2\varepsilon}\Pr[u_y \in E] + (1 + e^{\varepsilon})\delta\right)\Pr[\mathcal{A}(y) \notin F]\right) \\
&\leq e^{\varepsilon}\left(e^{2\varepsilon}\Pr[\mathcal{A}(y) \in E \cap F] + e^{2\varepsilon}\Pr[\mathcal{A}(y) \in E \setminus F]\right) + e^{\varepsilon}(1 + e^{\varepsilon})\delta \\
&\leq e^{3\varepsilon}\Pr[\mathcal{A}(y) \in E] + e^{\varepsilon}(1 + e^{\varepsilon})\delta.
\end{aligned}
$$

An identical calculation yields the corresponding upper bound for $\Pr[\mathcal{A}(y) \in E]$. $\qquad\square$