# OpenReview forum: "Covariance-Aware Private Mean Estimation Without Private Covariance Estimation"
_NeurIPS.cc/2021/Conference — NeurIPS 2021 Spotlight_

### Official Review · Reviewer_ubCi · 2021-06-30

**Rating:** 7
**Confidence:** 3

**Summary:**

The paper studies the problem of private mean/median estimation for Guassian and subGaussian variables. In order to reduce the noise required to satisfy privacy, the algorithms tailor the noise to the (unknown) covariance of the distribution, without directly estimating the covariance matrix. This is an important distinction from some prior methods, as privately estimating the covariance matrix is expensive. The methods used combine techniques including the exponential mechanism with Tukey depth, the Gaussian mechanism, and propose-test-release(PTR). While the authors mention that the privacy analysis uses standard techniques, they give a rigorous proof of the utility of the mechanism which they mention uses novel techniques.


**Ethical Concerns:**

No issues.

**Limitations And Societal Impact:**

No issues.

**Main Review:**

Originality:
Mean estimation is one of the oldest problems in DP, and there have been several works developing techniques to improve the utility of mean estimation in DP. One of the key distinctions of this paper is that the noise is scaled according to the (unknown) covariance matrix, without privately releasing the estimated covariance function. As the authors mention, this greatly improves the sample complexity. It also offers much more realistic assumptions on the data generating distribution, rather than requiring tight apriori bounds on the covariance matrix, such as in Karwa & Vadhan.

The proposed mechanism builds upon several existing DP techniques, including the exponential mechanism with Tukey depth, the Gaussian mechanism, and propose-test-release(PTR). Still, these techniques are combined in a novel manner. More importantly, the utility analysis is novel and this is where the main contributions of the paper are.

Quality:
The theoretical results in the paper support both the privacy and utility of the proposed mechanisms.

The authors suggest that the proposed mechanism is primarily of theoretical interest, as it has exponential running time in both the dimension and sample size. The lack of a practical algorithm limits the work, making it seem somewhat incomplete.

Furthermore, the paper does not have any empirical evaluations of the mechanism. The inclusion of simulations would help to demonstrate the utility of the proposed mechanism compared to prior approaches.

Significance:
From a theoretical standpoint, the paper provides a novel and important result demonstrating that mean estimation can be achieved privately with much improved utility compared to previous results. Furthermore, the results and techniques could provide building blocks for future researchers to design efficient DP algorithms for other problems. Or, potentially future researchers could modify the techniques of this paper to produce a computationally efficient mechanism with similar utility.

As mentioned earlier, the lack of a computationally efficient algorithm limits the usefulness of this paper. As the authors mentioned, the proposed algorithm has running time exponential in both n and d, making it entirely impractical for the analysis of real datasets. As such, the results of the paper are currently only of interest to other privacy experts rather than to practitioners looking to implement DP methods.

Another limitation of the result is that it is framed in approximate-DP. Approximate-DP has been criticized since it allows for a catastrophic failure of privacy with a small probability (approx  delta). Due to this weakness, there has been increasing interest in Renyi-divergence based DP frameworks, which also allow for a relaxation of pure DP, but protect against the catastrophic failure that approx-DP allows. Since the authors are using the PTR technique for their results, I suspect that their results cannot be analyzed using Renyi-DP.

Clarity:
The authors do a good job of communicating the prior approaches to the problem, and the sample complexity & assumptions of the previous methods. They effectively communicate how their proposed method improves over the prior work in terms of both sample complexity and more realistic assumptions.

The Tukey depth is introduced in equation (4), but little intuition is given about why it is a useful score function. Some exposition discussing the intuition of Tukey depth, both why it is useful to estimate the location as well as why it is useful in the exponential mechanism would be helpful.

In Theorem 3.2, it is not clear what is meant by the superscript $\otimes n$ means.

In line 303, it is not clear what is meant by Y_{t,x}. I gathered later that this may be the support of M_{epsilon,t}(x), but this should be clearly defined.

Conclusion:
The paper proposes new DP mean estimation algorithms that improve the sample complexity over previous techniques, and have less stringent assumptions. However, the algorithm proposed in the paper is not computationally practical, limiting the results to only theoretical interest. Nevertheless, the methods of the paper could be of interest to the DP community, and could be potentially used to develop new DP algorithms with high utility. Because I believe that this paper would be of interest to the DP community, I rate it 7 "good paper, accept".

Update: After reading the other reviews and the author feedback, my opinion of the paper is unchanged. My score remains 7.


**Time Spent Reviewing:**

2

---

> ### Author Response · Authors · 2021-08-10
> **Response**
>
> We thank all Reviewers for their thoughtful reviews and Reviewer ubCi for the detailed comments on notation and suggestions on the presentation of our paper, which we will address in preparing future versions.
>
> We agree that important future directions include extending this work to algorithms that are provably accurate under milder distributional assumptions, satisfy stronger privacy guarantees (such as RDP), and are computationally efficient. We note that although extending to RDP is an interesting direction, it would necessarily require a priori bounds on the parameters which would make the two results incomparable. In both algorithms, the preprocessing steps, that is, the “safety” check in the Tukey Depth mechanism and the projection to “goodness” in the Empirically Rescaled Gaussian mechanism, are the main obstacles to making these mechanisms computationally efficient.
>
> Clarifications:
> - $Y_{t,x}$ is indeed the support of $M_{\epsilon,t}(x)$, i.e., all points with Tukey depth at least t with respect to dataset x.
> - The superscript $\otimes n$ on a distribution $P$, as used in Theorem 3.2, means that the random variable consists of $n$ i.i.d. draws from $P$.
> - Tukey Depth: We will provide more explanation of why the Tukey depth is useful (both in classical/non-private settings and in prior work that uses it as part of the exponential mechanism). We had omitted that discussion due to space constraints.

---

### Official Review · Reviewer_8L9y · 2021-07-16

**Rating:** 8
**Confidence:** 4

**Summary:**

This paper investigates mean estimation of high dimensional Gaussian (and sub-Gaussian) distributions under the constraint of differential privacy (DP). More specifically, the authors aim at privately estimating the mean of a high-dimensional Gaussian in the Mahalanobis distance (w.r.t. to the covariance matrix).

One simple approach for this task is to first privately estimating the covariance matrix (say up to a constant factor in spectral distance) after which the problem has known solutions. Unfortunately, this approach is suboptimal as estimating the covariance matrix is harder task. Thus, the authors provide a way to circumvent the issue and construct simple estimators whose sample complexity scales only linearly with the dimension.

The first algorithm is based on combination of the propose-test-release (PTR) framework, exponential mechanism, and the concept of Tukey depth. Roughly speaking, the authors show that by sampling from the distribution defined by the exponential mechanism with a score function based on the Tukey depth restricted to a data dependent set of possible outputs (some points with high Tukey depth), one can get an accurate and private algorithm for mean estimation. In order to maintain privacy while restricting the output to be a data-dependent output set, the authors make use of the PTR framework to test whether the specific dataset is ``safe’’ for use. They then show that when sampled from Gaussian data, a dataset will be safe with high probability, proving that the algorithm will often output an estimate of the mean. Finally, upon success the output will w.h.p. be a point of high Tukey depth, so the estimator is also accurate.

The second algorithm relaxes the requirement of strict Gaussian data to sub-Gaussian data. This algorithm is also based on PTR framework, with a clever application of "skewed" Gaussian noise addition. The standard Gaussian mechanism used in DP adds Gaussian noise scaled equally in all directions. The authors propose to use an additive Gaussian mechanism (to the empirical mean) that adds noise proportional to the empirical covariance of the data instead. Intuitively, this preserves the scale of the data (and thus preserves accuracy w.r.t. to the Mahalanobis distance). While this is not private for worst case datasets, the authors circumvent this by using the PTR to first test whether the dataset is ``good’’ (appropriately defined), and if need be, project the dataset to the set of good datasets before running the algorithm. Finally, they show that with high probability, sub-Gaussian data will be good and there will be no projection, thus the algorithm will output the empirical mean (perturbed with skewed Gaussian noise), that will be accurate w.r.t. the Mahalanobis distance.


**Limitations And Societal Impact:**

not applicable.

**Main Review:**

The authors have applied techniques such as the exponential mechanism, propose test release framework, and additive Gaussian noise in very clever ways to prove new results for a fundamental problem in DP statistics. The arguments about the privacy and accuracy of the method are sound. The paper is well written and the high level discussions helps with understanding the big picture.

Before this paper, it was not known whether it is possible to estimate the mean of a Gaussian/sub-Gaussan distribution w.r.t. the Mahalanobis distance under the constraint of DP without first approximating the covariance. Non-privately, the empirical mean works well for this task. Thus, this paper is the first to prove that it is possible to achieve the same in the private case.

While the methods proposed in this paper are not computationally efficient, the approach is of broad interest to theoreticians and can be a starting point for the design of more practical approaches.


-----------
update after rebuttal: the authors did not specifically respond to my review and I am happy to keep the score after checking out the other reviews/responses.

**Time Spent Reviewing:**

5

---

### Official Review · Reviewer_bd7q · 2021-07-16

**Rating:** 8
**Confidence:** 3

**Summary:**

The authors considered the problem of estimating the mean of d-dimensional Gaussian and sub-Gaussian distribution under DP constraint. The error bounds for the mean estimate is given in terms of the Mahalanobis distance, however, interestingly enough, the proposed method does not estimate the covariance matrix. The tool proposed by the authors is a sampling mechanism which is based on Tukey depth: the probability of choosing a point is exponentially proportional to the negative Tukey depth. Then one can run a mean estimator on the sampled data. This alone is not enough to accomplish DP, therefore the authors make use of a propose-test-release framework which is basically checks whether the sampled data contains too many points for which the Tukey depth is too small. The proposed methods of order D^{2/3} and in addition, it does not assume anything about the covariance matrix, and does not need a prior estimate for the covariance matrix as previous methods required. The sub-gaussian case basically find the closest gaussian model and applies the same mechanism.


**Limitations And Societal Impact:**

There is a dedicated paragraph to address the societal impact.

**Main Review:**

I am on the positive side with this paper pretty much. It is really dense. Based on the main paper, one can have only a high level impression on how complex the technical details are. Nevertheless, the appendix is well-structured, and self-contained ( however I have not read all 42 pages of Appendix yet). The basic idea of the paper is very nice and novel as far as I know.


**Time Spent Reviewing:**

5

---

### Official Review · Reviewer_6nea · 2021-07-18

**Rating:** 7
**Confidence:** 4

**Summary:**

This paper considered the DP mean estimation problem with unknown covariance under Mahalanobis norm. In the known covariance setting, previous work can estimate gaussian mean near optimally. But in the unknown covariance setting, previous algorithms either need to pay an additional conditional number factor, or require d^3/2 samples to estimate the covariance which are both unsatisfactory. The key difficulty of the problem is getting Mahalanobis norm guarantee without first estimating the covariance privately.

The authors present two different (exponential time) algorithms for Gaussian and sub-Gaussian settings. The gaussian algorithm is based on an exponential mechanism with Tukey depth score. Although this approach has been used in [45] before, they need to choose the sampling region adaptively based on the data in order to achieve a good Mahalanobis guarantee. This brings additional difficulty in making the algorithm private, and they use the classical propose-test-release framework to obtain privacy guarantees. The second estimator uses the empirically rescaled Gaussian mechanism, where the gaussian noise is scaled by the empirical covariance such that Mahalanobis guarantee is naturally achieved. This estimator can be applied more generally on sub-gaussian distribution, though the sample complexity has a d/\alpha\eps^2 instead of the optimal d/\alpha\eps.


**Limitations And Societal Impact:**

Yes

**Main Review:**

In summary, this paper answered a fundamental question in DP and brings interesting new ideas along the way. I will be interested in the authors’ thoughts on the more efficient algorithms on this problem.


**Time Spent Reviewing:**

4

---

### Decision · Program_Chairs · 2021-09-27

**Decision:**

Accept (Spotlight)

**Comment:**

All reviewers agree that this paper provides a non-trivial advancement for the important problem of differentially private mean estimation. Reviewers 6nea, 8L9y, and ubCi found the proof techniques and algorithmic strategies insightful and novel. Reviewer ubCi found the assumptions underlying this paper to be significantly more realistic than prior work.  The only complaint about the paper is that the proposed algorithms have exponential running time, but reviewers 8L9y and ubCi feel that these algorithms could be the starting point for more practical algorithms. I therefore recommend that this paper be accepted.